# Efficient reduction-oxidation coupling degradation of nitroaromatic compounds in continuous flow processes

Yueshuang Mao[1,2,3], Bingnan Yu[1,3], Pengfei Wang[1], Shuai Yue [1] & Sihui Zhan [1] ✉

Nitroaromatic compounds (NACs) with electron-withdrawing nitro (-NO$_2$) groups are typical refractory pollutants. Despite advanced oxidation processes (AOPs) being appealing degradation technologies, inefficient ring-opening oxidation of NACs and practical large-scale applications remain challenges. Here we tackle these challenges by designing a reduction-oxidation coupling (ROC) degradation process in LaFe$_{0.95}$Cu$_{0.05}$O$_3$@carbon fiber cloth (LFCO@CFC)/PMS/Vis continuous flow system. Cu doping enhances the photoelectron transfer, thus triggering the -NO$_2$ photoreduction and breaking the barriers in the ring opening. Also, it modulates surface electronic configuration to generate radicals and non-radicals for subsequent oxidation of reduction products. Based on this, the ROC process can effectively remove and mineralize NACs under the environmental background. More importantly, the LFCO catalyst outperformed most of the recently reported catalysts with lower cost (13.72 CNY/ton) and higher processing capacity (3600 t/month). Furthermore, the high scalability, material durability, and catalytic activity of LFCO@CFC under various realistic environmental conditions prove the potential ability for large-scale applications.

Nitroaromatic compounds (NACs) are vital raw materials or intermediates in industrial activities, which have been versatilely applied for dye, pesticide, plastic, and medicine manufacturing, with approximately 65,000 kinds of high-concentration NACs being discovered in industrial wastewater[1,2]. The presence of the strong electron-withdrawing nitro groups (-NO$_2$) induces the overall delocalization effect of π electrons in the benzene ring structure, thus enhancing the stability of the benzene ring and making NACs difficult to oxidize[3]. Meanwhile, NACs are refractory pollutants with the feature of carcinogenic, teratogenic, and mutagenic that pose threats to environmental safety and human health, which have been listed as priority pollutants by the United States Environmental Protection Agency[4,5]. Therefore, NAC-contained industrial wastewater has the characteristics of high salinity, high concentration, complex composition, high

toxicity, and poor biodegradability. Its remediation is still a difficult task. Physical adsorption or extraction results in secondary pollution and biodegradation takes a longer time and is sensitive to the impact load of NACs and environmental changes[6,7]. In a more environmentally friendly approach, chemical oxidation can degrade NACs, but the direct oxidation and complete mineralization of parent NACs is kinetically limited by the unique -NO$_2$ and the water quality conditions, leading to incomplete mineralization with low efficiency and high cost[8,9]. Therefore, developing efficient approaches suitable for practical application is a great challenge.

For the first challenge, recent studies have reported that the reduction of -NO$_2$ to aniline groups (-NH$_2$) via reductants can weaken the key structural units of -NO$_2$ and decrease the chemical stability of NACs, thereby breaking the barriers in benzene

[1]Key Laboratory of Pollution Processes and Environmental Criteria (Ministry of Education), College of Environmental Science and Engineering, Nankai University, Tianjin, China. [2]College of Resources and Environment Science, Shanxi University, Taiyuan, China. [3]These authors contributed equally: Yueshuang Mao, Bingnan Yu. ✉e-mail: sihuizhan@nankai.edu.cn

ring-opening[10–12]. Nevertheless, a single reduction process could not remove NACs safely from an environmental perspective for there are possibilities to form more toxic intermediates (e.g., arylamines and azo/oxygen compounds) than parent NACs and a reduction process that gives electrons difficulty to further degrade and mineralize the reduction products, resulting in incomplete degradation and bringing greater environmental risks[13]. As a result, both single reduction or oxidation methods can not simultaneously implement the efficient degradation of reduction products and complete mineralization of parent NACs. Therefore, constructing reduction-oxidative coupling degradation processes are effective strategy to solve the first challenge. Generally, advanced oxidation processes (AOPs) make it possible for the simultaneous occurrence of reduction and oxidation reactions by producing highly active reductive species (electron and H*, etc.) and oxidative species (hole and •OH, etc.) at the same time[14,15]. For example, zero-valence metal-Fenton-like technology and electrochemical technology adopt two-step reactions with separated reduction and oxidation systems, which are difficult to operate and sensitive to pH[3,11]. In comparison, a single catalytic system with the advantages of being easy to operate and low cost, however, the mutual consumption of reductive/ oxidative species needs to be solved (e.g., photocatalysis electron-hole pairs)[16]. For another challenge, the current technologies are almost •OH-based oxidation with higher oxidation ability (9−2.7 V vs NHE) but indiscriminateness, they are easily quenched by water matrix in industrial wastewater, resulting in ineffective NACs removal rate and limiting the large-scale application of AOPs[17]. Currently, non-radical oxidation processes with relatively weaker oxidation ability (e.g., $^1O_2$, 0.81 V vs NHE) have been extensively researched due to their specificity to electron-rich groups and higher resistance to environmental interference[18]. Therefore, one promising strategy for solving the second challenge is to make use of the complementary advantages of radicals and non-radicals. To sum up, it is urgent to develop a coupling of reduction and oxidation processes in one system and synergistically generate radical and non-radical for efficiently removing NACs in industrial wastewater.

Herein, we designed a LaFe$_{0.95}$Cu$_{0.05}$O$_3$@carbon fiber cloth (LFCO@CFC) fixed reaction bed (FRB) to construct a reduction-oxidation coupling (ROC) degradation of NACs in PMS/Visible light (Vis) system. The primary engineering parameters of the continuous flow process were evaluated and optimized at first. Then, we find that Cu substituting induces the super-exchange effect and promotes the transfer of photoelectrons, which can reduce -NO$_2$ to initiate o-nitrophenol (ONP) degradation, also, it can modulate local electronic configuration to form electron-poor/rich sites for PMS dual activation to synergetically generate radicals and non-radicals in the subsequent oxidation process. Finally, the LFCO/PMS/Vis system achieved ROC degradation of ONP with a degradation rate of 0.079 min⁻¹, which is 7.7 times higher than that of LFO (0.009 min⁻¹). More importantly, LFCO@CFC has lower cost and higher efficiency compared with peer catalysts. Also, it can prevent the leaching of catalysts and provide long-term material stability, thus exhibiting practical advantages for large-scale NACs degradation.

## Results

### Overview of fixed reaction bed (FRB) reactor

FRB reactors have been a predominant approach for scale-up application, which can provide a more effective catalytic system than the suspended system with collective advantages of continuous flow, catalyst recovery, and low cost[19,20]. The schematic and photo of our flow-through wastewater treatment system is shown in Fig. 1A, B. The thoroughly mixed ONP and PMS solution flows through a FRB reactor by a peristaltic pump via consuming electricity, where the organic contaminants are degraded under Xe light irradiation before the effluent is discharged. The FBR reactor is the heart of this continuous flow system, which consists of the Cu-doped LaFeO$_3$ (LaFe$_{1-x}$Cu$_x$O$_3$, $x_{molar\ ratios}$ = 0, 0.01, 0.03, 0.05, 0.1) and carbon fiber cloth (CFC). First, LaFe$_{1-x}$Cu$_x$O$_3$ was synthesized via sol-gel methods (Supplementary Fig. 1)[21]. X-ray diffraction (XRD) results present the successful doping of Cu atoms into the lattice without forming any impurity (Fig. 1E). Using CFC as carrier materials due to its rich transport pathways, non-pollution potential, and low maintenance costs (Supplementary

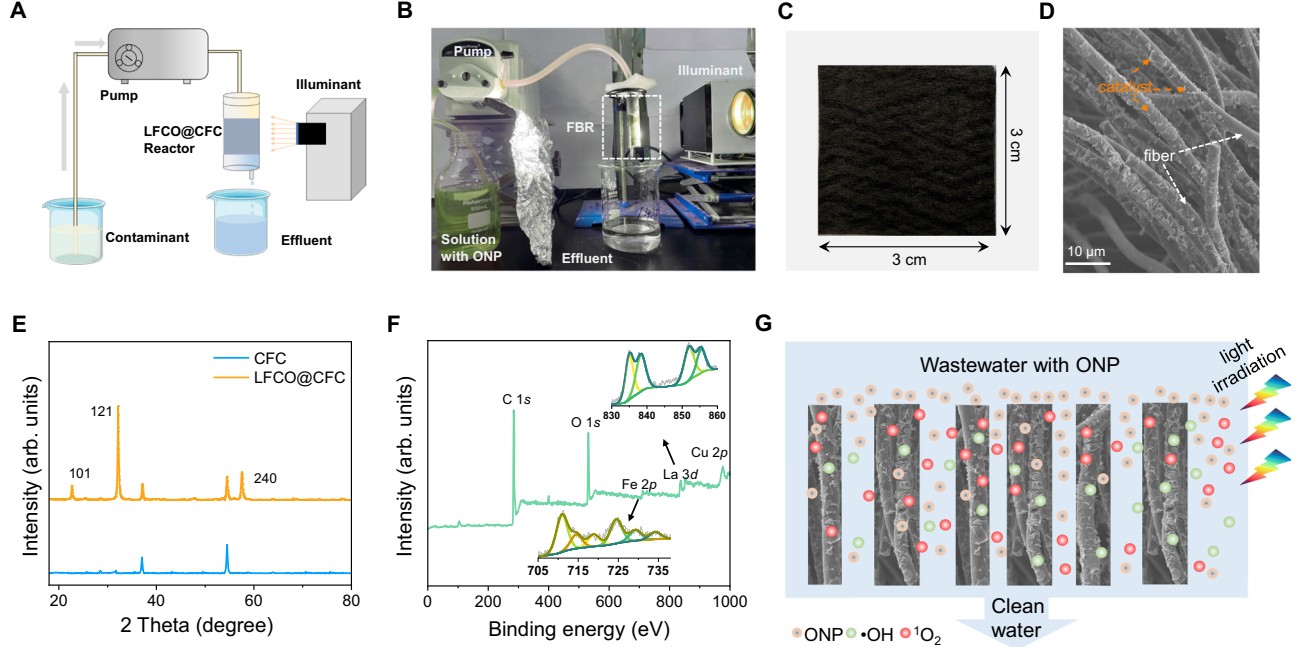

**Fig. 1 | Practical applications of FRB. A** Schematic diagram of the FRB. **B** Photograph of experiment device (ONP as indicator pollutant). Photograph (**C**) and SEM image (**D**) of the LFCO@CFC. **E** XRD of CFC and LFCO@CFC ('a.u.' represents the arbitrary units). **F** XPS spectra of the overall survey of LFCO@CFC. **G** The simplified schematic showing cross-section view of ONP, •OH, and $^1O_2$ in the photocatalytic membrane.

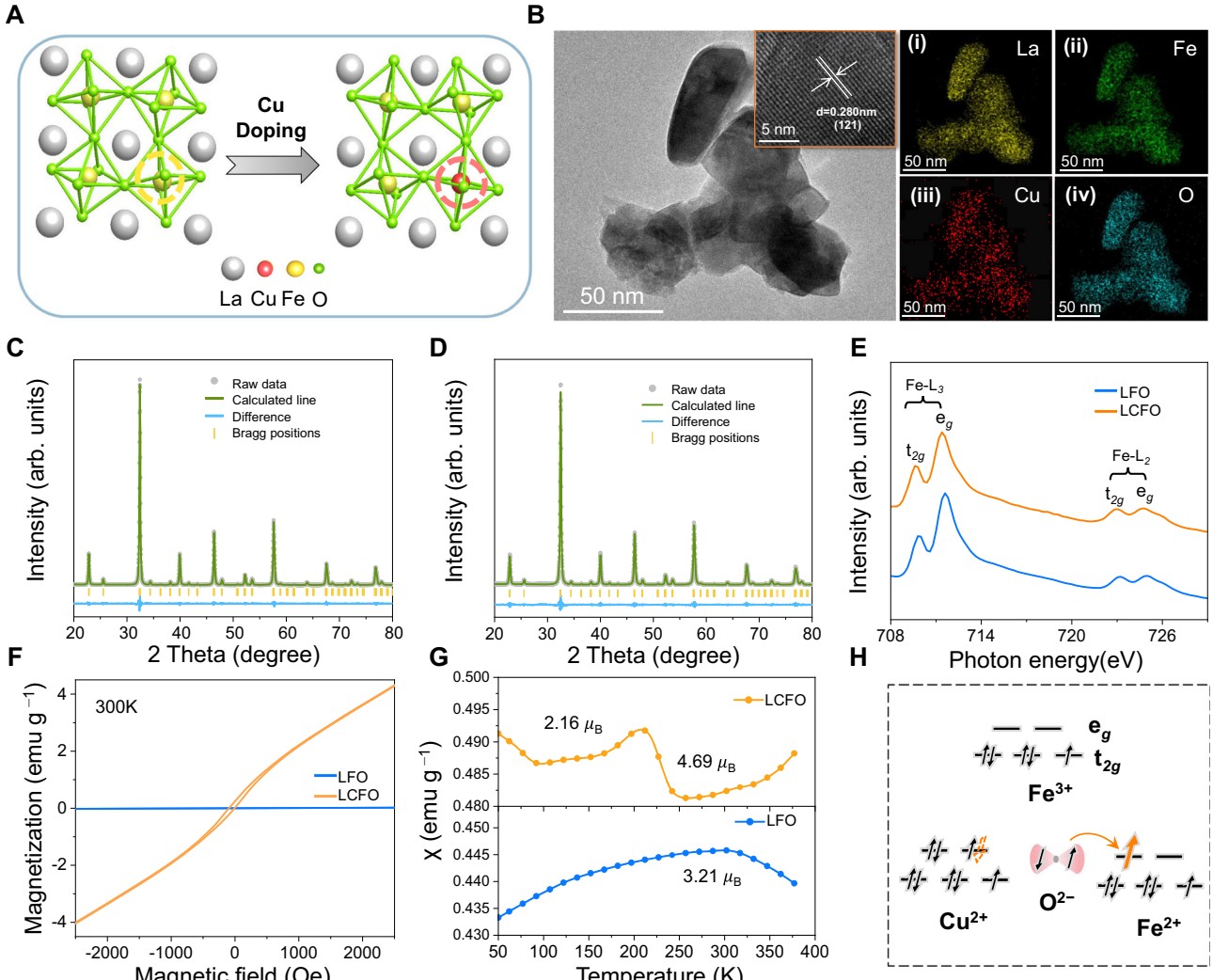

**Fig. 2 | Structural characterizations. A** Schematic presentation of LFO and LFCO perovskite structure. **B** TEM image of LFCO in 50 nm (inset: TEM in 5 nm with lattice fringe). The EDS mapping images of La (**i**), Fe (**ii**), Cu (**iii**), and O (**iv**) in LFCO. Refined XRD of LFO (**C**) and LFCO (**D**). **E** Normalized Fe $L_{2,3}$-edge XAS spectra of LFO and LFCO. **F** Magnetic hysteresis loop of LFO and LFCO. **G** Temperature dependence of magnetization (M − T) curves under ZFC of LFO and LFCO **H** Schematic illustration of the super-exchange interaction in LFO and LFCO.

Fig. 2)[22,23]. Finally, the FRB was prepared via in-situ growing methods. Before FBR reactor operation, beaker experiments were conducted first to explore the optimal reaction system. In dark conditions, all catalysts reached adsorption equilibrium in 30 min with <20% adsorption capacity of ONP, which confirms the catalytic degradation mainly contributes to the ONP concentration decrease. In addition, experiments without PMS, catalysts, or light irradiation have negligible degradation ability (Supplementary Figs. 3−4). Under light irradiation, $LaFe_{0.95}Cu_{0.05}O_3$ (LFCO) shows the highest ONP removal rate and was chosen for further study here. The photo of the as-prepared LFCO@CFC is shown in Fig. 1C. Scanning electron microscopy (SEM), XRD, and X-ray photoelectron spectroscopy (XPS) analysis of LFCO@CFC validate their morphology and composition (Fig. 1D−F and Supplementary Fig. 5). Without changing the basic properties and characteristics of CFC, the catalyst load can improve the adsorption capacity of the FBR, to improve the degradation effect of the reaction bed on pollutants (Supplementary Fig. 6). The particulate LFCO covers the entire surface of the CFC, and the chemical state of various elements on the surface of the CFC was studied by XPS analysis, and the results also indicated that La, Fe, and Cu are co-existed in the LFCO@CFC. All the above results showed that LFCO was successfully loaded on the CFC. After 7 days of cycling test, the weight loss rate of

LFCO@CFC is only 2%. The leached heavy metal ions will not cause environmental pollution and have high strength stability. Figure 1G shows the anticipated enhancement of contact among catalyst, ONP molecules, and ROS due to the higher catalyst-loading capacity and mass transfer ability of CFC, thus gaining superior performance and ultralong cycling life.

## Structural characterizations of LFCO

The substitution of Cu at the B site (Fig. 2A) is expected to yield a unique super-exchange effect that is advantageous for PMS activation and pollutant degradation[24]. Therefore, the structural contribution of Cu species in LFCO was investigated first. The overall morphology of LFCO and LFO were studied by transmission electron microscopy (TEM) (Fig. 2B and Supplementary Fig. 7). They all show similar nanoparticle morphology with a clean surface, consisting of the Brunauer-Emmett-Teller (BET) results (Supplementary Fig. 8 and Supplementary Table 1). High-resolution (HR)-TEM image of LFCO (Fig. 2B inset) shows the lattice distance expanded to 0.28 nm as compared to LFO (0.278 nm). In addition, the corresponding energy-dispersive X-ray spectroscopy (EDS) mapping images (Fig. 2B i- iv) demonstrate a homogeneous distribution of all the constituent elements including La, Cu, Fe, and O, and the content of Cu is 4.43%, close

to the theoretical value of 5% (Supplementary Table 2). Furthermore, as shown in Fig. 2C, D, the Rietveld refinement XRD pattern of LFCO shows similar diffraction peaks with LFO, which can be well labeled as orthorhombic perovskite structures (*Pnma*) (JCPDS# 37-1493). As expected, the prominent peaks of LFCO shift slightly to lower angles, and the volume of a unit cell was enlarged to 244.04 $Å^{-3}$ (243.74 $Å^{-3}$ for LFO, Supplementary Table 3), suggesting the smaller $Fe^{3+}$ ($r = 0.65$ Å) is partially replaced by larger $Cu^{2+}$ ($r = 0.73$ Å) at B-sites and forms Fe-O-Cu sites (Fig. 2A, Supplementary Fig. 9 and Supplementary Data 1). To determine the internal chemical environments of Cu in LFCO, we conducted an XPS analysis. XPS survey spectra confirm the existence of Cu elements in LFCO with Cu/Fe ratios close to 0.05:0.95 (Supplementary Fig. 10a and Supplementary Table 4). As shown in the O $1s$ spectra (Supplementary Fig. 10b), peaks at 528.7 eV, 529.2 eV, and 531.3 eV were assigned to La-O, Fe-O, and hydroxyl groups in LFO, respectively[25]. Remarkably, LFCO showed an additional peak (529.1 eV) between La-O and Fe-O, which was assigned to Cu-O in the Fe-O-Cu unit[26]. Raman and infrared (IR) spectroscopy were further analyzed to reveal the construction of Fe-O-Cu sites. The characteristic Raman peaks of LFO appear at 290 $cm^{-1}$, 432 $cm^{-1}$, and 690 $cm^{-1}$ representing the $A_{1g}$ mode vibration of Fe-O bonds $B_{3g}$ mode bending vibration of $FeO_6$ and the structural Jahn-Teller distortions around $Fe^{3+}$ ions (Supplementary Fig. 11). Different from LFO, a new peak of LFCO appears at 556 $cm^{-1}$ representing the $A_{1g}$ stretching vibration of Cu-O bonds. This reveals the co-existence of Fe-O and Cu-O vibration in LFCO, which can be confirmed by the Cu-O peak in FTIR data (Supplementary Fig. 12). Furthermore, the peak intensity at 690 $cm^{-1}$ increased due to the reduction of the cubic symmetry of the $FeO_6$ octahedra after Cu substitution. All the above results demonstrate the successful construction of LFCO catalyst with asymmetry Fe-O-Cu units.

## Electronic distribution of LFCO

The electronic structure of LFO and LFCO was obtained from XPS spectra and soft X-ray absorption spectra (XAS) measurements. Figure 2E shows that Fe L-edge XAS spectra of LFO and LFCO possess $L_2$ and $L_3$ doublets, which are caused by the electron excitation from Fe $2p1/2$ and $2p3/2$ to unoccupied $3d$ orbitals, respectively. Both of these doublets split into $e_g$ and $t_{2g}$ peaks due to the splitting of the crystal field with octahedral symmetry[27]. Furthermore, for LFCO, all peaks shift to lower energy and show broader asymmetry as compared to pristine LFO, implying the reduction of the Fe valence state after low-level Cu doping. Besides, an obvious positive shift for the Fe $2p$ XPS spectra of LFCO relative to LFO was observed, indicating an increased electron density around Fe sites after Cu doping and a decreased valence state from $Fe^{3+}$ to $Fe^{2+}$ (Supplementary Fig. 13)[28]. This can be intuitively shown by the charge density difference, where electrons transfer from adjacent Cu to Fe via O atoms (Supplementary Fig. 14).

Magnetism measurement offers a beneficial platform to get a detailed insight into the nature of charge redistribution. As shown in the field dependence of the specific magnetization (M−H) curve at 300 K (Fig. 2G), both LFO and LFCO show typical hysteresis loops and represent ferromagnetic (FM) behavior, which is caused by spin canted of $Fe^{3+}$ as the source of the magnetic moments, while a stronger FM coupling for LFCO is due to the further disordering spins induced by Cu doping[29,30]. Furthermore, Zero-field cooling temperature-dependent susceptibility (M-T) curves were measured, as shown in Fig. 2H. LFO exhibits a phase transition at ~320 K, which originates from a static interaction of $Fe^{3+}$-$O^{2-}$-$Fe^{3+}$. For LFCO, an additional phase transition is observed at ~103 K, which is caused by the vibronic super-exchange interaction of $Fe^{2+}$-$O^{2-}$-$Cu^{2+}$[31–33]. Moreover, the effective magnetic moments ($\mu_{eff}$) and the unpaired d electron number (n) were calculated according to the literature[34]. Results show that the $\mu_{eff}$ of Fe in LFCO increases to 4.69 $\mu_B$ as compared with LFO (3.21 $\mu_B$) due to dominant $Fe^{2+}$-$O^{2-}$-$Cu^{2+}$ interactions. In addition, the n of Fe in LFCO (3.8) is larger than that in LFO (2.6), indicating the electrons transfer

from $Cu^{2+}$ to $Fe^{2+}$ via O $2p$ orbital (Fig. 2H right). As a result, the unusual super-exchange interaction successfully induces the directed migration of charge and results in the formation of electron-poor Cu sites and electron-rich Fe sites. Generally, single electron transfers from metal sites to PMS can generate radicals and the opposite pathway to generate non-radicals[24]. Thus, the Fe-O-Cu unit is expected to promote the synergetic production of radicals and non-radicals for practical application.

## LFCO@CFC FRB system operation

The operational parameters of the FBR system (water flux, hydraulic residence time (HRT), the dosage of catalyst and PMS, and light intensity) were investigated to understand the variation in catalytic activity of the FBR system and determine the limiting factors therein.

Water flux and residence time. The increase in water flux indicates that the molar flux of pollutants increases, representing more ONP can be brought to the reactor. As shown in Fig. 3A, the removal efficiency and the apparent rate constant ($k_{obs}$) followed a similar trend when increasing the water flux. The removal efficiency of ONP was 99.123% at a low flux value (1500 mL $h^{-1}$). When the water flux increased from 1500 mL $h^{-1}$ to 7500 mL $h^{-1}$, the removal efficiency was relatively stable. However, with the further increase of water flux to 15000 mL $h^{-1}$, removal efficiency sharply decreased to 69.665%. Therefore, the optimal water flux parameter proposed for the FBR system is 7500 mL $h^{-1}$. The HRT is another limiting factor of the FBR system, which greatly affects the mass transfer rate. In Fig. 3B, the removal efficiency and the apparent rate constant ($k_{obs}$) followed the opposite trend when increasing the HRT. Though higher $k_{obs}$ can be achieved at extremely short HRT (0−50 s), the removal efficiencies are low due to the limited pollutant diffusion. This results in substandard effluent quality. The prolonged HRT increases the interaction time between ROS and ONP, thus a higher ONP removal efficiency (more than 99%) was achieved within a residence time of 60−90 s. On this basis, the hydraulic retention time is set to 60 s with relatively high $k_{obs}$. To sum up, the water flux and HRT were set to 7500 mL $h^{-1}$ and 60 s, respectively, to optimize the degradation effect and reduce the cost of this process.

The dosage of catalyst and PMS, and light intensity. Actual degradation rate (r″) was defined taking the mass transfer into account and used as the indicator here. Firstly, by changing the amount of chemicals, reaction beds loaded with different amounts of catalysts were prepared. Figure 3C (left) shows that when the weight of the LFCO increased to 1.5 g, the highest r″ was observed, this may be due to the increase of active sites to activate PMS. However, the r″ cannot be significantly improved by increasing the weight of the LFCO. Similarly, the r″ increased gradually by increasing the PMS concentration from 1 mM to 3 mM, causing more ROS can be generated by PMS activation. Further increasing the PMS concentration to 4 mM did not enhance the r″ due to the saturation of active sites. Therefore, LFCO@CFC-1.5 and 3 mM PMS are selected for the FBR system. Light intensity defines the number of photons available to the photocatalyst and determines the number of produced electron-hole pairs and reactive species[35]. Initially, the r″ increased with the increase of light intensity from 0 W to 15 W and then remained unchanged above 15 W. In this case, the light intensity is optimized at 15 W from the perspective of energy and cost saving. Unexpectedly, LFCO also shows 100% degradation ability within 40 min under near-infrared (NIR) light while that is absent for LFO (Supplementary Fig. 15). Finally, under optimum conditions, LFCO with asymmetry Fe-O-Cu units realize 100% removal of ONP with $k_{obs}$ reaching 0.079 $min^{-1}$, which is 7.7 times higher than that of LFO ($k_{obs} = 0.009$ $min^{-1}$) (Fig. 3D and Supplementary Fig. 16). Moreover, the small difference between $k_{obs}$ and r″ reflects that the process design is reasonable. Besides, the degradation efficiency of ONP remained at >99% in 7-days continuous experiments with low metal leakage (Fig. 3E and Supplementary Table 10). The similar XRD, SEM, TEM, and XPS

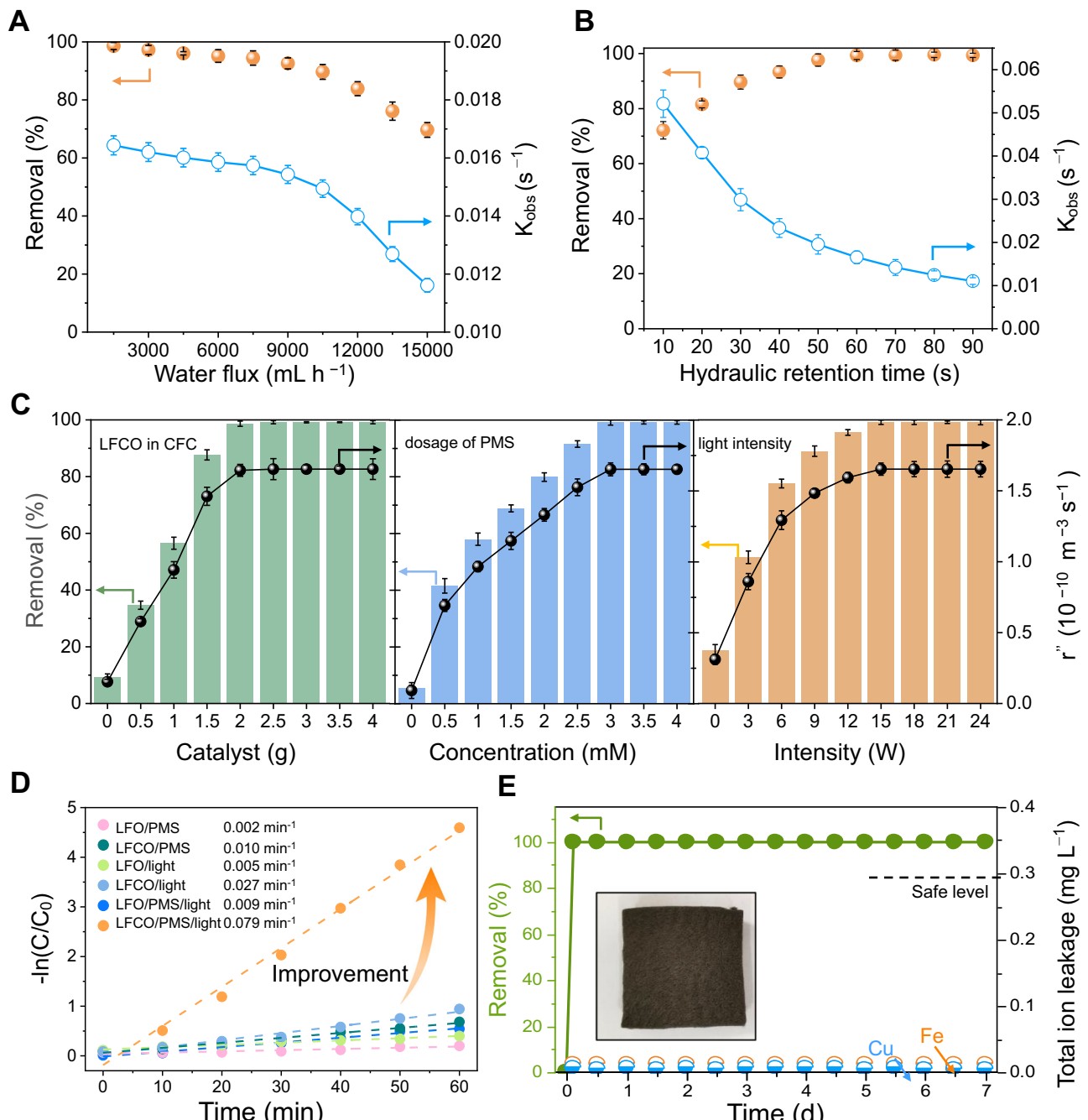

**Fig. 3 | Removal of ONP under different operational parameters.** The removal efficiency and $K_{obs}$ of ONP were investigated by changing water flux (**A**) and hydraulic retention time (**B**). **C** The removal efficiency and r″ of ONP were investigated for varying the weight ratio of LFCO vs CFC, the dosage of PMS, and visible light intensity in the FRB system. **D** The degradation efficiency of ONP in different systems. **E** Long-term operation performance of the LFCO@CFC FRB system (Illustration is the photograph of the used LFCO@CFC). FBR conditions (unless indicated otherwise): λ > 420 nm radiation, Water flux 7500 mL h⁻¹, HRT 60 s, 298 K, initial pH 7.1, 30 ppm ONP, and 3 mM PMS (introduced with the ONP stock solution). Error bars are standard error values of three tests ($n = 3$).

results of fresh and LFCO (Supplementary Figs. 17, 18) together proved the stability and durability of the catalyst.

### Role of Fe-O-Cu units in PMS activation

To illustrate the origin of high catalytic performance behind LFCO under light irradiation, the optical properties and band structures were determined first by ultraviolet-visible diffuse reflectance spectra (UV-Vis DRS), XPS and UV photoelectron spectroscopy (UPS) (Fig. 4A and Supplementary Figs. 19–21). Compared with the LFO, LFCO exhibits enhanced light absorption capability from Vis to NIR wavelength and the newly formed dopant-related mid-gap states (MS) due to the step-like absorption tail extending to 900 nm (Supplementary Fig. 19). According to the Kubelka-Munk method, the bandgaps of LFCO decreased to 2.25 eV comparing with that of LFO (2.36 eV)[36]. Moreover, a new bandgap of 1.68 eV was also identified, corresponding to the new charge-transfer transitions opened up by introducing Cu atoms (Fig. 4A)[37]. Calculated by valence band (VB) XPS and UPS, the schematic diagram of band structures is shown in Fig. 4A and the projected electronic density of states (DOS) of both materials was then investigated (Supplementary Fig. 22). We find that the doped Cu contributes to the decrease of bandgap and its *d* states induce significant peaks, which significantly extends the VB bandwidth and concentrates the

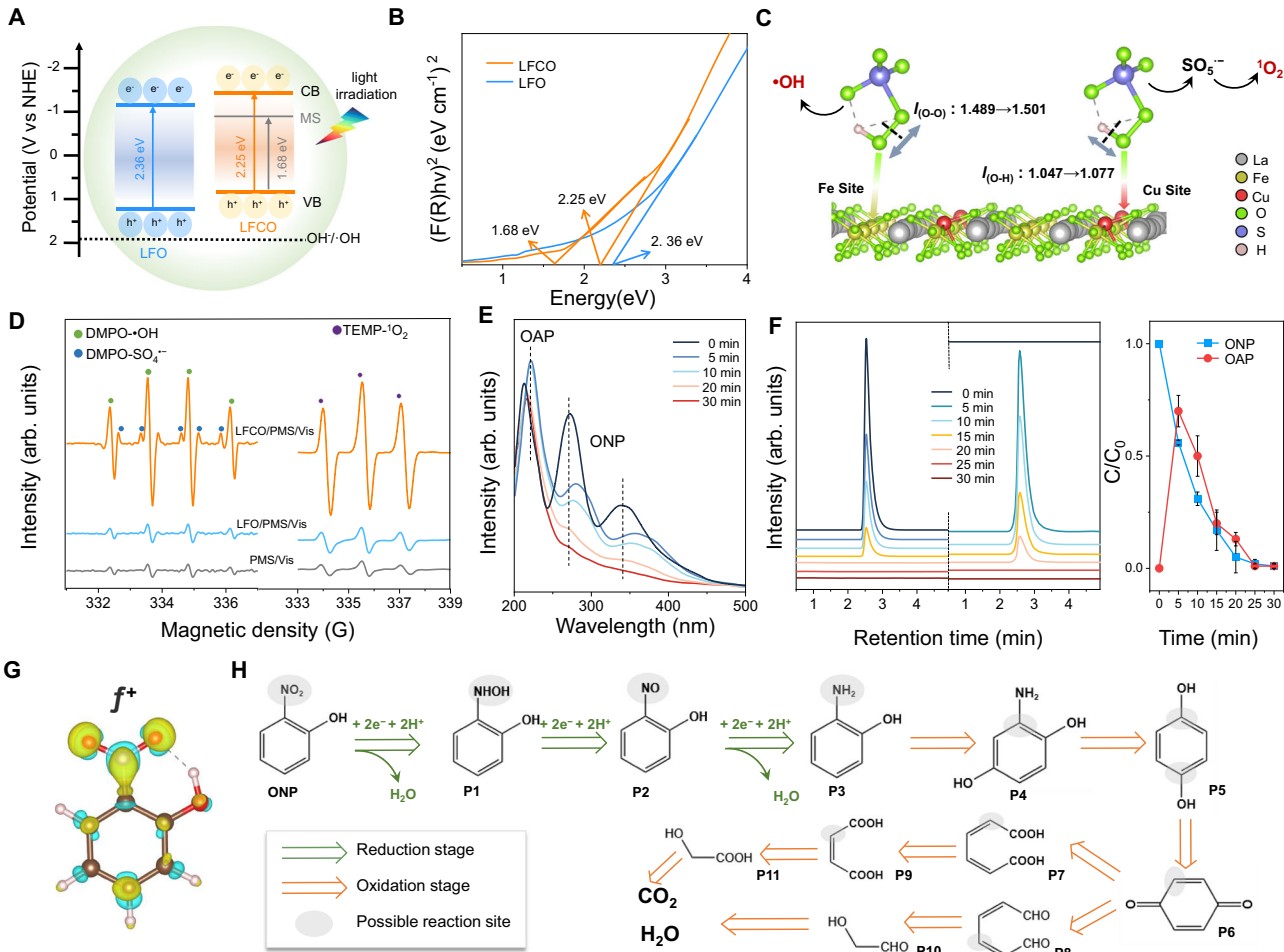

**Fig. 4 | ROC mechanism and degradation pathway of ONP. A** The band gap diagram of LFO and LFCO. **B** Tauc plots for LFO and LFCO using $(F(R)h\nu)^2$ (Kubelka-Munk parameter) as a function versus the photon energy (The x-axis intersection point of the linear fit of the Tauc plot gives an estimate of the band gap energy). **C** Proposed mechanism for the PMS reaction on the surface of LFCO. **D** EPR spectra of DMPO-•OH/DMPO-SO$_4^{•-}$ and TEMP-$^1O_2$ in PMS/Vis, LFO/PMS/Vis, and LFCO/PMS/Vis system, respectively. **E** In-situ UV-Vis absorption spectra of ONP degradation in the LFCO/PMS/Vis system. **F** HPLC spectra and corresponding concentration changes of ONP and OAP during degradation in the LFCO/PMS/Vis system. **G** Fukui index ($f^+$) of ONP. (Yellow and cyan regions represent the electron accumulation and the electron depletion, respectively). **H** The degradation pathway of ONP in the LFCO/PMS/Vis system. The green arrow represents the reduction stage induced by e$^-$, The green arrow represents the oxidization stage induced by •OH and $^1O_2$. Conditions: $\lambda > 420$ nm, 50 mL 30 ppm ONP solution, 0.3 mM PMS, 0.3 g·L$^{-1}$ powder catalyst, 298 K, initial pH 7.1.

DOS near the VB maximum, resulting in extremely strong photoabsorption[38]. More importantly, Cu states act as trapping sites for photogenerated carriers and can prevent them from rapid recombination, thus promoting electron transfer and photocatalytic reactions[39,40].

The intrinsic carrier-transport characteristics were then studied by surface photovoltage spectroscopy (SPV) (Supplementary Fig. 23). The peaks of LFCO at ~370 nm and ~475 nm corresponding to the electronic transition from VB to CB and the transition from the VB to the MS and/or from the MS to the CB, respectively[41]. Upon light irradiation, the higher SPV response of LFCO over LFO indicates an effective separation of photogenerated charges, which can be further proved by steady-state photoluminescence spectra (PL) and transient photocurrent response (Supplementary Figs. 24, 25). Thereafter, in-situ synchronous illumination XPS at different exposure times were tested to observe dynamic electron transfer of LFCO (Supplementary Fig. 26). Accompanied by Vis light irradiation, the intensity of O 2*p* peaks related to the M-O bond was enhanced, indicating the stronger interaction between metal and oxygen atoms during the reaction. To be specific, the peaks of Fe 2*p* spectra shift toward low binding energy (from 710.2 eV to 709.95 eV) while peaks of Cu 2*p* spectra shift to high binding energy (from 931.75 eV to 932.38 eV), indicating that Fe receives electrons and Cu donors electrons in the Fe-O-Cu unit via the super-exchange effect. Additionally, the specific charge carrier dynamics were investigated based on the transient PL spectra which can be fitted by biexponential decay kinetics (Supplementary Fig. 27 and Supplementary Table 8). The observed average lifetime for LFCO is 3.05 ns, which is much longer than the 2.15 ns for LFO, suggesting the higher efficiency of charge separation under light irradiation with a slower recombination rate and more photoinduced electrons can participate in the ONP degradation[42]. Moreover, the electrochemical impidence spectroscopy (EIS) measurements and Mott-Schottky (M-S) plots together reveal that LFCO exhibits higher electrical conductivity and higher apparent charge carrier concentrations than LFO, confirming the accelerated photoexcited charge carrier transfer dynamics (Supplementary Fig. 28). The enhanced charge transfer can be further evidenced by the Hall effect measurement at room temperature. As shown in Supplementary Table 9, the semiconductive nature of LFCO with a higher carrier concentration ($4.45 \times 10^{15}$ cm$^{-3}$) and carrier mobility (70.7 cm$^2$ V$^{-1}$ s$^{-1}$) compared with LFO ($3.62 \times 10^{15}$ cm$^{-3}$ and 61.3 cm$^2$ V$^{-1}$ s$^{-1}$). Overall, the above photophysical studies unambiguously evidence the important role of the super-

exchange effect of the Fe-O-Cu unit in enhancing light absorption, charge separation, and transfer abilities of LFCO.

## PMS activation mechanism

We further built related models to stand for Fe-O-Cu units in LFCO and investigate PMS adsorption energies and configurations during the reaction. The oxygen atom of O-O group in PMS is more easily adsorbed at Fe/Cu sites in LFCO with lower adsorption energy ($E_{ads}$) compared with LFO, which also confirms the metal sites are catalytic sites (Supplementary Fig. 29 and Supplementary Table 6). As shown in Fig. 4c and Supplementary Table 7, when PMS uniquely adsorbed over Fe sites in LFCO, the length of the O-O bond ($l_{O-O}$) changed from 1.35 Å to 1.50 Å, implying the homolytic cleavage of O-O bond at Fe sites to generate •OH (Eq. (2)). Simultaneously, the elongated O-H bond (from 1.05 Å to 1.08 Å) at Cu sites (Eq. (3) ~ (4)) favors the cleavage of O-H to form $^1O_2$[43]. This can be further confirmed by the strongest EPR signal by using 2, 2, 6, 6-tetramethyl-4-piperidine (TEMP) and 5,5-dimethyl-1-pyrroline-N-oxide (DMPO) as spin-trapping agents for $^1O_2$ and •OH/$SO_4^{•-}$/•$O_2^-$ (Fig. 4d and Supplementary Figs. 30, 31), which is beneficial for higher ONP degradation performance[44]. Meanwhile, voltammetry (CV) and linear sweep voltammetry (LSV) experiments were conducted to evaluate the electron transfer between PMS and LFCO sample (Supplementary Fig. 32). The capacitance and current density after adding PMS are larger than that of the pristine LFCO, suggesting that more electrons transfer between PMS and LFCO[45,46].

Furthermore, in-situ Raman measurements and attenuated total reflectance Fourier transform infrared (ATR-FTIR) spectroscopy were used to capture the signals of the PMS intermediate (Supplementary Fig. 33). Three prominent Raman bands ascribed to $SO_3^-$, $SO_4^{2-}$ and $HSO_5^-$ groups in PMS molecular at 1059, 981, and 883 cm$^{-1}$, respectively, were observed[47]. The concomitant decreasing intensity of all peaks with the irradiation time implied the adsorption and rapid decomposition of PMS in the LFCO system, which can be further evidenced by decreased PMS concentration within 30 min (Supplementary Fig. 34)[48]. Moreover, the splitting of the $SO_3^-$ peak (1065 cm$^{-1}$) into two small peaks at 1071 and 1055 cm$^{-1}$, suggests that bidirectional electron transfer occurs between PMS and LFCO[49]. These computational and experimental results justify the efficient and synergetic radical/ non-radical generation ascribed to the unique PMS activation mechanism on LFCO with the feasibility of employing this system in actual wastewater.

$$LECO + h\nu \rightarrow e^- + h^+ \tag{1}$$

$$HSO_5^- + e^- (Fe\ site) \rightarrow •OH + 2SO_4^{2-} \tag{2}$$

$$HSO_5^- + e^- (Cu\ site) \rightarrow SO_5^- + H^+ \tag{3}$$

$$SO_5^- + SO_5^- \rightarrow 2SO_4^{2-} + {}^1O_2 \tag{4}$$

## Detecting ONP reduction and oxidization products

To reveal the variation of functional groups during ONP degradation in the LFCO/PMS/Vis system, UV-Vis spectra were recorded. According to the previous literature, ONP displays three characteristic bands, the more intense bands at 210 and 278 nm belong to the phenolic group while the band observed at 350 nm accounts for $\pi \rightarrow \pi^*$ transition of the -$NO_2$ group[50], while the characteristic absorption peaks at 230 and 285 nm are attributed to o-aminophenol (OAP), which is the reduction product of ONP (Supplementary Fig. 35)[51]. Interestingly, the absorption band at 210 nm slightly shifted to 230 nm with increased intensity within 10 min only in LFCO/PMS/Vis system, suggesting the generation of OAP at the initial stage (Fig. 4E and Supplementary Fig. 36a, b).

Meanwhile, the absorption band at 278 and 350 nm gradually disappeared, inferring that the aromatic ring was opened and mineralized as the reaction time increased. The variation of ONP and OAP concentration was further detected by HPLC (Fig. 4F and Supplementary Fig. 36c, d). Upon the decline of the ONP concentration, the formation of OAP was observed immediately (within 5 min). Notably, negligible concentration of both ONP and OAP was achieved upon completion of the reaction, which indicates complete oxidative degradation. This can be further confirmed by 86% of the TOC removal rate (Supplementary Fig. 37). In addition, the above phenomena also occur under NIR irradiation (Supplementary Fig. 38). Thus, ONP may undergo a sequential ROC pathway (ONP → OAP → $CO_2$ + $H_2O$) only in the LFCO/PMS/light system.

## Reduction-Oxidation Coupling (ROC) degradation of ONP

To understand how different ROS affect the ONP degradation process, scavenger quenching tests were conducted. Methanol (MeOH), superoxide dismutase (SOD), carotene, tert-butanol (TBA), and potassium dichromate ($K_2Cr_2O_7$) were used as quenchers of $SO_4^{•-}$, •$O_2^-$, $^1O_2$, •OH, and e$^-$, respectively[13,52]. As depicted in Supplementary Fig. 39, e$^-$, •OH and $^1O_2$ play a vital role in ONP removal and the quantified contributions are 26%, 32%, and 36%, respectively. Furthermore, the recorded UV-Vis spectra show that both the reduction and oxidization of ONP are inhibited after quenching e$^-$ while quenching •OH and $^1O_2$ merely influences the oxidization process (Supplementary Fig. 40). This result indicates that reduction is the first step for ONP degradation. Therefore, a rational ONP degradation pathway catalyzed by LFCO is proposed: e$^-$ initiates the ONP reduction process to form unstable OAP, •OH and $^1O_2$ dominate the OAP oxidization process.

Combined with DFT calculations and the high-performance liquid chromatography-mass spectrometry (HPLC-MS) measurements, we have further uncovered the degradation pathway of ONP in the presence of LFCO under visible-light illumination. Ten types of intermediates were detected after degradation and their chemical structures were shown in Supplementary Fig. 41. To investigate the mechanism in depth, the optimized chemical structure and Fukui function (CFF) of ONP are calculated and displayed in Fig. 4G, Supplementary Figs. 42, 43, and Table 5, in which higher values of $f^-$, $f^+$, and $f^0$ are more vulnerable to be attacked by electrophilic species, nucleophilic species, and general radical, repectively[53]. It is worth noting that the N12 atom ($f^+$ = 0.1121) with a high Fukui index is the most reactive site and easy to be attacked by e$^-$. As shown in Fig. 4H, the rapid reduction of the -$NO_2$ to -$NH_2$ groups to form OAP (**P3**) requires electron transfer (6 e$^-$) coupled with hydrogenation (proton from PMS oxidation, Eq. (3)), which is consistent with previous studies and our experiments[1,54,55]. In the subsequent oxidization process, hydroxylation, deamination, and dehydrogenation reactions on the benzene ring were induced by •OH due to its high reactivity with strong electron-donating substituents (e.g., -OH and -$NH_2$) to generate 2-aminobenzene-1,4-diol (**P4**), hydroquinone (**P5**), and benzoquinone (**P6**), while $^1O_2$ tends to attack the phenolic group with rich electrons and break up P6 into small molecular acids (**P7-P11**)[56]. Finally, the generated byproducts are completely mineralized into $CO_2$ and $H_2O$, which can be confirmed by TOC results (Supplementary Fig. 37).

## Large-scale application and cost analysis

Inspired by the above encouraging removal performance of the FBR system, environmentally feasible and cost-effective for large-scale applications were then investigated. Predicting the residual structure and toxicity of degradation products is an important process to comparably evaluate their ecosystem risks. The toxicity estimation software tool (T.E.S.T.) was used to calculate the toxicity indicators of all products, which includes the 50% lethal concentration of the product to organisms (Oral rat LD50), bioaccumulation factor (BAF), development toxicity and mutagenicity (Fig. 5A and Supplementary

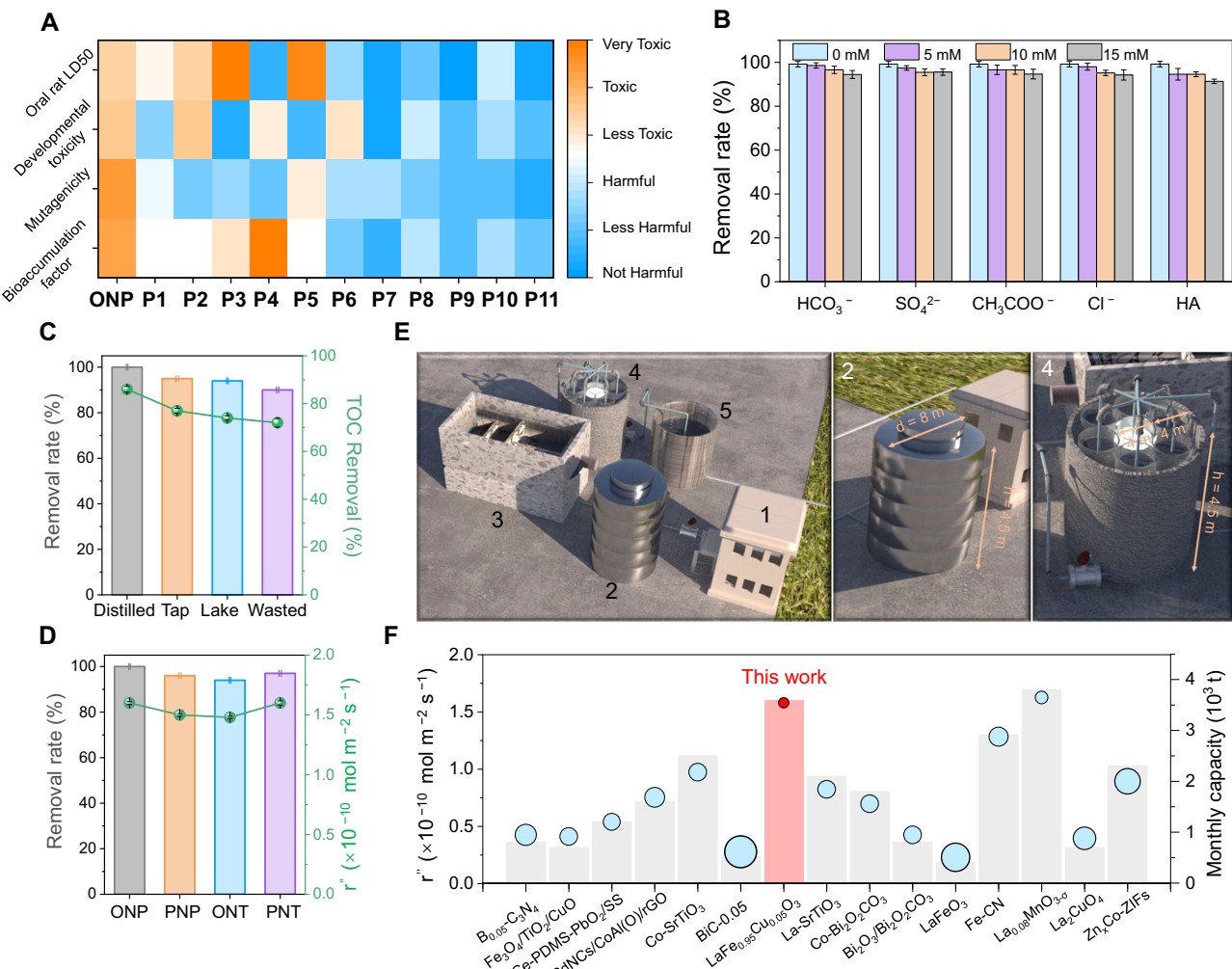

**Fig. 5 | Application in water treatment. A** Toxicity indicators of parent ONP and its degradation products. Effects of different inorganic anions, humic acids (**B**) and water sources (**C**) to the degradation efficiency. **D** Degradation of different nitro contaminants in LFCO@CFC FBR system. FBR conditions (unless indicated otherwise): λ > 420 nm radiation, Water flux 7500 mL h⁻¹, HRT 60 s, 298 K, initial pH 7.1, 30 ppm ONP, and 3 mM PMS (introduced with the ONP stock solution). **E** Diagram of the large-scale application plant including 5 main parts: (1) wastewater collection (2) PMS dosing tank, (3) physical filtration, (4) FBR reactor, (5) effluent tank (data from 1:50 for equal scale amplification). The dimensions of core parts 2 and 4 are marked in the enlarged image on the right. **F** Performance and cost comparison of reported catalysts. The area of the circle represents the cost, and its height represents efficiency (Supplementary Table 11, Supplementary references 36−47). The gray column height represents the monthly treatment capacity. Error bars are standard error values of three tests (*n* = 3).

Fig. 44)[57]. It clearly shows that OAP (**P1**) has relatively higher ecosystem risks than ONP, but all the oxidized products (**P6-P10**) have low toxicity levels. This proves the importance of the sequential reduction-oxidization pathway of ONP degradation. Then, we conducted a series of experiments to simulate the actual ONP degradation process in the FBR system. The wastewater matrices-dependent was investigated first, which exhibits a negative effect on the radical-based oxidation process[58]. Notably, a negligible effect was observed of inorganic anions and HA (Fig. 5B), demonstrating the synergetic effect of radical and non-radical for ONP degradation. In Fig. 5C, ONP removal efficiency remains unchanged even in tap water, lake water (Mati Lake in Nankai University), and wastewater (SHANXI COKING COAL GROUP CO., LTD.). Additionally, the FBR system also achieves an efficient removal rate for a wide variety of nitroaromatic pollutants including p-nitrophenol (PNP), o-nitrotoluene (ONT), and p-nitrotoluene (PNT) (Fig. 5D). These results prove the wide application of the ROC degradation mechanism of the FBR system. Then, the energy cost of ONP degradation in the FBR system was analyzed using the EE/O method, which is defined as the electrical energy to achieve one order of ONP removal (Supplementary Methods 1.5). Results show that the total cost of for the LFCO@CFC (0.23 KWHL⁻¹) is much lower than that of

LFO@CFC (1.43 KWHL⁻¹), further proving the its scale application is feasible. To further validate the application potential of the LFCO@CFC FBR system, a larger LFCO@CFC fixed bed reactor (diameter: 8 cm, high: 16 cm) was constructed for large-scale degradation of ONP. As depicted in Supplementary Fig. 45, The scale-up setup system can also realize high stability and durability in 7 days with the degradation efficiency of ONP remaining> 97 %. Then, 1:50 equal-scale amplification of small equipment was used to achieve real large-scale application with industrial significance, and the specific design parameters were shown in Fig. 5E and Supplementary Methods 1.6. As the core of the plant, 6 cylindrical FRBs are evenly distributed around the central light source (1326 W) for maximizing the utilization rate of light energy and reducing energy and investment costs. According to the cost calculation in Supplementary Methods 1.7, LFCO@CFC FBR costs about 13.72 CNY/ton for industrial wastewater treatment, which is much lower than the market price (30−60 CNY/ton). In addition, we prepared the catalysts that have been widely studied in photo-assisted PMS activation systems and tested their activities for comparison (Fig. 5F and Supplementary Table 11, Supplementary references. 36−47). The results clearly show that the LFCO@CFC FBR outperformed most of the recently reported photocatalysts of pollutant

removal performance and lower processing costs, showing strong environmental remediation capabilities.

## Discussions

In this article, we have presented strategies to address the two technological challenges for the safe degradation of NACs, including inefficient ring-opening oxidation and practical large-scale treatment. We firstly designed a LFCO@CFC FRB and applied it in the PMS/light system. The super-exchange effect induced by Cu doping facilitates the transfer of photoelectrons and induces the -NO$_2$ reduction process, which is critical for the unique ROC degradation of NACs. In addition, the synergetic production of radical and non-radical on Fe and Cu sites enables the efficient removal and mineralization of NACs without interference from water matrices. Ultimately, the LFCO/PMS/Vis system achieved the highest ONP degradation rate of 0.079 min$^{-1}$, which is 7.7 times higher than that of LFO (0.009 min$^{-1}$). Benefiting from the excellent degradation performance toward NACs and low cost, our work provides precious guidance for constructing ROC technology in continuous flow processes, which is highly promising for the removal of electron-withdrawing groups substituted aromatic contaminates and other organic pollutants.

## Methods

### Preparation of LaFe$_{1-x}$Cu$_x$O$_{3-\sigma}$@CFC FBR

Pretreatment of the carbon fiber cloth (CFC). The commercial CFC was purchased from CETECH CO., LTD. and cut into a 3 cm × 3 cm square, then they were washed under ultrasound with acetone and deionized (DI) water for 90 min to completely remove organic residues and other impurities on the surface of the CFC. To further improve the hydrophilicity of these substrates, they were immersed in a mixture with a 1:1 volume ratio of concentrated HNO$_3$ to concentrated H$_2$SO$_4$ for 48 h. After cleaning with a large amount of deionized water, the pretreated CFC substrate was dried overnight in a vacuum oven at 60 °C[4].

Preparation of powder LaFe$_{1-x}$Cu$_x$O$_{3-\sigma}$. According to the typical sol-gel method, the molar concentration ratio of citric acid: metal nitrate La(NO$_3$)$_2$·6H$_2$O: Cu(NO$_3$)$_3$·3H$_2$O: Fe(NO$_3$)$_3$·9H$_2$O = 1.05: 1: $x$: $1-x$ in LaFe$_{1-x}$Cu$_x$O$_{3-\sigma}$ (LFO-Cu$_x$, $x$ = 0, 0.01, 0.03, 0.05, 0.1). The catalysts were named LFO-Cu$_{0.01}$, LFO-Cu$_{0.03}$, LFO-Cu$_{0.05}$ (LFCO), and LFO-Cu$_{0.10}$ according to the doping concentration ($x$) of Cu. Taking LFCO as an example: the required amounts of nitrates (2.1650 g La(NO$_3$)$_2$·6H$_2$O, 1.943 g Fe(NO$_3$)$_3$·9H$_2$O, and 0.0604 g Cu(NO$_3$)$_3$·3H$_2$O) were added into 100 mL deionized water with stirring to prepare a mixed solution. After that, 2.2060 g citric acid powder were added to the mixed solution with stirring. Subsequently, the mixture was stirred continuously at 70 °C to evaporate water and make the obtained gel attach to the CFC and dry at 120 °C overnight. Finally, the LFCO powder was obtained by calcining in N$_2$ atmosphere at 800 °C for 3 h (heating rate of 5 °C·min$^{-1}$). After grinding and weighing, the yield of catalyst is about 1.5 g.

Preparation of LFCO@LFC. According to the typical sol-gel method, 2.1650 g La(NO$_3$)$_2$·6H$_2$O, 1.943 g Fe(NO$_3$)$_3$·9H$_2$O, and 0.0604 g Cu(NO$_3$)$_3$·3H$_2$O were added into 100 mL deionized water with stirring to prepare a mixed solution. After that, 2.2060 g citric acid powder and the prepared CFC were added to the mixed solution with stirring. Subsequently, the mixture was stirred continuously at 70 °C to evaporate water and make the obtained gel attach to the CFC and dry at 120 °C overnight. Finally, the LFCO@LFC with about 1.5 g catalyst was obtained by calcining in N$_2$ atmosphere at 800 °C for 3 h (heating rate of 5 °C·min$^{-1}$). By scale up or down the molar mass of citric acid and metal nitrate in the mixed solution, CFC with different loading amounts of LFCO were obtained. Unless otherwise stated, LFCO@LFC refers to 1.5 g loading amount of LFCO.

### FBR degradation experiments

Experiments were performed in flow-through (dead-end) mode. Under light irradiation at room temperature (298 K), the thoroughly mixed

ONP and PMS solution (500 mL) flowed through LFCO@CFC FRB by a peristaltic pump (30 W) with no feed recirculation. The key system elements are as described below: The as-prepared LFCO@CFC (9 cm$^2$) was inserted in a quartz glass tube to allow the transmission of light. A 350 W Xenon lamp (CEL-HX F300, Beijing, China) with a 420 nm and 780 nm cutoff filter was used as the light source and the light intensity is 560 mW cm$^{-2}$. The system was operated at different water fluxes and HRT by controlling the pump flow rate. Unless indicated, 7500 mL h$^{-1}$ flux, HRT 60 s, 298 K, initial pH 7.1, 3 mM PMS (introduced with the 30 ppm ONP stock solution) were chosen as optimal conditions for all experiments of LFCO@CFC FBR. All samples were collected and filtered through an organic membrane filter (0.22 μm) for UV-Vis, TOC, HPLC, and UPLC-MS analysis. At the same time, the obtained used LFCO@CFC by filtering were also further characterized.

HPLC data were used to calculate the ONP removal efficiency according to Eq. (1):

$$\text{Removal \%} = \frac{C_t}{C_O} \qquad (5)$$

where C$_t$ is the concentration of ONP at a certain reaction time (t) and C$_O$ refers to the initial concentration after adsorption equilibrium.

Ideally, the relationship between ln(C/C$_O$) and reaction time was matched well with the integral rate equation of the first-order reaction. Therefore, K$_{obs}$ can be expressed by the pseudo-first-order kinetic model as described below (Eq. (2)):

$$\ln\left(\frac{C_t}{C_O}\right) = -K_{obs}t \qquad (6)$$

Considering mass transfer, the actual degradation rate (r″, m$^{-3}$ s$^{-1}$) in a heterogeneous reaction can be calculated based on Eq. (3):

$$-r'' = -\frac{1}{A \times V} \times \frac{dN_{ONP}}{dt} = \frac{ONP_{reacted}}{A \times V \times t} = \frac{1}{A \times V} \times Q_f \times c_f \times r_f \qquad (7)$$

where A is the catalyst weight (g), V is the volume of carbon fiber felt, t is the reaction time (s), Q$_f$ (L s$^{-1}$) is the water flow rate, c$_f$ (g·L$^{-1}$) is the feed concentration of ONP, and r$_f$ is the removal efficiency of ONP.

### Beaker experiments

To facilitate the in-situ detection of reaction processes and intermediate products, we also used beaker experiments for ONP degradation. Unless indicated, 50 mL 30 ppm ONP solution including 0.3 mM PMS and 0.3 g·L$^{-1}$ powder catalyst under 560 mW cm$^{-2}$ light irradiation at 298 K initial pH 7.1 were chosen as standard conditions for all beaker experiments. Before the light irradiation and adding PMS, the solution was stirred in the dark for 30 min to reach adsorption-desorption equilibrium. During the reaction, 4 mL solution was extracted at intervals and immediately filtered through an organic membrane filter (0.22 μm) for further analysis. At the same time, the obtained solid catalysts after filtering were also further characterized.

### Calculation of contributions for radicals and non-radicals

Quenching experiments were performed to study the effects of ROS during ONP degradation with adding 0.05 mmol L$^{-1}$ scavengers before the degradation process. Methanol (MeOH, $k_{SO4\bullet^-/MeOH}$ = 3.2 × 10$^6$ M$^{-1}$ s$^{-1}$, $k_{\bullet OH/MeOH}$ = 9.7 × 10$^8$ M$^{-1}$ s$^{-1}$) can be used to quench both •OH and SO$_4$$^{•-}$, while tert-butyl alcohol (TBA) has much higher rate constant with •OH (3.8 × 10$^8$–7.6 × 10$^8$ L mol$^{-1}$ s$^{-1}$) than SO$_4$$^{•-}$ (4.0 × 10$^5$–9.1 × 10$^5$ L mol$^{-1}$ s$^{-1}$). Carotene, superoxide dismutase (SOD), and K$_2$Cr$_2$O$_7$ are easier react with $^1$O$_2$, •O$_2$$^-$, and photogenerated electrons (e$^-$), respectively.

Then we quantitatively calculated the contributions of e$^-$, $^1$O$_2$, and •OH according to Eq. (4) ~ (6). The reaction rate constants after the addition of K$_2$Cr$_2$O$_7$, carotene, and MeOH were noted as k$_1$, k$_2$, and k$_3$,

respectively, and the initial reaction rate constant without quencher was $k_O$.

$$\lambda(\bullet OH)\left[\frac{k_0 - k_3}{k_0}\right] \times 100\% \qquad (8)$$

$$\lambda(^1O_2) = \left[\frac{k_0 - k_2}{k_0}\right] \times 100\% \qquad (9)$$

$$\lambda(e^-) = \left[\frac{k_0 - k_1}{k_0}\right] \times 100\% \qquad (10)$$

Where $\lambda(\bullet OH)$, $\lambda(^1O_2)$, $\lambda(e^-)$ were the contribution of $\bullet OH$, $^1O_2$, $e^-$ in the degradation process of ONP, respectively.

### Reporting summary

Further information on research design is available in the Nature Portfolio Reporting Summary linked to this article.

## Data availability

The data supporting the findings of the study are included in the main text and supplementary information files. Raw data can be obtained from the corresponding author upon request.

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

## Acknowledgements

The authors gratefully acknowledge the financially support by the Natural Science Foundation of China as general projects (grant Nos. 22225604 and 22076082 to S.Z., grant Nos. 42277059 to P.W.), the Frontiers Science Center for New Organic Matter (grant No. 63181206 to S.Z.), and Haihe Laboratory of Sustainable Chemical Transformations.

## Author contributions

Y. Mao and B. Yu conducted catalyst syntheses, and performance tests and prepared the manuscript. P. Wang and S. Yue participated in the experiment. S. Zhan directed and supervised the project. All authors discussed the results and commented on the manuscript.

## Competing interests

The authors declare no competing interests.
