## [Peer Review File · Nature Communications]

Efficient Reduction-Oxidation Coupling Degradation of Nitroaromatic Compounds in Continuous Flow ProcessesEditorial Note: Parts of this Peer Review File have been redacted as indicated to remove third-party material where no permission to publish could be obtained.

REVIEWER COMMENTS

Reviewer #1 (Remarks to the Author):

This manuscript reported the reduction and oxidation coupling degradation for the nitroaromatic compounds in the continuous flow process. The composite used in the process was prepared, characterized and the performance was further evaluated. It is a very interesting works. While, in my opinion, the following comments needed to be concerned when revising this manuscript:

1. The effective mineralization of organic contaminants is mostly expected in water treatment. From the SI Figure 36, it seems the TOC removal rate is only 17%, which is far lower than the degradation rate indicated from UV-vis. How possible use the UV-vis data to support the effective removal of nitroaromatic compounds. The intermediate and final compound during the reduction-oxidation process should be illustrated. How about the toxicity of these intermediate and final small organic molecules?
2. For the durability of the composite, 12-hour continuous experiment is not enough. As a result, the cost evaluation might be not reasonable. How is the weight of composite used in the experiment and how is the volume treated in the 12 hours? As indicated for the SI Table 10, the Cu ion measured is about 0.05 ppm during the process, how is the total release of the Cu quantity in the process?
3. For the large-scale application, the effect of wastewater matrices is required but not fully support it. The real wastewater containing various nitroaromatic pollutants from factory should be used for supporting the feasibility of this technology.

Reviewer #2 (Remarks to the Author):

Nitroaromatic compounds (NACs) with electron-withdrawing nitro groups (-NO₂) contribute significantly to water pollution. In this work, Zhan and co-workers developed a reduction-oxidation coupling (ROC) degradation process in LaFe_{0.95}Cu_{0.05}O₃@carbon fiber cloth (LFCO@CFC)/PMS/Vis system, which has high scalability, material durability, and catalytic activity. This manuscript proposes a new degradation mechanism for NACs and makes relevant derivations for actual water treatment. At the onset, this topic is of importance and sounds necessary in the field. The degradation efficiency is high and the reaction pathway has been well studied. The catalyst structure-activity correlation has also been convincingly described. I recommend its acceptance for publication in Nat. Commun. after proper revision.

1. There are many catalysts suitable and efficient for PMS activation, such as carbon materials and metal oxides. By changing its composition, morphology or microstructure, the catalytic performance is continuously enhanced. Please further explain why choose perovskite materials in this manuscript and adopt B-site doping for modification.

2. During ONP degradation, the authors show that the photoinduced electrons are generated and used for ONP reduction, which is the core of the reaction. What is the role of the photogenerated holes in the reaction process?
3. In terms of large-scale applications, how are the main parameters selected and set? The author's explanation of the condition setting of the simulation is not sufficient, please provide sufficient evidence.
4. How to ensure that the material can achieve a 30-day operation cycle in large-scale applications?
5. At present, the large-scale application of advanced devices in AOPs is imminent, and many researches have made efforts for it. Does the reduction-oxidation coupling mechanism and large-scale treatment equipment have the same removal effect on other refractory aromatic compounds and hydrocarbon chain compounds?
6. Degradation and removal appear many times in the article, the definitions of the two are different, please pay attention to distinguish and confirm.
7. In general, the manuscript is very well written. However, I did not see some confused mistakes in the paper. Material subscript formats for Line 83 and Line 84 need to be corrected.
8. All nitro groups (-NO₂) must be written in the same format.

Reviewer #3 (Remarks to the Author):

The manuscript presents a systematic study on the synthesis of a photocatalytic material and its application. Authors fail to provide the key pieces of information and background to support their hypothesis. The results presented do not demonstrate the dual role as the lack of blank experiments difficults getting to any sound conclusion. The most worrisome is that several claims are done all thorough the text in disagreement with current literature. Similarly, several subjective interpretations result in conclusions not proven experimentally in the manuscript. Unfortunately the manuscript cannot be recommended for publication.

Reviewer #4 (Remarks to the Author):

The authors describe a new system for the reduction-oxidation coupling of nitroaromatic pollutants. They present the design and synthesis of a photocatalyst to accomplish such a goal, including an extensive characterization of the synthesized materials and their application on a large-scale system under flow. This piece of work is well conducted, but it also shows several points that should be addressed before reaching quality for publication in this journal:

Major points:

- 1) In the introduction section, the authors discuss the inefficiency of reported approaches to deal

with the removal of nitroaromatic compounds. Then, in lines 62-63: "Therefore, it is urgent to develop a controlled coupling of reduction and oxidation processes in one system with high resistance to environmental interference." The authors should clarify the novelty of this approach or enlarge the discussion on the advantages of it in previous articles.

2) In the section: Overview of fixed reaction bed (FRB) reactor, the authors do not discuss the ratio LFCO vs CFC. It is hard to imagine that only one trial has been done, but still, in that case, a discussion on the ratio should be included.

3) Still, in this section, the authors refer to control experiments (Figures S3-S4) in which the amount of PMS and LFCO differ from those reported in section FBR system operation; see below.

4) In the first two sections of the results, the authors describe the synthesis and characterization of a bunch of materials with different ratios Cu/Fe (LFCO). In the discussion, they should include references to the same materials or claim their novelty.

5) Section: FBR system operation. It is hard to imagine the FRB system, even with Figure 5, since the photograph has no scale. Is the explanation in the supplementary text 4 the one that applies? It could be worth better clarifying in the text when the authors refer to the results of a batch experiment or the FBR system.

6) Lines 163-164: "In Figure 3a, when the water flux decreased from 15000 mL/h to 1500 mL/h, the pollutant removal rate decreased from 99.123% to 69.665%". It seems to be the opposite of the graph. Similar to residence time, please verify.

7) Line 177 (The dosage of catalyst and PMS, and light intensity). Please check the amount of catalyst commented on in the text and the one shown in Figure 3c. Moreover, catalyst refers to LFCO or LFCO@CFC?

8) With respect to the PMS concentration, the authors comment on the optimum 3 mM=92 mg/L; however, control experiments were performed with 0.3 mg/L of PMS (see Figure S3). How do we compare the two experiments? Figure S15 corresponds to LFCO or LFCO@CFC? Figure S16: it looks pretty similar to S3 and S4. Moreover, in Figure S3: is LFCO soluble, or is it dispersed? Is the pH adjusted, or is the pH resulting directly from the mixture?

9) Lines 187-189: the authors should comment more deeply on the results observed under Nir conditions. It looks like they only presented them as a curiosity. Could it be more economically favored to use Nir light?

10) Is the legend in Figure S33 correct? Figure S34 a and b: why the ONP spectrum at time 0 is not identical in both cases? Moreover, check the evolution of the peaks in Figure S34 a because it looks to this reviewer that the three bands decrease to the same extent. Also, check Figure 4f.

11) In the experiments reported in Figure S37, the experimental conditions, including concentrations of scavengers, are needed to extract conclusions safely.

12) Lines 326-328 and Tables S11 and S12: the references where the numbers are extracted from should be included.

13) How are the values extracted from graph 4b compared to those from graph S18?

14) This reviewer expected an overall mechanism of the reaction at the end of the discussion. The scheme shown in Fig 4h starts with the expected reduction of ONP, which needs two H⁺. What about the pH of the solution is 7.1 always? what about the evolution of the pH with time? We can assume that the reduction starts in the LFCO, and then the ROS from PMS play the oxidation role. However, is it reasonable to eliminate an Ar-NH₂ to give an Ar-H by the action of 1O₂? Technically speaking, this process is a reduction.

15) In the experimental section, important details are missing: 1) source of CFC; 2) in the synthesis of LFCO@CFC, the ratio LFCO vs CFC is crucial for readers to reproduce the synthesis of the catalysts; 3) in the section degradation experiments the authors describe a system in which the amount of ONP solution is 50 mL; however, they do not mention any detail on the degradation in the flow system. Please add these details.

Minor points:

-Line 68: ONP definition should appear here instead of in line 71

-Line 347: is rhodamine used along the MS?

-Figure S1: typo: "Illustration"

-The temperature of calcination should be coincident with the one described in the experimental section (line 357: 800 °C).

-Figures S3 and S4: details on the ONP concentration are missing.

-The legend of Figure S4 does not correspond to graphs.

-References S23 and S35 are the same

Response to the comments of the reviewers

Response to Reviewer 1	1
Response to Reviewer 2	18
Response to Reviewer 3	33
Response to Reviewer 4	49

General comments from Reviewer 1:

This manuscript reported the reduction and oxidation coupling degradation for the nitroaromatic compounds in the continuous flow process. The composite used in the process was prepared, characterized and the performance was further evaluated. It is a very interesting works. While, in my opinion, the following comments needed to be concerned when revising this manuscript:

Response: We thank the reviewer for the positive evaluation of our manuscript. Here, we provide a point-to-point response to your specific comments below.

Comment 1. The effective mineralization of organic contaminants is mostly expected in water treatment. From the SI Figure 36, it seems the TOC removal rate is only 17%, which is far lower than the degradation rate indicated from UV-vis. How possible use the UV-vis data to support the effective removal of nitroaromatic compounds. The intermediate and final compound during the reduction-oxidation process should be illustrated. How about the toxicity of these intermediate and final small organic molecules?

Response: We thank the reviewer for the thoughtful comment.

1. About UV-Vis and TOC

Both TOC and UV-Vis absorbance are important parameters for evaluating the content and quality of organic matter in water. However, there are differences between them. By correctly understanding the relationship between the two, we can avoid misinterpreting water quality monitoring data. The content of organic matter in water is expressed by the amount of carbon, the main element in organic matter, which is called total organic carbon (TOC). The determination of TOC is carried out at a high temperature of 950 °C. So that the organic matter in the water sample is vaporized and burned to generate CO₂. The total amount of organic carbon can be known by measuring the amount of CO₂ generated by the infrared analyzer, and then accurately estimating the concentration of organic matter. Therefore, TOC has become a widely adopted method for controlling and regulating the concentration

of organic matter. UV-vis spectrophotometry is a continuous spectrum of electromagnetic waves in the UV-Visible region (usually 200-800 nm) as a light source to irradiate the sample and study the relative intensity of light absorption of material molecules. The molecule or group in the substance absorbs the incident UV-visible energy, and the energy transition between electrons produces a characteristic UV-visible spectrum, which can be used to determine the structure of the compound and characterize the properties of the compound. Its application mainly includes: (1) By comparing the spectral characteristics of the sample, such as the number, position, intensity (molar absorption coefficient), and shape (maximum, minimum, and inflection point) of the absorption peaks, with pure compounds or standard UV spectra reported in the literature, the structure and purity of the compound can be inferred. (2) Comparing the relevant properties of the sample based on the strength of the absorption peak. A compound may have multiple absorption peaks, and the relative strength changes of these peaks can infer the relevant properties of the compound. Before comparison, we should understand the meaning represented by each absorption peak of the compound based on the literature. In summary, TOC is a concise measurement of organic matter concentration, while UV absorbance can provide a supplementary basis for characterizing samples. UV absorbance must be used together with TOC data for sample comparison.

In our experiment, we used *an in-situ* UV-vis absorption spectrogram to mainly analyze the functional group changes (-NO₂ or -NH₂) in ONP during the reaction process by the changes in the position and intensity of the absorption peak. Based on this experiment, we found an interesting ordered reduction-oxidation process of ONP. On the other hand, the effective removal rate of nitroaromatic compounds was evidenced by HPLC by standard curve method. It has the advantages of a wide analysis range, strong separation ability, reliable qualitative analysis results, low detection limit, and fast analysis time, and is currently the most powerful tool for environmental monitoring. The difference between TOC and HPLC-MS data in the LFO@CFC system is mainly because the parent pollutant is only degraded into small molecule compounds but not mineralized. Therefore, we use Cu doping to improve the degradation and mineralization performance of LFO, which is also one of the innovative points of the article.

To clarify the above issues, we have made some changes in the revised supplementary information:

The structure changes of ONP were tested by *in-situ* UV-Visible (UV-Vis) spectrophotometry (UV-3200, Mapada, China). The concentration of ONP was analyzed by high-performance liquid

chromatography (HPLC), (SPD-20A, Shimadzu Corporation, Japan) using a 60:40 methanol-water mixture as solvent, and 278 nm wavelength. Degradation intermediates were analyzed by ultra-performance liquid chromatography-mass spectrometry (UPLC-MS, Orbitrap Fusion, Thermo, USA). The mineralization rate was detected by total organic carbon (TOC, Analytikjena multi N/C analyzer).

2. About intermediates and final compounds

In our manuscript, the intermediates were detected by UPLC-MS. As a high throughput analysis technology, UPLC-MS is one of the effective means of product analysis, which generally uses mass spectrometry as a detector for analyzing the mass charge ratio of ionized samples to achieve qualitative and quantitative analysis of tested compounds. When the tested sample enters the mass spectrometer, in the mass spectrometer ion source, the compound is bombarded by ions, ionized into molecular ions and fragment ions, these ions in the mass analyzer, due to the mass charge ratio is different, and the movement trajectory is different, through the electron multiplier tube detection amplified signal into the display, showing a complete mass spectrum. The horizontal coordinate of the general mass spectrum is the mass charge ratio, and the vertical coordinate is the strength of the ion. Compared with other detection technologies, UPLC-MS has the advantages of large load, good applicability, and high separation can provide higher selectivity, sensitivity, and more abundant structural information. Additionally, the final compounds were CO₂ and H₂O, which were proved by a high TOC removal rate. Therefore, all intermediates and final compounds we detected are reliable.

In addition, as suggested by the reviewers, we added OAP concentration tests to further demonstrate the reduction-oxidation coupling process and modified the figures. We have made the following changes in the revised manuscript and supplementary information:

(1) Interestingly, the absorption band at 210 nm slightly shifted to 230 nm with increased intensity within 10 minutes only in LFCO/PMS/Vis system, suggesting the generation of OAP at the initial stage (Fig. R1e and Fig. R2a-b). Meanwhile, the absorption band at 278 and 350 nm gradually disappeared, inferring that the aromatic ring was opened and mineralized as the reaction time increased. The variation of ONP and OAP concentration was further detected by HPLC (Fig. R1f and Supplementary Fig. R2c-d). Upon the decline of the ONP concentration, the formation of OAP was observed immediately (within 5 min). Notably, negligible concentration of both ONP and OAP was achieved upon completion of the reaction, which indicates complete oxidative degradation.

Fig. R1 Reduction-oxidation coupling (ROC) mechanism and degradation pathway of ONP. A The band gap diagram of LFO and LFCO. **B** The XPS valence band spectra of LFO and LFCO. **C** Proposed mechanism for the PMS reaction on the surface of LFCO. **D** EPR spectra of DMPO- $\cdot\text{OH}$ /DMPO- $\text{SO}_4^{\cdot-}$ and TEMP- $^1\text{O}_2$ in PMS/Vis, LFO/PMS/Vis, and LFCO /PMS/Vis system, respectively. **E** In-situ UV-Vis absorption spectra of ONP degradation in the LFCO/PMS/Vis system. **F** HPLC spectra and corresponding concentration changes of ONP and OAP during degradation in the LFCO/PMS/Vis system. **G** Charge density difference of condensed Fukui index (f^+) of ONP. **H** The degradation pathway of ONP in the LFCO/PMS/Vis system system. Conditions: $\lambda > 420$ nm, 50 mL 30 ppm ONP solution, 0.3 mM PMS, 0.3 $\text{g}\cdot\text{L}^{-1}$ powder catalyst, 298 K, initial pH 7.1.

(2) Fig. R2a and c show that in the PMS system, the degradation rate of ONP is low and OAP is not generated. Fig. R2b and d show that under the photocatalytic condition, though the peak of ONP decreases slowly, the characteristic peaks of OAP increase first and then decrease, indicating that part of ONP is reduced to OAP during degradation. This also shows the dominance of photoreduction in the process of reducing ONP to OAP and reactive oxygen species are responsible for the subsequent ring-opening process of OAP.

Fig. R2. *In-situ* UV-vis absorption spectra of ONP degradation in the **a** LFCO/PMS and **b** LFO/Vis system. HPLC spectra of generated OAP in the **c** LFCO/PMS and **d** LFO/Vis system.

Fig. R1 corresponds to Figure 4; Fig. R2 corresponds to Supplementary Figure 36.

3. About toxicity

According to the previous description, we detected 11 main degradation products using mass spectrometry and high-performance liquid chromatography. Predicting the residual structure and toxicity of pollutants in ecosystems is an important process. Using biological activity experiments and other methods to test the toxicity of products requires a lot of manpower and resources. However, using model calculations and analysis to link chemical activity with molecular structure and composition can more economically and efficiently predict degradation products and their toxicity. Research has shown that the toxicity estimation software tool (T.E.S.T.) is considered one of the best predictive programs for estimating aquatic toxicity. As a quantitative structure-activity relationship modeling software recommended by the US Environmental Protection Agency and the European Chemicals Agency, TEST has a high predictive coverage (covering a wide range of heterocyclic and macromolecular substances).

The analysis steps are as follows in our manuscript: Firstly, the degradation path was derived and analyzed according to the m/z data obtained from LC-MS, and the structural formula of degradation intermediate products was obtained. Then, the structural formula of each product was drawn using ChemDraw software. Finally, the T.E.S.T. software was used to calculate the toxicity parameters, which include the 50% lethal dose or 50% lethal concentration of the product to organisms (Oral rat LD50), bioaccumulation factor (BAF), developmental toxicity, and mutagenicity[1]. In Figure R3, it clearly shows that OAP (P1) has relatively higher ecosystem risks than ONP, but all the oxidized products (P6-P10) have low toxicity levels. This proves the importance of the sequential reduction-oxidization pathway of ONP degradation.

Fig. R3 a Oral rat LD50, b Developmental, c Toxicity mutagenicity and d Bioaccumulation factor of ONP and its transformation products.

To clarify the above issues, we added the following modifications in the revised manuscript:

Fig. R4. Thermal map of toxicity analysis.

We hope these explanations can help you understand the relevant questions, and thank you again for your professional comments.

Fig. R3 corresponds to Supplementary Figure 44; Fig. R4 corresponds to Figure 5a.

Reference:

1. Sun, Z. *et al.* Persistent Free Radicals from Low-Molecular-Weight Organic Compounds Enhance Cross-Coupling Reactions and Toxicity of Anthracene on Amorphous Silica Surfaces under Light. *Environ. Sci. Technol.* **55**, 3716-3726, (2021).

Comment 2. For the durability of the composite, 12-hour continuous experiment is not enough. As a result, the cost evaluation might be not reasonable. How is the weight of composite used in the experiment and how is the volume treated in the 12 hours? As indicated for the SI Table 10, the Cu ion measured is about 0.05 ppm during the process, how is the total release of the Cu quantity in the process?

Response: Thanks for raising the important comments.

1. For the “durability of the composite”

According to the suggestions of the reviewer, we have prolonged the continuous experiment to 7 days, also we have added more characterizations and discussions for used LFCO to prove the durability.

(1) Besides, the degradation efficiency of ONP remained at >99 % in 7-day continuous experiments with low metal leakage (Fig. R5e and Table R1). The similar XRD, SEM, TEM, and XPS results of fresh and LFCO (Fig. R6-7) together proved the stability and durability of the catalyst.

Fig. R5 Long-term operation performance of the LFCO@CFC FRB system (Illustration is the photograph of the used LFCO@CFC).

(2) The intensity/half peak width of XRD peaks is not only dependent on the particle size but also the content of the crystal phase. After a 7-day catalytic reaction, we can see that the used LFCO catalyst shows the same peaks with fresh LFCO (Fig. R6a), indicating that no phase transition during the reaction. As shown in SEM results (Fig. R6b), the catalyst is still tightly bound to CFC, indicating the active components were not lost with time. Besides, TEM results of used LFCO catalysts show that they maintained the particle structure and size with a clean surface and uniform distribution of elements,

which is well agreed with XRD data. These results together proved the structure stability of LFCO catalysts.

Fig. R6 a XRD, b SEM, c TEM images and corresponding EDS spectra of used LFCO.

(3) XPS analysis of the used and fresh LFCO catalysts offered valuable information about the valence state and the surface composition. As displayed in Fig. R7 and Table R1, all elements remain in clear outline and unchanged ratio, demonstrating the excellent stability of LFCO. As shown in Fig. R7b, although Fe^{3+} ions are mainly in the form of structure for both catalysts, the binding energy of Fe 2p_{1/2} level in used LFCO is lower than that of fresh LFCO, while Cu^{2+} has the opposite trend. Charge transfer from partially reduced neighboring Cu atoms would increase the electron density on Fe causing the shift of binding energy. By following the redox cycle between Fe and Cu, the superior catalytic activity and outstanding stability of the LFCO catalyst perpetuated even after 7-days.

Fig. R7 a The La 3d, b Fe 2p, c Cu 2p, and d O 2p XPS spectra of fresh and used LFCO.

Table R2. Elements ratio of catalyst detected by XPS.

Catalyst	La (atomic %)	Cu (atomic %)	Fe (atomic %)	O (atomic %)
LFO	13.30	0	22.91	63.79
LFCO	14.76	4.61	17.36	63.27
LFCO-used	14.47	4.59	17.41	63.53

2. About cost

The energy cost is calculated by the EE/O method, which is defined as the electrical energy to achieve one order of ONP removal, and the units are KWHL^{-1} , therefore, it is not affected by the running time.

3. About weight

The weight of the composite used in the experiment was 1.5 g LFCO on the CFC. The ONP solution was placed in a 500 mL volumetric flask for a constant supply, therefore, the exact volume is not required. We have added the preparation details in the revised manuscript:

(1) Firstly, by changing the amount of chemicals, reaction beds loaded with different amounts of catalysts were prepared. Fig. R5c (left) shows that when the weight of the LFCO increased to 1.5 g, the highest r'' was observed, this may be due to the increase of active sites to activate PMS. However, the r'' cannot be significantly improved by increasing the weight of the LFCO.

(2) Preparation of different weight ratios of LFCO vs CFC. By changing the total amount of chemicals (including citric acid and metal nitrate) in the mixed solution, CFC with different loading amounts of LFCO were obtained. The weight of LFCO is determined by the difference of CFC and LFCO@LFC. Unless indicated, LFCO@LFC in the manuscript refers to the sample with 1.5 g LFCO.

4. About Cu quantity

The Cu ion measured was the total release of the Cu quantity in the process. we have clarified in the revised version:

Table R1. Total metal ions detected by ICP-MS during reaction in 7 days.

Time (day)	Fe (mg/L)	Cu (mg/L)
0	0.011	0.005
1	0.012	0.004
2	0.012	0.006
3	0.012	0.006
4	0.012	0.006
5	0.014	0.006
6	0.014	0.006
7	0.014	0.006

We hope these explanations can help you understand the relevant questions, and thank you again for your professional comments.

Fig. R5 corresponds to Figure 3; Fig. R6 corresponds to Supplementary Figure 17;

Fig. R7 corresponds to Supplementary Figure 18; Table R1 corresponds to Supplementary Table 10;

Table R2 corresponds to Supplementary Table 4.

Comment 3. For the large-scale application, the effect of wastewater matrices is required but not fully support it. The real wastewater containing various nitroaromatic pollutants from factory should be used for supporting the feasibility of this technology.

Response: We sincerely appreciate you for your time in reviewing the manuscript and giving your valuable comments. We are very sorry that we did not indicate the source of the water sample used in Figure 5c. In fact, the lake water and wastewater were collected from Mati Lake at Nankai University and SHANXI COKING COAL GROUP CO. LTD., respectively. Furthermore, we analyzed the samples after degradation for 10 and 30 minutes using gas chromatography-mass spectrometry (GC-MS) to persuade reviewers.

GC-MS makes comprehensive use of the separation capability of GC, qualitative analysis of MS, and qualitative analysis of unknown components by spectral library retrieval, and retains the quantitative function of GC for known substances. It is often used for qualitative and quantitative analysis of trace or trace volatile and semi-volatile organic compounds. Matrix interference can be eliminated in the analysis of complex mechanism samples and the detection sensitivity can be improved. When the mixed samples of many components enter the column, they are separated from each other in the column due to the difference in the partition coefficient or adsorption coefficient between the mobile phase and the fixed phase. The gas molecules separated by gas chromatography are bombarded by ion sources and electrolytically cracked into ions. Under the combined action of electric and magnetic fields, they are separated according to the size of m/z and reach the detector for detection, recording, and sorting, and the mass spectrogram is obtained to achieve qualitative and quantitative analysis of the samples. The main qualitative method of GC-MS is database retrieval. The mass spectrum of any component can be obtained from the total ion chromatogram, which can be retrieved from the database by computer. The search results give several of the most likely compounds. Including the name of the compound, molecular formula, molecular weight, base peak, and matching degree.

As shown in Fig. R8 and Table R2, the stock wastewater solution (0 min) contained a variety of benzene species including ONP. After ten minutes of degradation, we found that the intensity of the ONP peak decreased and a characteristic peak of OAP appeared (Table R3), while after 30 minutes, both the ONP and OAP characteristic peaks disappeared (Table R4). In addition, the benzene ring structure has disappeared, and many new short-chain small molecule compounds have emerged. This

strongly demonstrates the practical applicability of the reduction-oxidation coupling mechanism.

Fig. R8 GC-MS spectra of industrial wastewater during degradation at 0min, 10 min, and 30 min.

Table R2. Compounds detected in wastewater by GC-MS at 0 min.

Peak	Retention time	Compound	Area%
1	3.028	Acetic acid	20.72
2	3.629	Dimethyl sulfone	0.5
3	3.806	Methanesulfonic anhydride	2.25
4	4.167	Trimethylsilyl ethaneperoxoate	56.32
5	4.396	Silanediol	6.94
6	10.926	Cyclotetrasiloxane,	2.72
7	11.916	4-Hydroxypiperidine	1.7
8	12.29	2-Azabicyclo[3.2.1]octan-3-one	0.3
9	12.619	Cyclopentanecarboxylic acid	0.42

Table R2 (continued). Compounds detected in wastewater by GC-MS at 0 min.

10	12.923	3-Hydroxybenzoic acid	0.24
11	13.252	5H-1-Pyridine	0.91
12	13.608	O-nitrophenol	1.38
13	13.859	2-(2-Hydroxycyclohexyloxy)pyridine-N-oxide	0.23
14	14.588	3-Amino-Phenol	0.76
15	15.882	p-Isopropenylphenol	0.05
16	16.021	2-Allylphenol	0.12
17	16.095	2,6-Dihydroxybenzoic acid	0.27
18	16.184	Cyclohexasiloxane	0.96
19	16.305	Benzocycloheptatriene	0.1
20	16.419	2-Naphthalenol	1.04
21	16.788	Cinnoline	0.15
22	16.921	2,6-Dihydroxybenzoic acid	0.11
23	17.291	Cyclopentasiloxane, decamethyl-	0.19
24	17.794	7-Methyl-1-naphthol	0.05
25	19.15	2,6-Dihydroxybenzoic acid	0.29
26	19.377	3-Methylsalicylic acid	0.37
27	19.456	1-Naphthalenecarbonitrile	0.12
28	19.663	1-Naphthalenecarbonitrile	0.08
29	36.247	Palmitoleamide	1.11

Table R3. Compounds detected in wastewater by GC-MS at 10 min.

Peak	Ret. time	Compound	Area%
1	3.038	Ethane	24.33
2	3.605	Methanesulfonic acid	0.31
3	3.681	Methanesulfonic anhydride	1.71
4	3.785	Cholest-8-ene-3,6-diol	2.47
5	3.845	Silanediol	3.65
6	3.904	Trimethylsilyl ethaneperoxoate	7.74
7	3.951	Silanediol	12.82
8	4.008	Silanediol	7.98
9	4.091	Trimethylsilyl ethaneperoxoate	28.92
10	4.313	1-Methylbutylmandelate	0.11
11	10.85	Cyclotetrasiloxane, octamethyl-	0.96
12	10.93	Cyclotetrasiloxane, octamethyl-	0.64
13	12.767	5H-1-Pyridine	0.64
14	13.249	5H-1-Pyridine	1.38
15	13.604	Cyclopentasiloxane	1.18
16	13.729	O-nitrophenol	0.13
17	14.594	Ortho-aminophenol	0.37
18	14.635	Cyclotetrasiloxane, octamethyl-	0.15
19	16.021	O-nitrosophenol	0.21

Table R3 (continued). Compounds detected in wastewater by GC-MS at 10 min.

Peak	Ret. time	Compound	Area%
20	16.093	2,6-Dihydroxybenzoic acid	0.32
21	16.183	Cyclohexasiloxane	0.94
22	16.419	2-Naphthalenol	1.23
23	16.789	Cinnoline	0.26
24	19.377	Cyclohexadione diene	0.33
25	23.319	5H-Indeno[1,2-b]pyridine	0.25
26	28.354	Butenedioic acid	0.24
27	33.104	Muconic acid	0.13
28	36.248	Palmitoleamide	0.59

Table R4. Compounds detected in wastewater by GC-MS at 30 min.

Peak	Ret. time	Compound	Area%
1	3.029	Ethane	24.8
2	3.515	tert-Butyldimethylsilanol	1.25
3	3.568	2,4,5,6,8-Pentathianonane	1.08
4	3.668	Silanediol	2.71
5	3.716	Silanediol	2.93
6	3.768	Silanediol	9.46
7	3.879	Trimethylsilyl ethaneperoxoate	9.34
8	3.985	Silanediol	16.36

Table R4 (continued). Compounds detected in wastewater by GC-MS at 30 min.

9	4.04	1-Methylbutylmandelate	10.14
10	4.08	Trimethylsilyl ethaneperoxoate	12.8
11	10.836	Cyclotetrasiloxane	2.59
12	12.753	5H-1-Pyridine	0.82
13	13.235	5H-1-Pyridine	1.25
14	13.603	Ortho-aminophenol	0.17
15	14.326	1H-Indole	0.17
16	14.588	Phenol	0.49
17	16.182	Cyclohexasiloxane	0.69
18	16.418	2-Naphthalenol	0.65
19	16.786	Cyclohexadione diene	0.05
20	19.373	Hexasiloxane	0.27
21	19.451	1-Naphthalenecarbonitrile	0.11
22	22.548	5H-Indeno[1,2-b]pyridine	0.12
23	23.029	2,6-Dihydroxybenzoic acid	0.14
24	23.318	5H-Indeno[1,2-b]pyridine	0.25
25	28.347	Butenedioic acid	0.07
26	36.244	Muconic acid	0.02

We also found that after the coupling mechanism of reoxidation and reduction, the contents of long -NH₂ chain alkanes (Palmitoleamide), aromatic hydrocarbons alcohols and unsaturated aromatic hydrocarbons (1-Naphthalenecarbonitrile, etc.) at 0 minutes were reduced, while the contents of sulfur alkanes and long chain alkanes containing Si were also continuously decreased, indicating that

the degradation system can not only effectively degrade nitrobenzene organics, but also mineralize and remove other common pollutants in coking wastewater, such as long-chain alkane, aromatic hydrocarbon and sulfur-based compound. The degradation system has strong degradation ability and stability, and a high application prospect in actual wastewater treatment.

In the reaction process, the content and types of aminobenzene organic compounds increased at 10 minutes, indicating that photogenerated electrons can not only reduce $-NO_2$ of o-nitrophenol, but also reduce other nitrobenzene compounds, which verified the experimental results of this work. The content of amino alkanes and ethane also increased, which may be due to the changes in the molecular structure of organic matter caused by the reaction, 1O_2 and $\bullet OH$ can attack many kinds of pollutants in coking wastewater, so that the gradual cracking of aromatic hydrocarbons and long chain alkanes into short chain small molecule structures, converting them into small molecule chemical substances that can be recycled, and then related separation and recovery can be carried out.

We hope these explanations can help you understand the relevant questions, and thank you again for your professional comments.

General comments from Reviewer 2:

Nitroaromatic compounds (NACs) with electron-withdrawing nitro groups (-NO₂) contribute significantly to water pollution. In this work, Zhan and co-workers developed a reduction-oxidation coupling (ROC) degradation process in LaFe_{0.95}Cu_{0.05}O₃@carbon fiber cloth (LFCO@CFC)/PMS/Vis system, which has high scalability, material durability, and catalytic activity. This manuscript proposes a new degradation mechanism for NACs and makes relevant derivations for actual water treatment. At the onset, this topic is of importance and sounds necessary in the field. The degradation efficiency is high and the reaction pathway has been well studied. The catalyst structure-activity correlation has also been convincingly described. I recommend its acceptance for publication in Nat. Commun. after proper revision.

Response: We appreciate your positive comments and revisions as suggested. Here, we provide a point-to-point response to your specific comments below.

Comment 1. There are many catalysts suitable and efficient for PMS activation, such as carbon materials and metal oxides. By changing its composition, morphology or microstructure, the catalytic performance is continuously enhanced. Please further explain why choose perovskite materials in this manuscript and adopt B-site doping for modification.

Response: Thanks for the Reviewer's suggestion.

As a new type of inorganic nonmetallic material, perovskite oxides with unique physical and chemical properties have garnered significant attention in numerous applications, such as electronics, energy conversion and storage, and catalysts[1]. The ideal single-crystal perovskite oxides have a unit formula ABO₃, and the symmetric cubic structure is shown in Fig. R1. B site is coordinated with six oxygens and presented as the corner-sharing BO₆ octahedrons, and is often occupied by the transition metals with stereochemically active lone pair of electrons[2]. The larger A-site cations are accommodated in the interstices among eight BO₆ octahedrons, which can only accommodate alkaline, alkaline-earth, or lanthanide rare-earth metal elements. Among them, metal ion B determines the catalytic performance of the catalyst, and the role of metal ion A is mainly to stabilize the crystal structure. Due to the flexible composition, stable crystal structure, and controllable physical and chemical properties, ABO₃-type perovskite is an ideal sample for studying the surface and catalytic performance of catalysts, thus has been widely applied in many catalytic reactions, such as hydrogen evolution reaction (HER), oxygen reduction reaction (ORR) and oxygen evolution reaction (OER).

[REDACTED]

Fig. R1 ABO₃ perovskite structure showing the surrounding oxygen cage of A-site and B-site cations[2].

Compared with single metal oxides such as CuO, Fe₃O₄, and Co₃O₄, the multiple ions (including metal and oxygen ions) and variable structures of ABO₃ perovskite oxides can bring about some unique electronic and conductive properties, which then modulate the binding energies of reaction intermediates and electron-transport behavior, and consequently their electrocatalytic activities[3]. For example, the A and B sites of simple perovskites can be doped with certain elements such as Sr to increase oxygen vacancies and surface hydroxyl groups for PMS activation[4]. This capability can also be used to produce catalysts with good properties for specific applications. According to the characteristics of dopants, calculation studies show that doping rare earth metals at the A site and less active metals at the B site can improve the durability of the catalyst[5]. As metal-based catalysts, perovskites have better stability (less self-oxidation) than carbon catalysts (modified graphene, biochar, carbon nanotubes, biochar, etc.). The improved stability allows the catalyst to have better durability in the oxidizing environment and can be used for a long time. In addition, carbon catalysts such as carbon nanotubes, nanodiamonds, and ordered mesoporous carbon are relatively inert due to their highly graphitized structures. They are expensive to develop and less attractive than perovskite-based nanomaterials[6].

Generally, PMS activation over perovskite oxides is triggered by withdrawing/donating electrons from catalysts, which mainly rely on electron-rich transition metals at B sites[7,8]. With partial substitution, a unique B-O-B' structure can be formed, which benefits the electronic conductivity due to the strong overlapping between B-site ions and O²⁻ orbital. For example, an unusual super-exchange

effect in Ru-doped SrTiO₃ (STRO) perovskite oxide was discovered, which contributes to the 180° interaction between neighboring Ti³⁺ (with a 3d¹ configuration) and Ru⁵⁺ (with a 4d³ configuration) ions according to the Goodenough-Karamori-Anderson rule, and enhanced electrical conductivity[9]. Besides, the difference in ionic radius can induce structure distortion — the rotations and off-centering of octahedrons, which further enlarges B-O covalency in octahedral configuration, thereby lowering the charge transfer energy to favor catalyst-PMS interaction. According to this process, the electronic conduction relies on the B-O-B' network to greatly promote the operation of the B⁽ⁿ⁺¹⁾⁺/Bⁿ⁺ redox cycle for transition metals, thus resulting in better catalytic performance.

Fig. R3 Schematic illustration of the super-exchange interaction in STRO[9].

Inspired by the above conclusions, we choose the method of doping Cu element at the B-site to induce super-exchange effect and promote electron transfer of LaFeO₃. And as we suspected, LFCO showed superior degradation performance in the removal of nitro aromatic compounds.

We hope these explanations can help you understand the relevant questions, and thank you again for your professional comments.

References:

1. Baek, J. *et al.* Discovery of LaAlO₃ as an efficient catalyst for two-electron water electrolysis towards hydrogen peroxide. *Nat. Commun.* **13**, 7256, (2022).
2. Liu, H. *et al.* Emergence of high piezoelectricity from competing local polar order-disorder in relaxor ferroelectrics. *Nat. Commun.* **14**, 1007, (2023).
3. Hwang, J. *et al.* Perovskites in catalysis and electrocatalysis. *Science.* **358**, 751-756, (2017).
4. Lu, Y. *et al.* Engineering Oxygen Vacancies into LaCoO₃ Perovskite for Efficient Electrocatalytic

Oxygen Evolution. *ACS Sustainable Chem. Eng.* **7**, 2906-2910, (2018).

5. Jacobs, R. *et al.* Material Discovery and Design Principles for Stable, High Activity Perovskite Cathodes for Solid Oxide Fuel Cells. *Adv. Energy Mater.* **8**, 1702708, (2018).
6. Oh, W.-D.; Dong, Z.; Lim, T.-T., Generation of sulfate radical through heterogeneous catalysis for organic contaminants removal: Current development, challenges and prospects. *Appl. Catal. B: Environ.* **194**, 169-201, (2016).
7. Huang, M. *et al.* Facilely tuning the intrinsic catalytic sites of the spinel oxide for peroxymonosulfate activation: From fundamental investigation to pilot-scale demonstration. *Proc Natl Acad Sci U S A.* **119**, e2202682119, (2022).
8. Guo, Z. Y. *et al.* Mn–O covalency governs the intrinsic activity of Co-Mn spinel oxides for boosted peroxymonosulfate activation. *Angew. Chem. Int. Ed.* **60**, 274-280, (2021).
9. Dai, J. *et al.* Single-phase perovskite oxide with super-exchange induced atomic-scale synergistic active centers enables ultrafast hydrogen evolution. *Nat. Commun.* **11**, 5657, (2020).

Comment 2. During ONP degradation, the authors show that the photoinduced electrons are generated and used for ONP reduction, which is the core of the reaction. What is the role of the photogenerated holes in the reaction process?

Response: Thanks for raising the important comments.

As the most renowned earth-saving projects, photocatalysis has been advancing the frontiers for maximizing the efficacious utilization of renewable solar energy to promote a cornucopia of practically feasible chemical reactions since its first discovery in the 19th century. By replicating one of nature's miracles, photocatalysis, an artificial photosynthesis technology employing semiconductor-based materials, is capable of harvesting inexhaustible sunlight to drive thermodynamically uphill reactions. For example, the dissociation of H₂O or oxygen (O₂) generates highly active oxidants which eventually result in the removal of organic pollutants, reduction of heavy compounds, and inactivation of microorganisms. Therefore, photocatalysis technology as an efficient, safe, and environmentally friendly environmental purification technology, for the improvement of water quality has been recognized by the international academic community.

Photocatalysis involves a series of interdependent and continuous photo-charge physics and chemistry. Charge behavior includes generation, separation, recombination, transfer, and capture. Under light irradiation, an electron (e⁻) from the valence band (VB) of the semiconductor experiences

photoexcitation and migrates to the conduction band (CB), leaving an identical amount of holes (h^+) in the VB (Fig. R4 Step (i)). Upon gaining adequate energy from light irradiation, spatial separation, migration, and transportation of photogenerated electron–hole pairs to the surface of the photocatalyst take place (Fig. R4 Step (ii)). On the semiconductor-based photocatalyst's surface, the light-excited charge carriers are readily available to participate in redox reactions with the absorbed electron acceptors (A) and electron donors (D) (Fig. R4 Step (iii)); the semiconductor-based photocatalyst therefore functions as a microscopic electrochemical device[1].

[REDACTED]

Fig. R4 Schematic depiction of the rudimentary steps involved in semiconductor-based photocatalysis[1].

Photogenerated electrons have strong reducing properties, while photogenerated holes have strong oxidizing properties, and the two can form an oxidation-reduction system. Photogenerated holes can interact with H_2O or oxidants adsorbed on the catalyst surface to generate a variety of reactive oxygen species, which can oxidize and mineralize the organic matter. It can also oxidize ions from a low valence state high valence state. In our experiment, effective separation of electrons and holes was achieved after photoexcitation to form various reactive species. We believe that photoelectrons are the key factor for the reduction of ONP in the first step to form active and unstable OAP, while $\bullet OH$ and 1O_2 from PMS activation ensure the rapid degradation and mineralization of pollutants. In addition, the contributions of ROS in the LFCO/PMS/Vis system were evaluated by adding typical chemical probes. To be specific, Quenching experiments were performed to study the effects of ROS during

ONP degradation with adding 0.05 mmol L^{-1} scavengers before the degradation process. Methanol (MeOH, $k_{\text{SO}_4^{\cdot-}/\text{MeOH}} = 3.2 \times 10^6 \text{ M}^{-1} \text{ s}^{-1}$, $k_{\cdot\text{OH}/\text{MeOH}} = 9.7 \times 10^8 \text{ M}^{-1} \text{ s}^{-1}$) can be used to quench both $\cdot\text{OH}$ and $\text{SO}_4^{\cdot-}$, while tert-butyl alcohol (TBA) has much higher rate constant with $\cdot\text{OH}$ ($3.8 \times 10^8 \sim 7.6 \times 10^8 \text{ L mol}^{-1} \text{ s}^{-1}$) than $\text{SO}_4^{\cdot-}$ ($4.0 \times 10^5 \sim 9.1 \times 10^5 \text{ L mol}^{-1} \text{ s}^{-1}$). Carotene, superoxide dismutase (SOD), and $\text{K}_2\text{Cr}_2\text{O}_7$ are easier react with $^1\text{O}_2$, $\cdot\text{O}_2^-$, and photogenerated electrons (e^-), respectively.

As shown in Fig. R5, the contribution of holes during ONP degradation is only 3%, which is negligible. Therefore, we didn't pay too much attention to the photogenerated holes due to their little effect on the degradation process.

Fig. R5. a Effects of different quenchers on the degradation of ONP in the LFCO/PMS/Vis system. **b** Contributions of different ROS. Conditions: $\lambda > 420 \text{ nm}$, $50 \text{ mL } 30 \text{ ppm ONP solution}$, 0.3 mM PMS , $0.3 \text{ g}\cdot\text{L}^{-1}$ powder catalyst, 298 K , initial pH 7.1. 0.05 mmol L^{-1} quenchers were added to the solution before light irradiation.

In addition, the literature has shown that the transfer rate of holes is about two to three orders of magnitude slower than that of electrons, and it is widely accepted that surface holes can be consumed by electron donors[2,3]. In the following work, we will try to construct a photocatalytic system with high hole potential to make full use of electron-hole pairs without compromising the reduction ability of the photocatalyst.

We hope these explanations can help you understand the relevant questions, and thank you again for your professional comments.

Fig. R5. Corresponds to Supplementary Figure 39.

References:

1. Fung, C. M. *et al.* Red Phosphorus: An Up-and-Coming Photocatalyst on the Horizon for

Sustainable Energy Development and Environmental Remediation. *Chem. Rev.* **122**, 3879-3965, (2022).

2. Xin, Z. K. *et al.* Rational Design of Dot-on-Rod Nano-Heterostructure for Photocatalytic CO₂ Reduction: Pivotal Role of Hole Transfer and Utilization. *Adv. Mater.* **34**, 2106662, (2022).
3. Li, Y. *et al.* Revealing the Importance of Hole Transfer: Boosting Photocatalytic Hydrogen Evolution by Delicate Modulation of Photogenerated Holes. *ACS Catal.* **13**, 8281-8292, (2023).

Comment 3. In terms of large-scale applications, how are the main parameters selected and set? The author's explanation of the condition setting of the simulation is not sufficient, please provide sufficient evidence.

Response: Thanks for raising the important comments.

Advanced oxidation processes (AOPs) are unique techniques for removing organic pollutants in liquid media using the oxidizing potential of radicals generated via chemical or physical pathways. Despite fundamentals of AOPs are well understood, industrial applications of AOPs are yet to adopt scalable approaches that allow more efficient, portable, cost-effective, and environment-friendly AOPs operations. Continuous growth in industrial business operations needs technologies to adapt to evolving industrial demand. For example, Saravanane et al. estimated that around 50 GL wastewater will be generated in India during the projected year of 2051[1]. These challenges might be better addressed with AOP systems, however, the principal challenges in adapting AOPs to cope with the widely varying demand of the industry are scalability of the AOPs (i.e., up-scaling techniques of ROS production), cost of reagent and energy usage, as well as difficulties in rapid and in-situ measurement of ROS production efficiency via AOPs. In line with the above principles, we consider and set the parameters according to the following aspects:

1. Using continuous-flow reactor

The core function of an AOP is its ability to produce ROS. The capability of transporting radicals from the point of production to the point of delivery as well as the capability of their efficient dosing will have an impact on the scalable application of different radicals for industrial purposes. While short half-lives are not problematic in AOPs (sometimes can be beneficial), this fact needs to be taken into consideration during the engineering design of continuous-flow reactors for scalable production of radical species. For example, radicals with short half-lives such as •OH (10^{-4} μsec) and SO₄^{•-} (30 ~ 40 μsec) must be produced in situ at the location of their application rather than transported after

production. Therefore, it is imperative to employ a continuous-flow reactor for in-situ production and dosing of ROS. Notable continuous-flow technologies employed for AOPs for the treatment of wastewater are fixed-bed reactors, which are energy-efficient, low-cost, easy to construct and set up, flexible to adapt to the fluctuation in influent water characteristics and flow rate, and simplicity of operation. This is why we chose CFC as a fixed bed.

2. Optimizing the parameter

Maintaining the effective concentration of radicals until reaching the target is a major obstacle in the development of highly efficient AOPs. From the beaker experiments under laboratory conditions, we find that LFCO achieves the highest PMS activation rate for generating high-concentration ROS, resulting nearly 100% ONP removal rate. This shows that our work has large-scale application potential. As lab-scale experimental conditions are significantly different from actual water/wastewater matrix and operational conditions, a successful laboratory treatment process does not guarantee its success on a larger scale. Pilot-scale studies are, thus, required to simulate real treatment processes before full-scale execution. A proper pilot-scale plant with a whole set of “real” operational parameters can provide an accurate portrait of a full-scale plant, albeit some cautions of the scaling must be taken into account. Then we constructed the LFCO@CFC FRB reactor and optimized the operational parameters including water flux, hydraulic residence time, the dosage of catalyst and PMS, and light intensity. Finally, a toxic assessment was conducted. According to LC-MS analysis, the degradation and transformation process of pollutants was determined. Finally, T.E.S.T. was applied to analyze the toxicity of the products to ensure the non-toxicity and harmlessness of the final products. At the same time, in the reaction process, the leaching concentration of heavy metal ions is also obviously in line with the national sewage discharge standard.

3. large-scale applications

we first combined with our own small laboratory pilot equipment, and referred to the current plant treatment scale of o-nitrophenol, according to the ratio of 1:50 for equal scale amplification. To prevent unexpected situations in the actual treatment, we increased the catalyst load to twice the original content; At the same time, various measures such as light intensity and coverage area were increased to ensure the treatment effect of the pilot plant. In terms of the operation cycle of the device, we have compared the cyclic experiment results in the laboratory to show that the material has strong stability performance, only 2% loss in the seven-day reaction cycle, and the reaction rate is maintained at nearly

100%, indicating that the material has very superior stability and removal effect. Therefore, we have reason to believe that after strengthening the setting of various optimal parameters and improving the catalyst load rate, the operation cycle can be maintained for 30 days.

We hope these explanations can help you understand the relevant questions, and thank you again for your professional comments.

Reference:

1. Saravanane, R. , et al. Urban Wastewater Treatment for Recycling and Reuse in Industrial Applications. *Industrial Wastewater Treatment, Recycling and Reuse*, 283-322, (2014).

Comment 4. How to ensure that the material can achieve a 30-day operation cycle in large-scale applications?

Response: Thanks for raising the important comments. Perovskite is a kind of material with a special crystal structure and excellent photoelectric properties, which is widely used in solar cells, light-emitting diodes, lasers, and other fields[1]. In our manuscript, the LFCO/PMS/Vis system achieved the highest ONP degradation rate of 0.079 min^{-1} , which is 7.7 times higher than that of the LFO/PMS/Vis system (0.009 min^{-1}). This encouraged us to move from laboratory-scale towards device modules. However, perovskite materials also have serious stability problems, mainly in the temperature, humidity, light, and other factors, perovskite crystals will undergo phase transition, resulting in device performance degradation or failure. Therefore, preserving the catalytic activity and structure stability of perovskites is a key factor in achieving long-term operational stability.

Hence, the stability and durability of LFCO@CFC were investigated by continuous experiments with time extended to 7 days. The result showed that the degradation efficiency of ONP remained > 99 %. Besides, ICP-MS analysis showed that the leaching concentration of Cu and Fe was negligible. Furthermore, a clear consensus has been reached that the fast activity decay is predominantly rooted in surface chemical reactions and particle cracking of catalysts.

To further illustrate the above questions, we add the following characterizations and descriptions in the revised version:

1. The intensity/half peak width of XRD peaks is not only dependent on the particle size but also the content of the crystal phase. After a 7-day catalytic reaction, we can see that the used LFCO catalyst shows the same peaks with fresh LFCO (Fig. S18a), indicating that no phase transition during the reaction. As shown in SEM results (Fig. S18b), the catalyst is still tightly bound to CFC, indicating the

active components were not lost with time. Besides, TEM results of used LFCO catalysts show that they maintained the particle structure and size with a clean surface and uniform distribution of elements, which is well agreed with XRD data. Results together proved the structure stability of LFCO catalysts.

Fig. R6 a XRD, b SEM, c TEM images and corresponding EDS spectra of used LFCO.

2. XPS analysis of the used and fresh LFCO catalysts offered valuable information about the valence state and the surface composition. As displayed in Fig. R7 and Table R1, all elements remain in clear outline and unchanged ratio, demonstrating the excellent stability of LFCO. As shown in Fig. R7b, although Fe^{3+} ions are mainly in the form of structure for both catalysts, the binding energy of Fe 2p_{1/2} level in used LFCO is lower than that of fresh LFCO, while Cu^{2+} has the opposite trend. Charge transfer from partially reduced neighboring Cu atoms would increase the electron density on Fe causing the shift of binding energy. By following the redox cycle between Fe and Cu, the superior catalytic activity and outstanding stability of the LFCO catalyst were perpetuated even after 7 days.

Fig. R7 a The La 3d, b Fe 2p, c Cu 2p, and d O 2p XPS spectra of fresh and used LFCO.

Table R1. Elements ratio of catalyst detected by XPS.

Catalyst	La (atomic %)	Cu (atomic %)	Fe (atomic %)	O (atomic %)
LFO	13.30	0	22.91	63.79
LFCO	14.76	4.61	17.36	63.27
LFCO-used	14.47	4.59	17.41	63.53

3. In the manuscript:

Moreover, the degradation efficiency of ONP remained at > 99 % in 7-day continuous experiments with low metal leakage (Fig. 3e). Besides, the similar XRD, SEM, TEM, and XPS results of fresh and used LFCO together proved the stability and durability of the catalyst.

We hope these explanations can help you understand the relevant questions, and thank you again for

your professional comments.

Fig. R6 corresponds to Supplementary Figure 17;

Fig. R7 corresponds to Supplementary Figure 18;

Table R1 corresponds to Supplementary Table 4.

Reference:

1. Liu, X. *et al.* Stabilization of photoactive phases for perovskite photovoltaics. *Nat. Rev. Chem.* **7**, 462-479, (2023).

Comment 5. At present, the large-scale application of advanced devices in AOPs is imminent, and many researches have made efforts for it. Does the reduction-oxidation coupling mechanism and large-scale treatment equipment have the same removal effect on other refractory aromatic compounds and hydrocarbon chain compounds?

Response: Thanks for raising the important comments.

1. Theoretical feasibility

In AOPs, ROS, such as $\bullet\text{OH}$, $\text{SO}_4^{\bullet-}$, and $^1\text{O}_2$, are among the strongest aqueous redox species, and their effective and efficient production is desired in biological, chemical, and environmental fields. Researchers have begun to perceive the high activation barrier of the electron-withdrawing groups (EWGs, like X-F, Cl, Br, I; $-\text{NO}_2$; $-\text{COCH}_3$, $-\text{CN}$, etc.) not as a movable obstacle, but rather as a practical challenge that needs to be addressed. Lai et al proved that contaminants' molecular structure affects their degradation efficiencies and dominant reactive oxygen species (Fig. R8)[1]. For instance, the phenolic group with rich electrons tends to react with $^1\text{O}_2$, whereas $\bullet\text{OH}$ prefers to directly attack specific functional groups via an electron transfer (ET) reaction and trends to react with electron-donation groups such as amino ($-\text{NH}_2$), hydroxyl ($-\text{OH}$), and unsaturated bonds, etc. Also, electron-donating groups will promote degradation, whereas the strong electron-withdrawing group could extremely restrain the degradation of contaminants. Therefore, the structure of pollutants may affect contaminants' degradation efficiencies and the dominant working species of ROS in the multiple ROS system. It is very important to select a suitable degradation path for a class of contaminants according to the structure. The combination of quenching experiments and/or probe compound tests, EPR tests, and density functional theory (DFT) can preferably be conducted with a deep and comprehensive exploration of the selective attack of ROS toward organic contaminants with special structure.

[REDACTED]

Fig. R8 Correlation of k_{obs} to (a) $EHOMO$, (b) VIP , (c) ΔE , and (d) simultaneous $EHOMO$, VIP , and $\Delta E[1]$.

Reductive electrons have a strong electrophilicity and are easy to react with electron-rich groups. Besides, the Fukui function index can indicate the reactive sites on molecules with the principle that a higher Fukui index indicates a higher reactivity of the corresponding reactive sites. Fig. R9 is a visual representation of the electron distribution. It is worth noting that the N12 atom ($f^+ = 0.1121$) with a high Fukui index is the most reactive site and easy to be attacked by e^- , leading to the rapid reduction of the $-NO_2$ to $-NH_2$ groups to form OAP (P3). Therefore, pollutants with EWGs, like polychlorinated biphenyl, perfluorochemical, and chlorine-containing plastics, are all possible to be removed by the reduction-oxidation coupling degradation process.

Fig. R9 Charge density difference of condensed Fukui index (f^+) of ONP.

2. Experimental feasibility

In this work, the LFCO@CFC FBR system achieves an efficient removal rate for a wide variety of nitroaromatic pollutants including ONP, p-nitrophenol (PNP), o-nitrotoluene (ONT), and p-nitrotoluene (PNT) (Fig. R10c and d). For other pollutants, such as chlorine-containing plastics, it is also the direction of our future work, and the reduction-oxidation coupling system will be further studied.

Fig. R10 Application in water treatment. **A** Toxicity indicators of parent ONP and its degradation products. **B, C** Effects of different inorganic anions(**B**) and water quality (**C**) to the degradation efficiency. **D** Degradation of different nitro contaminants in LFCO@CFC FBR system. FBR conditions (unless indicated otherwise): $\lambda > 420 \text{ nm}$ radiation, Water flux 7500 mL h^{-1} , HRT 60 s , 298 K , initial pH 7.1 , 30 ppm ONP, and 3 mM PMS (introduced with the ONP stock solution). **E** Diagram of the large-scale application plant including 5 main parts: (1) wastewater collection (2) PMS dosing tank, (3) physical filtration, (4) FBR reactor, (5) effluent tank (data from 1:50 for equal scale amplification). The dimensions of core parts 2 and 4 are marked in the enlarged image on the right. **F** Performance and cost comparison of reported catalysts.

We hope these explanations can help you understand the relevant questions, and thank you again for your professional comments.

Fig. R9 corresponds to Fig. 4g; Fig. R10 corresponds to Fig. 5.

Reference:

1. Xie, Z.H. *et al.* Effects of Molecular Structure on Organic Contaminants' Degradation Efficiency and Dominant ROS in the Advanced Oxidation Process with Multiple ROS. *Environ. Sci. Technol.* **56**, 8784-8795, (2022).

Comment 6. Degradation and removal appear many times in the article, the definitions of the two are different, please pay attention to distinguish and confirm.

Response: Thanks for raising the important comments.

Degradation refers to the conversion of pollutants into non-toxic and harmless carbon dioxide and water, while removal refers to the conversion of pollutants into small molecule chemicals. Therefore, when emphasizing the conversion steps of o-nitrophenol, we focus on the removal to express the transformation and migration of o-nitrophenol; While focusing on the whole reaction process, we focus on the expression of complete mineralization of o-nitrophenol by degradation.

Thanks for your tip, we hope these explanations can help you understand the relevant questions, and thank you again for your professional comments.

Comment 7. In general, the manuscript is very well written. However, I did not see some confused mistakes in the paper. Material subscript formats for Line 83 and Line 84 need to be corrected.

Response: Thanks for raising the important comments. According to the reviewer's suggestion, we have carefully revised the mistakes.

Comment 8. All nitro groups (-NO₂) must be written in the same format.

Response: Thanks for raising the important comments. According to the reviewer's suggestion, we have carefully revised the mistakes.

General comments from Reviewe 3:

The manuscript presents a systematic study on the synthesis of a photocatalytic material and its application. Authors fail to provide the key pieces of information and background to support their hypothesis. The results presented do not demonstrate the dual role as the lack of blank experiments difficults getting to any sound conclusion. The most worrisome is that several claims are done all thorough the text in disagreement with current literature. Similarly, several subjective interpretations result in conclusions not proven experimentally in the manuscript. Unfortunately the manuscript cannot be recommended for publication.

Response: Thanks for raising the important comments.

Nitroaromatic compounds (NACs) are a typical kind of environmental heterogeneous biomass, widely used in the production of various drugs, dyes, pesticides, pesticides, and other chemical products. Most of these compounds have carcinogenic, teratogenic, and mutagenic properties, with high toxicity, carcinogenicity, and bioaccumulation. Therefore, NACs will cause serious harm to the human and ecological environment if they are not effectively treated. A comprehensive understanding of the characteristics of the target water is crucial for tailoring the treatment process for satisfied treatment efficiency, low oxidant usage, and energy input, which, in many cases, is ignored. According to previous studies, NAC-contained industrial wastewater has the characteristics of high salinity, high concentration, complex composition, high toxicity, and poor biodegradability. Its remediation is still a difficult task. How to efficiently degrade NACs in complex water quality in practical application is a great challenge.

1. For the first challenge: developing suitable degradation approaches

At present, most of the research on the degradation of NACs focuses on a single oxidation system, and a single oxidation method makes it difficult to achieve efficient removal. Because the $-\text{NO}_2$ group is a strong electron-withdrawing group and induces the overall delocalization effect of π electrons in the benzene ring structure, thus promoting stability. Therefore, contaminants with the electron-withdrawing group are difficult to degradation. Researchers have begun to perceive the high activation barrier of the electron-withdrawing groups (like X-F, Cl, Br, I; $-\text{NO}_2$; $-\text{COCH}_3$, $-\text{CN}$, etc.) not as an immovable obstacle, but rather as a practical challenge that needs to be addressed. It is very important to select a suitable degradation path for a class of contaminants according to the structure. The combination of quenching experiments and/or probe compound tests, EPR tests, and density functional

theory (DFT) can preferably be conducted with a deep and comprehensive exploration of the selective attack of ROS toward organic contaminants with special structures.

Mao et al think that the efficiency of persulfate-assisted advanced oxidation processes (PS-AOPs) in degrading organic pollutants is affected by the electron-donating capability of organic substances present in the water source[1]. They systematically investigate the electron-donating capacity (EDC) difference between groundwater and surface water and demonstrate the dependence of removal efficiency on the EDC of target water by PS-AOPs with carbon nanotubes (CNTs) as a catalyst (Fig. R1). The study provides valuable insights into the design and application of PS-AOPs for feeding water treatment based on the selection of catalyst, the regulation of the reaction pathway, and the EDC of target water. This study offers implications for groundwater quality control and surface water management.

[REDACTED]

Fig. R1 Schematic illustration of the selective removal of organic pollutants for groundwater and surface water by the CNT/PDS activation system[1].

For another example, Lai et al disclosed that the dominant ROS was the same one when treating similar types of contaminants (Fig. R2) [2]. This phenomenon may be caused by the contaminants' structures, especially the commonly shared or basic parent structures which can affect their effective reaction time and second-order rate constants with ROS. They also revealed the non-negligible role of substituents on organic compounds in affecting the removal efficiency of the compounds in the PMS activation system. The existence of -F, a very strong electron-withdrawing group, could extremely restrain the degradation of OFL and CIP. It is worth mentioning that degradation intermediates, the existing forms of target contaminants, and the anions in the reaction system have limited effects on the degradation efficiencies, which hardly influence the degradation orders of various contaminants.

Overall, the new insights gained in this study provide a basis for designing more effective AOPs to improve their practical application in wastewater treatment. Many studies have confirmed this and put forward that break the electron-withdrawing groups is one of the most critical steps during degradation[3-6], which is also consistent with our research background.

[REDACTED]

Fig. R2 Correlation of k_{obs} to (a) $EHOMO$, (b) VIP , (c) ΔE , and (d) simultaneous $EHOMO$, VIP , and ΔE [2].

Although NACs are not easy to oxidize, it is easy to reduce to form amino aromatics. Zhao et al modeled the k (the second order rate constants of e^- with organic compounds) values for aliphatic compounds and phenyl-based compounds for the first time by the quantitative structure-activity relationship (QSAR) method[7]. The electron-donating groups will promote degradation, whereas the strong electron-withdrawing group could restrain the degradation of contaminants. The thermodynamic investigation suggests that the compounds with electron-withdrawing groups tend to possess higher k and lower Gibbs free energy and Gibbs free energies of activation than the ones with electron-donating groups, indicating the electron transfer pathway occurs more readily (Fig. R3). This is also proved by the Fukui function index in our manuscript, which can indicate the reactive sites on molecules with the principle that a higher Fukui index indicates a higher reactivity of the corresponding

reactive sites. Fig. R4 is a visual representation of the electron distribution. It is worth noting that the N12 atom ($f^+ = 0.1121$) with a high Fukui index is the most reactive site and easy to be attacked by e^- , leading to the rapid reduction of the $-NO_2$ to $-NH_2$ groups to form OAP. On the one hand, the reduction product of the $-NO_2$ group $-NH_2$ group can greatly reduce the toxicity of NACs wastewater. On the other hand, the $-NH_2$ group is an electron-donating group, which can weaken the stability of the benzene ring and is easily oxidized into non-toxic and harmless substances in water, which greatly improves the biodegradability of wastewater through the effect of electron-donating. Therefore, it is necessary to explore the reduction-oxidation coupling degradation mechanism of nitroaromatic hydrocarbons based on current chemical oxidation methods.

[REDACTED]

Fig. R3 Box plots of $\log k$ values (A) and ΔG_{SET} values (B) for 103 substituted benzenes (A compound containing multiple functional groups is counted in the stack of each functional group present, and the dots represent outliers)[7].

Fig. R4 Charge density difference of condensed Fukui index (f^+) of ONP.

As for the inconsistent literature mentioned by the reviewer, we searched relevant literature and found that Zhang et al believed that nitro, as an electron-absorbing substituent, was not conducive to

the oxidation ring-opening of pollutants. A reduction-oxidation coupling (ROC) process was constructed by zero-valent iron (ZVI) and hydrogen peroxide (H_2O_2) to achieve high-efficiency mineralization of nitrochlorobenzene (CNBs)¹⁵. Cheng et al believe that the removal of nitro-aromatic pollutants is a difficult problem in the field of environmental engineering, but the oxidation and degradation of nitro-aromatic pollutants can be effectively promoted by converting them into aromatic amines. Then the micro-electrolysis process with both oxidation and reduction characteristics is designed to provide ideas for efficient treatment of nitro-aromatic compounds[8]. Yang et al believe that the degradation of NACs can be achieved by chemical oxidation, but the $-\text{NO}_2$ group is a strong electron-absorbing group, which promotes the stability of the benzene ring structure, making NACs not easy to oxidize but easy to reduce. Therefore, it is necessary to develop an efficient redox degradation method of high concentration NACs[9]. Therefore, this work is in line with the current research focus.

2. For the second challenge: practical application in complex water quality

In AOPs, ROS, such as $\bullet\text{OH}$, $\text{SO}_4^{\bullet-}$, and $^1\text{O}_2$, are among the strongest aqueous redox species, and their effective and efficient production is desired in biological, chemical, and environmental fields. Radical oxidation and nonradical oxidation processes have significant differences in oxidation mechanism, redox potential (relatively mild), evolutionary pathways, and reaction zone (surface region). For instance, $\bullet\text{OH}$ -based oxidation with higher oxidation ability (2.4 ~ 2.7 V vs NHE) but indiscriminateness, they are easily quenched by water matrix in industrial wastewater, resulting in ineffective NACs removal rate and limiting the large-scale application of AOPs. Nonradical oxidation processes with relatively weaker oxidation ability (e.g. $^1\text{O}_2$, 0.81 V vs NHE), however, they have excellent independence of solution pH, strong resistance to versatile inorganic ions and background organic matters, which universally exist in domestic sewage and industrial wastewaters, and can significant reduction of toxic by-products. Also, nonradical oxidation processes can avoid the inefficient consumption of PMS, thus improving its stoichiometric efficiency. These characteristics fully indicate that nonradical-based catalytic oxidation reaction has a broad research and application prospect in the field of environmental pollution treatment[10]. At the same time, the reaction rate of organic pollutants with $\text{SO}_4^{\bullet-}$ and $\bullet\text{OH}$ is generally high ($10^7 \sim 10^9 \text{ M}^{-1} \text{ s}^{-1}$), and the reaction rate of organic pollutants with $^1\text{O}_2$ is low ($10^4 \sim 10^7 \text{ M}^{-1} \text{ s}^{-1}$), indicating that $^1\text{O}_2$ is difficult to act as the dominant ROS in the degradation process of pollutants. However, in the degradation of certain phenols,

although $\text{SO}_4^{\cdot-}$ and $\cdot\text{OH}$ are present in higher concentrations, the ROS that plays a dominant role in the system is $^1\text{O}_2$ rather than $\text{SO}_4^{\cdot-}$ and $\cdot\text{OH}$. Studies have shown that some ROS are more likely to attack certain special structures or groups of pollutants. For example, electron-rich phenol groups tend to react with $^1\text{O}_2$, while $\text{SO}_4^{\cdot-}$ tends to directly attack specific functional groups through electron transfer reactions, and tends to react with electron-donating groups such as amino ($-\text{NH}_2$), hydroxyl ($-\text{OH}$), and unsaturated bonds. Therefore, how to mobilize and give full play to the reactivity and characteristics of different free radicals is also our research focus. Therefore, one promising strategy for solving the second challenge is to make use of the complementary advantages of radicals and non-radicals.

The synergistic generation mechanism and principle of action have been extensively studied and consensus reached in the AOPs field. Lai Lyu and his co-worker proposed the “dual-reaction-center” theory[11]. Doping with O induced bidirectional transfer of electrons on aromatic rings through C-O-C bonding [$\text{C}\rightarrow\text{O}\leftarrow\text{C}$], which led to the formation of micro-electronic fields on the catalyst surface. As shown in Fig. R5, PMS molecules are oxidized around the electron-poor C areas, meanwhile, a small proportion of the PMS molecules was reduced around the electron-rich O areas. These synergistic multi-pathways provide new insights into the use of the energy of pollutants for heterogeneous PMS activation. The construction of surface micro-electronic fields on metal-free materials also provides a facile approach to significantly improve the catalytic activity of Fenton-like catalysts for processes such as environmental remediation.

[REDACTED]

Fig. R5 Schematic diagram of the heterogeneous PMS activation process[11].

Zhan et al constructed asymmetric Co-O-Bi sites to achieve rapid electron cycling while achieving site synergy, thus effectively activating PMS to generate radicals and nonradicals (Fig. R6)[12]. After Co doping, charge redistribution through Co-O-Bi sites promotes the feedback of electrons from Co to Bi, which not only provides electron-rich Bi active sites but also creates electron-poor Co catalytic

sites. Specifically, the construction of an asymmetric site can increase the charge density of individual Bi atoms and promote the reduction of PMS to form more $\text{SO}_4^{\cdot-}$ and $\cdot\text{OH}$. On the contrary, the electron-poor Co sites make it possible for PMS to be oxidized to $^1\text{O}_2$. Therefore, the efficient redox of PMS can be achieved through the electron cycle between the Co and Bi sites on the asymmetric Co-O-Bi site, realizing the efficient removal and mineralization of pollutants without interference from organic and inorganic compounds under the environmental background. Through the synergistic effect of free radicals ($\text{SO}_4^{\cdot-}$ and $\cdot\text{OH}$) and non-free radicals ($^1\text{O}_2$), 10 ppm SMX can decompose almost 100% within 3 min with a k-value of up to $82.95 \text{ min}^{-1} \text{ M}^{-1}$, making it superior to the existing catalysts reported so far.

[REDACTED]

Fig. R6 Proposed mechanism for the SMX oxidative catalyzed by Co-BOC[12].

In our work, Cu Substituting induces the super-exchange effect and promotes the transfer of photoelectrons, which can reduce $-\text{NO}_2$ to initiate o-nitrophenol (ONP) degradation, also, it can modulate local electronic configuration to form electron-poor/rich sites for PMS dual activation to synergetically generate radicals and non-radicals in the subsequent oxidation process. Finally, the LFCO/PMS/Vis system achieved reduction-oxidation coupling (ROC) degradation of ONP with a degradation rate of 0.079 min^{-1} , which is 7.7 times higher than that of LFO (0.009 min^{-1}). Therefore, our results are in disagreement with current literature.

3. blank experiments and more evidence.

In our manuscript, we conducted a double-blank experiment to investigate the degradation effects of single reduction and oxidation methods on nitroaromatic hydrocarbons in the photosystem and PMS

systems, respectively. To further investigate the catalytic performance, catalysts with different Cu ratios were tested in different systems. The HPLC results in Supplementary Figures 3 and 4 showed that the degradation effect of ONP under a single reduction or oxidation system was not ideal, which was far inferior to the removal rate in the reduction-oxidation coupling system. Meanwhile, LFCO has the best catalytic performance and was chosen for further study. Besides, *in-situ* UV-Vis absorption spectra of ONP during the reaction (Supplementary Figure 36-38) proved that in the PMS system, the degradation rate of ONP is low and OAP is not generated. While under the photocatalytic condition, though the peak of ONP decreases slowly, the characteristic peaks of OAP increase first and then decrease, indicating that part of ONP is reduced to OAP during degradation. Results show that a reduction-oxidation coupled degradation process system was successfully designed, in which the parent NACs were reduced firstly to accelerate the reaction rate, and then the reduction products were oxidized and mineralized. This can be proved by TOC results. TOC is a concise measurement of organic matter concentration, while UV absorbance can provide a supplementary basis for characterizing samples. UV absorbance must be used together with TOC data for sample comparison. Furthermore, this also shows the dominance of photoreduction in the process of reducing ONP to OAP, and reactive oxygen species are responsible for the subsequent ring-opening process of OAP.

According to the reviewer's suggestion, we added some key pieces of evidence. Firstly, we have added the HPLC experiments to test OAP formed during the reaction. It has the advantages of a wide analysis range, strong separation ability, reliable qualitative analysis results, low detection limit, and fast analysis time, and is currently the most powerful tool for environmental monitoring. As shown in Fig. R7f, upon the decline of the ONP concentration, the formation of OAP was observed immediately (within 5 min). Notably, negligible concentration of both ONP and OAP was achieved upon completion of the reaction, which indicates complete oxidative degradation. Thus, ONP may undergo a sequential ROC pathway ($\text{ONP} \rightarrow \text{OAP} \rightarrow \text{CO}_2 + \text{H}_2\text{O}$) only in the LFCO/PMS/light system. For other errors in the paper, we have corrected them according to the opinions of other reviewers and made detailed explanations. By measuring the concentration changes of OAP in different systems, it is proved that the first step of the reduction-oxidation coupling mechanism is that the $-\text{NO}_2$ group is reduced to the $-\text{NH}_2$ group through photogeneration of electrons, which proves that the reduction process is the premise and basis of the oxidation process, and then the product is oxidized and degraded by various free radicals generated by the activation of PMS. The rigor and authenticity of the reduction-oxidation

coupling

Fig. R7 Reduction-oxidation coupling (ROC) mechanism and degradation pathway of ONP. A The band gap diagram of LFO and LFCO. **B** Tauc plots for LFO and LFCO using $(F(R)h\nu)^2$ (Kubelka-Munk parameter) as a function versus the photon energy (The x-axis intersection point of the linear fit of the Tauc plot gives an estimate of the band gap energy). **C** Proposed mechanism for the PMS reaction on the surface of LFCO. **D** EPR spectra of DMPO- $\bullet\text{OH}$ /DMPO- $\text{SO}_4^{\cdot-}$ and TEMP- $^1\text{O}_2$ in PMS/Vis, LFO/PMS/Vis, and LFCO/PMS/Vis system, respectively. **E** In-situ UV-Vis absorption spectra of ONP degradation in the LFCO/PMS/Vis system. **F** HPLC spectra and corresponding concentration changes of ONP and OAP during degradation in the LFCO/PMS/Vis system. **G** Charge density difference of condensed Fukui index (f^+) of ONP. **H** The degradation pathway of ONP in the LFCO/PMS/Vis system. The green arrow represents the reduction stage induced by e^- , The green arrow represents the oxidation stage induced by $\bullet\text{OH}$ and $^1\text{O}_2$. Conditions: $\lambda > 420$ nm, 50 mL 30 ppm ONP solution, 0.3 mM PMS, 0.3 $\text{g}\cdot\text{L}^{-1}$ powder catalyst, 298 K, initial pH 7.1.

According to your suggestion, we have carried out degradation experiment using actual industrial wastewater. As shown in Fig. R8 and Table R2, the stock wastewater solution (0 min) contained a

variety of benzene species including ONP. After ten minutes of degradation, we found that the intensity of the ONP peak decreased and a characteristic peak of OAP appeared (Table R3), while after 30 minutes, both the ONP and OAP characteristic peaks disappeared (Table R4). In addition, the benzene ring structure has disappeared, and many new short-chain small molecule compounds have emerged. This strongly demonstrates the practical applicability of the reduction-oxidation coupling mechanism.

Fig. R8 GC-MS spectra of industrial wastewater during degradation at 0min, 10 min, and 30 min.

Table R2. Compounds detected in wastewater by GC-MS at 0 min.

Peak	Retention time	Compound	Area%
1	3.028	Acetic acid	20.72
2	3.629	Dimethyl sulfone	0.5
3	3.806	Methanesulfonic anhydride	2.25
4	4.167	Trimethylsilyl ethaneperoxoate	56.32
5	4.396	Silanediol	6.94
6	10.926	Cyclotetrasiloxane,	2.72
7	11.916	4-Hydroxypiperidine	1.7
8	12.29	2-Azabicyclo[3.2.1]octan-3-one	0.3

Table R2 (continued). Compounds detected in wastewater by GC-MS at 0 min.

9	12.619	Cyclopentanecarboxylic acid	0.42
10	12.923	3-Hydroxybenzoic acid	0.24
11	13.252	5H-1-Pyridine	0.91
12	13.608	O-nitrophenol	1.38
13	13.859	2-(2-Hydroxycyclohexyloxy)pyridine-N-oxide	0.23
14	14.588	3-Amino-Phenol	0.76
15	15.882	p-Isopropenylphenol	0.05
16	16.021	2-Allylphenol	0.12
17	16.095	2,6-Dihydroxybenzoic acid	0.27
18	16.184	Cyclohexasiloxane	0.96
19	16.305	Benzocycloheptatriene	0.1
20	16.419	2-Naphthalenol	1.04
21	16.788	Cinnoline	0.15
22	16.921	2,6-Dihydroxybenzoic acid	0.11
23	17.291	Cyclopentasiloxane, decamethyl-	0.19
24	17.794	7-Methyl-1-naphthol	0.05
25	19.15	2,6-Dihydroxybenzoic acid	0.29
26	19.377	3-Methylsalicylic acid	0.37
27	19.456	1-Naphthalenecarbonitrile	0.12
28	19.663	1-Naphthalenecarbonitrile	0.08
29	36.247	Palmitoleamide	1.11

Table R3. Compounds detected in wastewater by GC-MS at 10 min.

Peak	Ret. time	Compound	Area%
1	3.038	Ethane	24.33
2	3.605	Methanesulfonic acid	0.31
3	3.681	Methanesulfonic anhydride	1.71
4	3.785	Cholest-8-ene-3,6-diol	2.47
5	3.845	Silanediol	3.65
6	3.904	Trimethylsilyl ethaneperoxoate	7.74
7	3.951	Silanediol	12.82
8	4.008	Silanediol	7.98
9	4.091	Trimethylsilyl ethaneperoxoate	28.92
10	4.313	1-Methylbutylmandelate	0.11
11	10.85	Cyclotetrasiloxane, octamethyl-	0.96
12	10.93	Cyclotetrasiloxane, octamethyl-	0.64
13	12.767	5H-1-Pyridine	0.64
14	13.249	5H-1-Pyridine	1.38
15	13.604	Cyclopentasiloxane	1.18
16	13.729	O-nitrophenol	0.13
17	14.594	Ortho-aminophenol	0.37
18	14.635	Cyclotetrasiloxane, octamethyl-	0.15
19	16.021	O-nitrosophenol	0.21

Table R3 (continued). Compounds detected in wastewater by GC-MS at 10 min.

20	16.093	2,6-Dihydroxybenzoic acid	0.32
21	16.183	Cyclohexasiloxane	0.94
22	16.419	2-Naphthalenol	1.23
23	16.789	Cinnoline	0.26
24	19.377	Cyclohexadione diene	0.33
25	23.319	5H-Indeno[1,2-b]pyridine	0.25
26	28.354	Butenedioic acid	0.24
27	33.104	Muconic acid	0.13
28	36.248	Palmitoleamide	0.59

Table R4. Compounds detected in wastewater by GC-MS at 30 min.

Peak	Ret. time	Compound	Area%
1	3.029	Ethane	24.8
2	3.515	tert-Butyldimethylsilanol	1.25
3	3.568	2,4,5,6,8-Pentathianonane	1.08
4	3.668	Silanediol	2.71
5	3.716	Silanediol	2.93
6	3.768	Silanediol	9.46
7	3.879	Trimethylsilyl ethaneperoxoate	9.34
8	3.985	Silanediol	16.36

Table R4 (continued). Compounds detected in wastewater by GC-MS at 30 min.

9	4.04	1-Methylbutylmandelate	10.14
10	4.08	Trimethylsilyl ethaneperoxoate	12.8
11	10.836	Cyclotetrasiloxane	2.59
12	12.753	5H-1-Pyridine	0.82
13	13.235	5H-1-Pyridine	1.25
14	13.603	Ortho-aminophenol	0.17
15	14.326	1H-Indole	0.17
16	14.588	Phenol	0.49
17	16.182	Cyclohexasiloxane	0.69
18	16.418	2-Naphthalenol	0.65
19	16.786	Cyclohexadione diene	0.05
20	19.373	Hexasiloxane	0.27
21	19.451	1-Naphthalenecarbonitrile	0.11
22	22.548	5H-Indeno[1,2-b]pyridine	0.12
23	23.029	2,6-Dihydroxybenzoic acid	0.14
24	23.318	5H-Indeno[1,2-b]pyridine	0.25
25	28.347	Butenedioic acid	0.07
26	36.244	Muconic acid	0.02

We also found that after the coupling mechanism of reoxidation and reduction, the contents of long -NH₂ chain alkanes (Palmitoleamide), aromatic hydrocarbons alcohols and unsaturated aromatic hydrocarbons (1-Naphthalenecarbonitrile, etc.) at 0 minutes were reduced, while the contents of sulfur alkanes and long chain alkanes containing Si were also continuously decreased, indicating that

the degradation system can not only effectively degrade nitrobenzene organics, but also mineralize and remove other common pollutants in coking wastewater, such as long-chain alkane, aromatic hydrocarbon and sulfur-based compound. The degradation system has strong degradation ability and stability, and a high application prospect in actual wastewater treatment.

In the reaction process, the content and types of aminobenzene organic compounds increased at 10 minutes, indicating that photogenerated electrons can not only reduce -NO₂ of o-nitrophenol, but also reduce other nitrobenzene compounds, which verified the experimental results of this work. The content of amino alkanes and ethane also increased, which may be due to the changes in the molecular structure of organic matter caused by the reaction, ¹O₂ and •OH can attack many kinds of pollutants in coking wastewater, so that the gradual cracking of aromatic hydrocarbons and long chain alkanes into short chain small molecule structures, converting them into small molecule chemical substances that can be recycled, and then related separation and recovery can be carried out. The degradation results show that the reduction-oxidation coupling mechanism designed in this work maintains a high treatment effect in the degradation of coking wastewater, can effectively deal with the complex matrix interference in the actual environment, and has a great application prospect in the actual treatment. mechanism is ensured.

We hope these explanations can help you understand the relevant questions, and thank you again for your professional comments.

Fig. R4 corresponds to Figure 4g; Fig. R7 corresponds to Figure 4.

References:

1. Xie, Z.H. *et al.* Effects of Molecular Structure on Organic Contaminants' Degradation Efficiency and Dominant ROS in the Advanced Oxidation Process with Multiple ROS. *Environ. Sci. Technol.* **56**, 8784-8795, (2022).
2. Liu, X. *et al.* Selective Removal of Organic Pollutants in Groundwater and Surface Water by Persulfate-Assisted Advanced Oxidation: The Role of Electron-Donating Capacity. *Environ. Sci. Technol.* **57**, 13710-13720, (2023).
3. Wei, K. *et al.* A controllable reduction-oxidation coupling process for chloronitrobenzenes remediation: From lab to field trial. *Water Res.* **218**, 118453, (2022).
4. Jie, M. *et al.* Single-Atom MnN₅ Catalytic Sites Enable Efficient Peroxymonosulfate Activation by Forming Highly Reactive Mn(IV)-O_{xo} Species. *Environ. Sci. Technol.* 4266-4275, (2023).

5. Wu, S. *et al.* Efficient Electrochemical Hydrogenation of Nitroaromatics into Arylamines on a CuCo_2O_4 Spinel Cathode in an Alkaline Electrolyte. *ACS Catal.* **12**, 58-65, (2022).
6. Xuejie Zhang *et al.* Mineralization of tribromophenol under anoxic/oxic conditions in the presence of copper(II) doped green rust: Importance of sequential reduction-oxidation process. *Water Res.* **222** 118959, (2022).
7. Li, C. *et al.* Quantitative structure-activity relationship models for predicting reaction rate constants of organic contaminants with hydrated electrons and their mechanistic pathways. *Water Res.* **151**, 468-477, (2019).
8. Li, Y. *et al.* Application potential of zero-valent aluminum in nitrophenols wastewater decontamination: Enhanced reactivity, electron selectivity and anti-passivation capability. *J. Hazard. Mater.* **452**, 131313, (2023).
9. Xie, X. *et al.* Oxidation of Roxarsone Coupled with Sorptive Removal of the Inorganic Arsenic Released by Iron–Carbon (Fe–C) Microelectrolysis. *ACS ES&T Eng.* **1**, 1298-1310, (2021).
10. Zhou, X. *et al.* Nonradical oxidation processes in PMS-based heterogeneous catalytic system: Generation, identification, oxidation characteristics, challenges response and application prospects. *Chem. Eng. J.* **410**, 128312, (2021).
11. Cao, W. *et al.* l-Ascorbic acid oxygen-induced micro-electronic fields over metal-free polyimide for peroxymonosulfate activation to realize efficient multi-pathway destruction of contaminants. *J. Mater. Chem. A.* **8**, 810-819, (2020).
12. Zhou, Q. *et al.* Generating dual-active species by triple-atom sites through peroxymonosulfate activation for treating micropollutants in complex water. *Proc Natl Acad Sci U S A.* **120**, e2300085120, (2023).

General comments from Reviewer 4:

The authors describe a new system for the reduction-oxidation coupling of nitroaromatic pollutants. They present the design and synthesis of a photocatalyst to accomplish such a goal, including an extensive characterization of the synthesized materials and their application on a large-scale system under flow. This piece of work is well conducted, but it also shows several points that should be addressed before reaching quality for publication in this journal:

Response: We sincerely appreciate you for your time in reviewing the manuscript and giving your valuable comments. We have thoroughly considered your precious comments and suggestions and carefully addressed every one of them as best as we can. Revisions in the manuscript were highlighted in red. Here, we provide a point-to-point response to your major and specific comments below.

Major points:

Comment 1. In the introduction section, the authors discuss the inefficiency of reported approaches to deal with the removal of nitroaromatic compounds. Then, in lines 62-63: “Therefore, it is urgent to develop a controlled coupling of reduction and oxidation processes in one system with high resistance to environmental interference.” The authors should clarify the novelty of this approach or enlarge the discussion on the advantages of it in previous articles.

Response: Thank you for the professional suggestions of the reviewers. Nitroaromatic compounds (NACs) — organic molecules with one or more nitro-groups (-NO₂) connected to the benzene ring — are widely used as the most important organic chemical raw materials in the production of insecticides, pharmaceuticals, explosives, and dyes. Including nitrobenzene (NB), 4-nitrophenol (PNP), 2,4-dinitrobenzene (2,4-DNT), 2,4,6-trinitrotoluene (TNT), etc. At the same time, NACs are also persistent environmental pollutants with highly toxic, mutagenic, and carcinogenic characteristics. For example, they can oxidize heme to ferricheme, causing insufficient blood oxygen supply, leading to symptoms of central nervous system excitement such as dizziness, tinnitus, numbness of fingers, and general fatigue. In general, the more nitro groups, the greater the toxicity. Besides, the number of NACs that have been generated by various chemical industries is approximately 65,000, and 9000 metric tons of NB are entered into the water sources annually, which has been listed as a "priority pollutant" by the United States Environmental Protection Agency. Therefore, NACs pose serious threats to human health and natural ecosystems. According to the suggestions of viewers, we have made the following changes (marked yellow) in the introduction:

Nitroaromatic compounds (NACs) are vital raw materials or intermediates in industrial activities, which have been versatily applied for dye, pesticide, plastic, and medicine manufacturing, with approximately 65,000 kinds of high-concentration NACs being discovered in industrial wastewater. The presence of the strong electron-withdrawing nitro groups (-NO₂) induces the overall delocalization effect of π electrons in the benzene ring structure, thus enhancing the stability of the benzene ring and making NACs difficult to oxidize. Meanwhile, NACs are refractory pollutants with the feature of carcinogenic, teratogenic, and mutagenic that pose threats to environmental safety and human health, which have been listed as priority pollutants by the United States Environmental Protection Agency. Therefore, NAC-contained industrial wastewater has the characteristics of high salinity, high concentration, complex composition, high toxicity, and poor biodegradability. Its remediation is still a difficult task. Physical adsorption or extraction results in secondary pollution and biodegradation takes a longer time and is sensitive to the impact load of NACs and environmental changes. In a more environmentally friendly approach, chemical oxidation can degrade NACs, but the direct oxidation and complete mineralization of parent NACs is kinetically limited by the unique -NO₂ and the water quality conditions, leading to incomplete mineralization with low efficiency and high cost. Therefore, developing efficient approaches suitable for practical application is a great challenge.

For the first challenge, recent studies have reported that the reduction of -NO₂ to aniline groups (-NH₂) via reductants can weaken the key structural units of -NO₂ and decrease the chemical stability of NACs, thereby breaking the barriers in benzene ring-opening. Nevertheless, a single reduction process could not remove NACs safely from an environmental perspective for there are possibilities to form more toxic intermediates (e.g., arylamines and azo/oxygen compounds) than parent NACs and a reduction process that gives electrons difficult to further degrade and mineralize the reduction products, resulting in incomplete degradation and bringing greater environmental risks. As a result, both single reduction or oxidation methods can not simultaneously implement the efficient degradation of reduction products and complete mineralization of parent NACs. Therefore, constructing reduction-oxidative coupling degradation processes are effective strategy to solve the first challenge. Generally, advanced oxidation processes (AOPs) make it possible for the simultaneous occurrence of reduction and oxidation reactions by producing highly active reductive species (electron and H^{*}, etc.) and oxidative species (hole and •OH, etc.) at the same time. For example, zero-valence metal-Fenton-like technology and electrochemical technology adopt two-step reactions with separated reduction and

oxidation systems, which are difficult to operate and sensitive to pH. In comparison, a single catalytic system with the advantages of being easy to operate and low cost, however, the mutual consumption of reductive/ oxidative species needs to be solved (e.g. photocatalysis electron-hole pairs). For another challenge, the current technologies are almost $\bullet\text{OH}$ -based oxidation with higher oxidation ability (9 ~ 2.7 V vs NHE) but indiscriminateness, thus are easily quenched by water matrix in industrial wastewater, resulting in ineffective NACs removal rate and limiting the large-scale application of AOPs. Currently, non-radical oxidation processes with relatively weaker oxidation ability (e.g. $^1\text{O}_2$, 0.81 V vsNHE) have been extensively researched due to their specificity to electron-rich groups and higher resistance to environmental interference. Therefore, one promising strategy for solving the second challenge is to make use of the complementary advantages of radicals and non-radicals. To sum up, it is urgent to develop a coupling of reduction and oxidation processes in one system and synergistically generate radical and non-radical for efficiently removing NACs in industrial wastewater.

We hope these explanations can help you understand the relevant questions, and thank you again for your professional comments.

Comment 2. In the section: Overview of fixed reaction bed (FRB) reactor, the authors do not discuss the ratio LFCO vs CFC. It is hard to imagine that only one trial has been done, but still, in that case, a discussion on the ratio should be included.

Response: Thank you very much for your question. We are sorry for the confusion caused by the vague description. We have discussed the ratio of LFCO vs CFC in the manuscript before. For a clearer representation, we have made the following modifications:

1. Preparation of different weight ratios of LFCO vs CFC. By changing the total amount of chemicals (including citric acid and metal nitrate) in the mixed solution, CFC with different loading amounts of LFCO were obtained. The weight of LFCO is determined by the difference of CFC and LFCO@LFC. Unless indicated, LFCO@LFC in the manuscript refers to the sample with 1.5 g LFCO.
2. Fig. R1c (left) shows that when the weight of the LFCO increased to 1.5 g, the highest r'' was observed, this may be due to the increase of active sites to activate PMS. However, the r'' cannot be significantly improved by increasing the weight of the LFCO.
3. We have modified the relative figure.

Fig. R1 Removal of ONP under different operational parameters. A, B The removal efficiency and K_{obs} of ONP were investigated by changing water flux (A) and hydraulic retention time (B). **C** The removal efficiency and r'' of ONP were investigated for varying the dosage of catalyst, the dosage of PMS, and visible light intensity in the FRB system. **D** The degradation efficiency of ONP in different systems. **E** Long-term operation performance of the LFCO@CFC FRB system (Illustration is the photograph of the used LFCO@CFC). Optimal FBR conditions (unless indicated otherwise): 365 nm radiation, Water flux 7500 mL h⁻¹, HRT 60 s, 298 K, initial pH 7.1, 30 ppm ONP, and 3 mM PMS (introduced with the ONP stock solution).

We hope these explanations can help you understand the relevant questions, and thank you again for

your professional comments.

Fig. R1 corresponds to Figure 3.

Comment 3. Still, in this section, the authors refer to control experiments (Figures S3-S4) in which the amount of PMS and LFCO differ from those reported in section FBR system operation; see below.

Response: Thank you very much for your question. Our system goes through three stages: pre-experiment stage, pilot-scale stage, and scale-up application, which are three parts that are very closely related to each other. The reaction of all three is the same reaction, which means that their reaction principle is the same. But in the subtle operation, the three are always more or less different. 3 mM is the optimal concentration of PMS in the FBR system. Besides, to dynamically detect the change of pollutants during the reaction process and in pre-experiments, we also conducted beaker experiments using LFCO powder. 0.3 mM is the concentration of PMS in pre-experiments to simulate the continuous flow reaction process for exploring the maximum doping ratio of Cu (Figure S3) and reaction conditions (Figure S4). We are sorry for the confusion caused by the vague description. We have made changes based on your suggestions (see comments 8' response below).

We hope these explanations can help you understand the relevant questions, and thank you again for your professional comments.

Comment 4. In the first two sections of the results, the authors describe the synthesis and characterization of a bunch of materials with different ratios Cu/Fe (LFCO). In the discussion, they should include references to the same materials or claim their novelty.

Response: Thank you very much for your question. We are sorry for our mistakes in writing “LFCO” as “LCFO” in our manuscript and supplementary information. We have uniformly revised as “LFCO”.

We hope these explanations can help you understand the relevant questions, and thank you again for your professional comments.

Comment 5. Section: FBR system operation. It is hard to imagine the FRB system, even with Figure 5, since the photograph has no scale. Is the explanation in the supplementary text 4 the one that applies? It could be worth better clarifying in the text when the authors refer to the results of a batch experiment or the FBR system.

Response: Thank you very much for your question. We are sorry for the confusion caused by the vague description. According to the suggestions of the reviewer, we have made the following changes in the revised version:

1. Scales were added in the photograph.

Fig. R2 Application in water treatment. **A** Toxicity indicators of parent ONP and its degradation products. **B**, **C** Effects of different inorganic anions(**B**) and water quality (**C**) to the degradation efficiency. **D** Degradation of different nitro contaminants in LFCO@CFC FBR system. Optimal FBR conditions (unless indicated otherwise): 365 nm radiation, Water flux 7500 mL h⁻¹, HRT 60 s, 298 K, initial pH 7.1, 30 ppm ONP, and 3 mM PMS (introduced with the ONP stock solution). **E** Diagram of the large-scale application plant including 5 main parts: (1) wastewater collection (2) PMS dosing tank, (3) physical filtration, (4) FBR reactor, (5) effluent tank (data from 1:50 for equal scale amplification). The dimensions of core parts 2 and 4 are marked in the enlarged image on the right. **F** Performance and cost comparison of reported catalysts.

2. Supplementary Text 6. Design and cost of large-scale application.

We chose FRB practical application due to it can increase the contact area between pollutants and catalysts and improve the degradation rate. At the same time, the catalyst can be ensured to stay relatively stable in the reaction bed layer, which is not easy to lose and damage, and the service life of the FBR can reach 8-10 years, which can be effectively applied to the wastewater containing special

pollutants. 1:50 equal-scale amplification was used to achieve real large-scale applications with industrial significance. Specific design parameters are as follows: For part 2: the outer contour of the container was cylindrical with a diameter of 8 m and a height of 4.5 m. For part 4: the outer contour of the reactor was cylindrical with a diameter of 12.5 m and a height of 4.5 m, the diameter and height of the center light source are 4 m and 4 m, respectively. The 6 cylindrical FBRs are uniformly distributed outside the light source with a diameter of 4 m and a height of 4 m. The amount of catalyst covered in each packed reactor is 2 kg, which is replaced every month. We set the six reactors to rotate at a constant speed to allow sufficient light to further increase the reaction rate.

We hope these explanations can help you understand the relevant questions, and thank you again for your professional comments.

Fig. R2 corresponds to Figure 5.

Comment 6. Lines 163-164: “In Figure 3a, when the water flux decreased from 15000 mL/h to 1500 mL/h, the pollutant removal rate decreased from 99.123% to 69.665%”. It seems to be the opposite of the graph. Similar to residence time, please verify.

Response: Thank you very much for your question. We are sorry for our mistakes. After careful checking, we agree with the reviewer's opinion. With the sharp increase of water flux from 1500ml /h to 15000ml /h, the removal effect of pollutants will be greatly decreased from 99.123% to 69.665%, especially when the water flux is higher than 7500 ml/h. Therefore, the optimal water flux parameter proposed for the FBR system is 7500 ml/h. According to the suggestions of the reviewer, we have made the following changes in the revised version:

The increase in water flux indicates that the molar flux of pollutants increases, representing more ONP can be brought to the reactor. As shown in Figure 3a, the removal efficiency and the apparent rate constant (k_{obs}) followed a similar trend when increasing the water flux. The removal efficiency of ONP was 99.123% at a low flux value (1500 mLh^{-1}). When the water flux increased from 1500 mLh^{-1} to 7500 mLh^{-1} , the removal efficiency was relatively stable. However, with the further increase of water flux to 15000 mLh^{-1} , removal efficiency sharply decreased to 69.665%. Therefore, the optimal water flux parameter proposed for FBR system is 7500 mLh^{-1} . The hydraulic residence time (HRT) is another limiting factor of the FBR system, which greatly affects the mass transfer rate. In Figure 3b, the removal efficiency and the apparent rate constant (k_{obs}) followed the opposite trend when increasing the HRT. Though higher k_{obs} can be achieved at extremely short HRT ($0 \sim 50 \text{ s}$), the removal

efficiencies are low due to the limited pollutant diffusion. This results in substandard effluent quality. The prolonged HRT increases the interaction time between ROS and ONP, thus a higher ONP removal efficiency (more than 99%) was achieved within a residence time of 60 ~ 90 s. On this basis, the hydraulic retention time is set to 60 seconds with relatively high k_{obs} . To sum up, the water flux and HRT were set to 7500 mLh^{-1} and 60 s, respectively, to optimize the degradation effect and reduce the cost of this process.

We hope these explanations can help you understand the relevant questions, and thank you again for your professional comments.

Comment 7. Line 177 (The dosage of catalyst and PMS, and light intensity). Please check the amount of catalyst commented on in the text and the one shown in Figure 3c. Moreover, catalyst refers to LFCO or LFCO@CFC?

Response: Thank you very much for your question. “Dosage of catalyst” refers to the weight of LFCO fixed in CFC. According to the suggestions of reviewer, we have made the following changes in the revised version:

1. Actual degradation rate (r'') was defined taking the mass transfer into account and used as the indicator here. Firstly, by changing the amount of chemicals, CFCs loaded with different amounts of LFCO were prepared. Fig. 3c (left) shows that when the weight of the LFCO increased to 1.5 g, the highest r'' was observed, this may be due to the increase of active sites to activate PMS. However, the r'' cannot be significantly improved by increasing the weight of the LFCO. Similarly, the r'' increased gradually by increasing the PMS concentration from 1 mM to 3 mM, causing more ROS can be generated by PMS activation. Further increasing the PMS concentration to 4 mM did not enhance the r'' due to the saturation of active sites. Therefore, LFCO@CFC-1.5 and 3 mM PMS are selected for the FBR system. Light intensity defines the number of photons available to the photocatalyst and determines the number of produced electron-hole pairs and reactive species[1]. Initially, the r'' increased with the increase of light intensity from 0 W to 15 W and then remained unchanged above 15 W. In this case, the light intensity is optimized at 15 W from the perspective of energy and cost saving.

2. Preparation of different weight ratios of LFCO vs CFC. By changing the total amount of chemicals (including citric acid and metal nitrate) in the mixed solution, CFC with different loading amounts of LFCO were obtained. The weight of LFCO is determined by the difference of CFC and LFCO@LFC.

Unless indicated, LFCO@LFC in the manuscript refers to the sample with 1.5 g LFCO.

3. Powder $\text{LaFe}_{1-x}\text{Cu}_x\text{O}_{3-\sigma}$ was synthesized by the same method without adding CFC. The catalysts were named LFO-Cu_{0.01}, LFO-Cu_{0.03}, LFO-Cu_{0.05}, and LFO-Cu_{0.10} according to the doping concentration of Cu.

We hope these explanations can help you understand the relevant questions, and thank you again for your professional comments.

Reference:

1. Lotfi, S. *et al.* Photocatalytic degradation of steroid hormone micropollutants by TiO₂-coated polyethersulfone membranes in a continuous flow-through process. *Nat. Nanotechnol.* **17**, 417-423, (2022).

Comment 8. With respect to the PMS concentration, the authors comment on the optimum 3 mM=92 mg/L); however, control experiments were performed with 0.3 mg/L of PMS (see Figure S3). How do we compare the two experiments? Figure S15 corresponds to LFCO or LFCO@CFC? Figure S16: it looks pretty similar to S3 and S4. Moreover, in Figure S3: is LFCO soluble, or is it dispersed? Is the pH adjusted, or is the pH resulting directly from the mixture?

Response: Thank you for the professional question of the reviewer. Our system goes through three stages: pre-experiment stage, pilot-scale stage, and scale-up application, which are three parts that are very closely related to each other. The reaction of all three is the same reaction, which means that their reaction principle is the same. But in the subtle operation, the three are always more or less different: (1) Pre-experiment in the beaker. In this stage, LFCO powder was used to develop and optimize methods, determine the best process route, have simple equipment and technical conditions, process flow, abundant and inexpensive chemical material sources, and the best degradation efficiency. (2) Pilot-scale stage. Subsequently, we conducted a pilot scale testing using LFCO@CFC as an FBR reactor, which is a necessary transitional step from small-scale experiments to industrial applications. Based on small-scale experimental research on feasible industrialization plans, we further investigate whether the process is mature and reasonable in a certain scale of scale-up devices, and solve problems that cannot be solved or discovered in the laboratory, providing a design basis for industrial applications. Although the essence of chemical reactions does not change with different experimental production, the optimal parameters for reactions may vary with external conditions such as experimental scale and equipment. For example, when the reaction is amplified, the issue of mass

transfer is prominently exposed because of the increase in reaction space. Therefore, pilot scale-up is an important transitional stage from rapid and high-level to industrial application, and its level represents the level of industrialization. (3) Large-scale application. Finally, 1:50 equal-scale amplification was used to achieve the real industrial significance of the economic scale of large-scale applications.

According to the above descriptions, 3 mM is the optimal concentration of PMS in the FBR system, while 0.3 mg/L is the concentration of PMS in pre-experiments for exploring the maximum doping ratio of Cu (Figure S3) and reaction conditions (Figure S4). Besides, to dynamically detect the change of pollutants during the reaction process, we also conducted beaker experiments using LFCO powder, with 3 mM PMS to simulate the continuous flow reaction process. We are sorry for the confusion caused by the vague description. We have made the following modifications to the manuscript and supplementary information according to the reviewer's suggestion:

1. More details have been added in the experiment part

FBR Degradation experiments.

Experiments were performed in flow-through (dead-end) mode. Under light irradiation at room temperature (298 K), the thoroughly mixed ONP and PMS solution flowed through LFCO@CFC FRB by a peristaltic pump (30W) with no feed recirculation. The key system elements are as described below: The as-prepared LFCO@CFC (9 cm²) was inserted in a quartz glass tube to allow the transmission of light. A 350W Xenon lamp (CEL-HX F300, Beijing, China) with a 420 nm and 780 nm cutoff filter was used as the light source and the light intensity is 560 mW cm⁻². The system was operated at different water fluxes and hydraulic residence times by controlling the pump flow rate. Unless indicated, 7500 mL h⁻¹ flux, HRT 60 s, 298 K, initial pH 7.1, 30 ppm ONP, and 3 mM PMS (introduced with the ONP stock solution) were chosen as optimal conditions for all experiments. All samples were collected and analyzed by HPLC to calculate the ONP removal efficiency according to Eq. (1):

$$\text{Removal \%} = \frac{C_t}{C_0} \quad (1)$$

where C_t is the concentration of ONP at a certain reaction time (t) and C_0 refers to the initial concentration after adsorption equilibrium.

Ideally, the relationship between $\ln(C/C_0)$ and reaction time was matched well with the integral rate

equation of the first-order reaction. Therefore, K_{obs} can be expressed by the pseudo-first-order kinetic model as described below:

$$\ln\left(\frac{C_t}{C_0}\right) = -K_{\text{obs}}t \quad (2)$$

The actual degradation rate ($\text{m}^{-3} \text{s}^{-1}$) in a heterogeneous reaction can be calculated based on Eq. (3):

$$-r_{\text{ONP}}'' = -\frac{1}{A \times V} \times \frac{dN_{\text{ONP}}}{dt} = \frac{\text{ONP}_{\text{reacted}}}{A \times V \times t} = \frac{1}{A \times V} \times Q_f \times c_f \times r_f \quad (3)$$

where A is the catalyst weight (g), V is the volume of carbon fiber felt, t is the reaction time (s), Q_f (L s^{-1}) is the feed flow rate, c_f (g L^{-1}) is the feed concentration of ONP, and r_f is the removal efficiency of ONP.

Beaker experiments

To facilitate the *in-situ* detection of reaction processes and intermediate products, we also used beaker experiments for ONP degradation. Unless indicated, 50 mL 30 ppm ONP solution including 0.3 mM PMS and 0.3 g L^{-1} powder catalyst under 560 mW cm^{-2} light irradiation at room temperature (298 K) were chosen as standard conditions for all experiments. Before the light irradiation and adding PMS, the solution was stirred in the dark for 30 minutes to reach adsorption-desorption equilibrium. During the reaction, 4 mL solution was extracted at intervals and immediately filtered through an organic membrane filter ($0.22 \mu\text{m}$) for further analysis. At the same time, the obtained solid catalysts after filtering were also further characterized.

2. Figure S15 corresponds to LFCO powder. We are sorry for this, and the following modifications have been made according to the reviewer's suggestion:

Fig. R3. **a** The HPLC profiles for the degradation of ONP in the LFCO/PMS/Nir system. **b** The ONP degradation efficiency in PMS/Nir system with different powder catalyst. Conditions: $\lambda > 780 \text{ nm}$, 50

mL 30 ppm ONP solution, 0.3 mM PMS, 0.3 g·L⁻¹ powder catalyst, 298 K, initial pH 7.1.

3. Figure S16 was detected in optimal conditions with 0.3 g/L catalyst powder, while Figures S3-4 were pre-experiments by using 0.4 g/L catalyst powder. We have made the following modifications according to the reviewer's suggestion:

a) **Supplementary Figure 3.** Pre-experiments in beaker: Degradation of ONP in LaFe_{1-x}Cu_xO₃ (x = 0, 0.01, 0.03, 0.05, 0.10) in PMS system under visible light irradiation ($\lambda > 420$ nm, 50 mL 30 ppm ONP solution including 0.3 mM PMS and 0.4 g L⁻¹ catalyst, initial pH = 7.1).

b) **Supplementary Figure 4.** Pre-experiments in beaker: Degradation of ONP with LaFe_{1-x}Cu_xO_{3- δ} (x = 0, 0.01, 0.03, 0.05, 0.10) in **a** visible light and **b** PMS system ($\lambda > 420$ nm, 50 mL 30 ppm ONP solution including 0.3 mM PMS and 0.4 g L⁻¹ catalyst, initial pH = 7.1).

c)

Figure R4. **a** The HPLC profiles for the degradation of ONP in LFO/PMS/Vis system. **b** The degradation of ONP of different systems with powder catalyst. Conditions: $\lambda > 420$ nm, 50 mL 30 ppm ONP solution, 0.3 mM PMS, 0.3 g·L⁻¹ powder catalyst, 298 K, initial pH 7.1.

4. About pH

In our experiments, the pH results were directly detected from the mixture. Therefore, we modified it as the initial pH.

We hope these explanations can help you understand the relevant questions, and thank you again for your professional comments.

Fig. R3 corresponds to Supplementary Figure 15; Fig. R4 corresponds to Supplementary Figure 16.

Comment 9. Lines 187-189: the authors should comment more deeply on the results observed under

Nir conditions. It looks like they only presented them as a curiosity. Could it be more economically favored to use Nir light?

Response: Thank you for the professional questions.

It is well-documented that the utilization of solar energy is pivotal for the photocatalytic efficiency. The general semiconductor photocatalysts can only harvest the UV or visible photons and remain the Nir light unutilized. However, the Nir light accounts 53% of the solar light[1]. Thus, the solar energy utilization for these semiconductor photocatalysts is not efficient and still far from expectations and industrial applications. Besides, Nir light has obvious advantages over UV/visible light owing to its higher tissue penetrability and lower photo-toxicity[2]. Therefore, developing strategies to make fully use of the residual Nir light is highly desirable. However, the full utilization of Nir light energy is challenging at present, because the lower energy of Nir photons relative to UV/visible photons and energy must be harvested from two (or more) Nir photons to produce the desired excited state[3]. Adopting upconversion materials/chromophores, inducing SPR effect or employing bandgap engineering represent efficient strategies to expand the absorption spectrum of photocatalysts into NIR light region to improve the solar energy utilization and conversion efficiency[4]. Generally, doping, vacancies, or multivalent element can help to generate intermediate energy levels by bandgap engineering, hence, they are widely used in modifying semiconductor photocatalysts to harvest wide solar light.

In our manuscript, Cu-doping can create intermediate state locating between the VB and CB of semiconductor to assist the LFCO photocatalysts to harvest NIR photons (Fig. R5). Intermediate state can modulate the electronic structures over photocatalysts. Induced mid-gap states could provide extra mobilized electrons for catalytic process, which required much smaller excitation energy compared with that of bandgap energy[5]. And those increased mobilized electrons could also contribute to the improvement of light response. In addition, the existence of mid-gap states could inhibit the recombination between photoexcited electrons and holes, thus prolonging the lifetimes of photoexcited charges. Therefore, LFCO has superior ONP degradation performance than LFO in Nir light.

Fig. R5 Schematic illustration of Nir absorption mechanisms in LFCO.

However, this is only an interesting phenomenon found in the experiment and is not the focus of the article, so it is not discussed in depth. In future studies, we may carry out full-spectrum catalysis based on this according to the inspiration of reviewers.

We hope these explanations can help you understand the relevant questions, and thank you again for your professional comments.

Fig. R5 corresponds to Figure 4a.

References:

1. Kosco, J. *et al.* Generation of long-lived charges in organic semiconductor heterojunction nanoparticles for efficient photocatalytic hydrogen evolution. *Nat. Energy* **7**, 340-351, (2022).
2. Zhang, Y. *et al.* H₂O₂ generation from O₂ and H₂O on a near-infrared absorbing porphyrin supramolecular photocatalyst. *Nat. Energy* **8**, 361-371, (2023).
3. Zhao, B. *et al.* Photocatalysis-mediated drug-free sustainable cancer therapy using nanocatalyst. *Nat. Commun.* **12**, 1345, (2021).
4. Wang, L. *et al.* Near-Infrared-Driven Photocatalysts: Design, Construction, and Applications. *Small* **17**, 1904107, (2021).

5. Li, Y. *et al.* Implementing Metal-to-Ligand Charge Transfer in Organic Semiconductor for Improved Visible-Near-Infrared Photocatalysis. *Adv. Mater.* **28**, 6959-6965, (2016).

Comment 10. Is the legend in Figure S33 correct? Figure S34 a and b: why the ONP spectrum at time 0 is not identical in both cases? Moreover, check the evolution of the peaks in Figure S34 a because it looks to this reviewer that the three bands decrease to the same extent. Also, check Figure 4f.

Response: Thank you for the professional questions.

1. In the Figure S35, we are sorry for our mistake to write the wrong legend. We have made the following modifications according to the reviewer's suggestion:

Fig. R5. Typical UV-vis absorption spectra of ONP and OAP.

2. In Figure S36 a and b (Figure S34 a and b in the old version), the detected characteristic peaks correspond to ONP. We have enhanced the color contrast of the lines to clearly show the evolution of the peaks. In order to confirm the formation of OAP, we determined the OAP concentration by HPLC and added in Figure 4f and Figure S36 c-d. Explanations are as follows:

Supplementary Fig. 36a and c shows that in the PMS system, the degradation rate of ONP is low and OAP is not generated. Supplementary Fig. 36b and d shows that under the photocatalytic condition, though the peak of ONP decreases slowly, the characteristic peaks of OAP increase first and then decrease, indicating that part of ONP is reduced to OAP during degradation. This also shows the dominance of photoreduction in the process of reducing ONP to OAP and reactive oxygen species are responsible for the subsequent ring-opening process of OAP.

Fig. R6 *In-situ* UV-vis absorption spectra of ONP degradation in the **a** LFCO/PMS and **b** LFCO/Vis system. HPLC spectra of generated OAP in the **c** LFCO/PMS and **d** LFCO/Vis system.

3. Interestingly, the absorption band at 210 nm slightly shifted to 230 nm with increased intensity within 10 minutes only in LFCO/PMS/Vis system, suggesting the generation of OAP at the initial stage (Fig. R7e and Fig. R6a-b). Meanwhile, the absorption band at 278 and 350 nm gradually disappeared, inferring that the aromatic ring was opened and mineralized as the reaction time increased. The variation of ONP and OAP concentration was further detected by HPLC (Fig. R7f and Fig. R6c-d). Upon the decline of the ONP concentration, the formation of OAP was observed immediately (within 5 min). Notably, negligible concentration of both ONP and OAP was achieved upon completion of the reaction, which indicates complete oxidative degradation.

Fig. R7 Reduction-oxidation coupling (ROC) mechanism and degradation pathway of ONP. A The band gap diagram of LFO and LFCO. **B** The XPS valence band spectra of LFO and LFCO. **C** Proposed mechanism for the PMS reaction on the surface of LFCO. **D** EPR spectra of DMPO- $\cdot\text{OH}$ /DMPO- $\text{SO}_4^{\cdot-}$ and $\text{TEMP}\cdot^1\text{O}_2$ in PMS/Vis, LFO/PMS/Vis, and LFCO/PMS/Vis system, respectively. **E** In-situ UV-Vis absorption spectra of ONP degradation in the LFCO/PMS/Vis system. **F** HPLC spectra and corresponding concentration changes of ONP and OAP during degradation in the LFCO/PMS/Vis system. **G** Charge density difference of condensed Fukui index (f^+) of ONP. **H** The degradation pathway of ONP in the LFCO/PMS/Vis system system. Conditions: $\lambda > 420$ nm, 50 mL 30 ppm ONP solution, 0.3 mM PMS, 0.3 g·L⁻¹ powder catalyst, 298 K, initial pH 7.1.

We hope these explanations can help you understand the relevant questions, and thank you again for your professional comments.

Fig. R5 corresponds to Supplementary Figure 35; Fig. R6 corresponds to Supplementary Figure 36; Fig. R7 corresponds to Fig. 4.

Comment 11. In the experiments reported in Figure S37, the experimental conditions, including concentrations of scavengers, are needed to extract conclusions safely.

Response: Thank you for the professional question of the reviewer. Understanding the species and concentrations of ROS is necessary for designing an AOP in practical engineering. However, due to the transient lifetimes (nanoseconds to microseconds) and low concentrations (generally $< 10^{-10}$ M) of typical ROS, common analytical techniques such as chromatography and spectroscopy can rarely directly identify and quantify them. Instead, indirect measurements using probes and quenchers are usually employed to quantify ROS concentrations and examine their role[1]. Quenchers are often applied to compete with pollutants in reacting with specific ROS. The role of ROS can be clarified by observing the extent of inhibition of micropollutant degradation. By monitoring the changes in the probe concentrations, the role and relative contributions of ROS can be quantified. According to the suggestions of the reviewer, we have made the following changes in the revised version:

1. Quenching experiments were performed to study the effects of ROS during ONP degradation with adding 0.05 mmol L^{-1} scavengers before the degradation process. Methanol (MeOH, $k_{\text{SO}_4^{\cdot-}/\text{MeOH}} = 3.2 \times 10^6 \text{ M}^{-1} \text{ s}^{-1}$, $k_{\cdot\text{OH}/\text{MeOH}} = 9.7 \times 10^8 \text{ M}^{-1} \text{ s}^{-1}$) can be used to quench both $\cdot\text{OH}$ and $\text{SO}_4^{\cdot-}$, while tert-butyl alcohol (TBA) has much higher rate constant with $\cdot\text{OH}$ ($3.8 \times 10^8 \sim 7.6 \times 10^8 \text{ L mol}^{-1} \text{ s}^{-1}$) than $\text{SO}_4^{\cdot-}$ ($4.0 \times 10^5 \sim 9.1 \times 10^5 \text{ L mol}^{-1} \text{ s}^{-1}$). Carotene, superoxide dismutase (SOD), and $\text{K}_2\text{Cr}_2\text{O}_7$ are easier react with $^1\text{O}_2$, $\cdot\text{O}_2^-$, and photogenerated electrons (e^-), respectively.
2. The experimental conditions are added.

Fig. R8 a Effects of different quenchers on the degradation of ONP in the LFCO/PMS/Vis system. **b** Contributions of different ROS. Conditions: $\lambda > 420 \text{ nm}$, 50 mL 30 ppm ONP solution, 0.3 mM PMS, $0.3 \text{ g}\cdot\text{L}^{-1}$ powder catalyst, 298 K , initial pH 7.1 . 0.05 mmol L^{-1} quenchers were added to the solution before light irradiation.

We hope these explanations can help you understand the relevant questions, and thank you again for your professional comments.

Fig. R8 corresponds to Supplementary Figure 39.

Reference:

1. Lei, Y. *et al.* Assessing the Use of Probes and Quenchers for Understanding the Reactive Species in Advanced Oxidation Processes. *Environ. Sci. Technol.* **57**, 5433-5444, (2023).

Comment 12. Lines 326-328 and Tables S11 and S12: the references where the numbers are extracted from should be included.

Response: Thank you for the professional suggestions of the reviewer. According to the suggestions of the reviewer, we have made the following changes in the revised version:

Table R1. Performance and cost comparison of recently reported photocatalysts

Catalyst	k (min ⁻¹)	Cost (CNY)	Ref.
B _{0.05} -C ₃ N ₄	0.0213	44.57	37
Fe ₃ O ₄ /TiO ₂ /CuO	0.0206	35.27	38
Ce-PDMS-PbO ₂ /SS	0.0270	30.89	39
PdNCs/CoAl(O)/rGO	0.0377	27.54	40
Co-SrTiO ₃	0.0487	19.67	41
BiC-0.05	0.0139	51.72	42
LaFe _{0.95} Cu _{0.05} O ₃	0.0790	13.72	This work
La-SrTiO ₃	0.0412	21.19	42
Co-Bi ₂ O ₂ CO ₃	0.0349	34.63	43
Bi ₂ O ₃ /Bi ₂ O ₂ CO ₃	0.0213	42.91	44
LaFeO ₃	0.0115	47.39	This work
Fe-CN	0.0642	24.66	45
La _{0.08} MnO _{3-δ}	0.0813	16.89	46
La ₂ CuO ₄	0.0198	54.81	47
Zn _x Co-ZIFs	0.0447	41.23	48

37. Zhan, H. *et al.* Photocatalytic O₂ activation and reactive oxygen species evolution by surface B-N bond for organic pollutants degradation. *Appl Catal. B: Environ.* **310**, 121329, (2022).

38. Kianfar, A. H.; Arayesh, M. A. Synthesis, characterization and investigation of photocatalytic and catalytic applications of Fe₃O₄/TiO₂/CuO nanoparticles for degradation of MB and reduction of nitrophenols. *J. Environ. Chem. Eng.* **8**, 103640, (2020).
39. Li, H. *et al.* Ultrasonic-electrodeposition construction of high hydrophobic Ce-PDMS-PbO₂/SS electrode for p-nitrophenol degradation: Catalytic, kinetics and mechanism. *Appl Catal. B: Environ.* **335**, 122884, (2023).
40. Zhou, Q. *et al.* Novel hierarchical carbon quantum dots-decorated BiOCl nanosheet/carbonized eggshell membrane composites for improved removal of organic contaminants from water via synergistic adsorption and photocatalysis. *Chem. Eng. J.* **420**, 129582, (2021).
41. Wang, Q. *et al.* Hierarchical-Structured Pd Nanoclusters Catalysts x-PdNCs/CoAl(O)/rGO-T by the Captopril-Capped Pd Cluster Precursor Method for the Highly Efficient 4-Nitrophenol Reduction. *ACS Appl. Mater. Interfaces*, **14**, 27775-27790, (2022).
42. Zhang, D. *et al.* Dynamic active-site induced by host-guest interactions boost the Fenton-like reaction for organic wastewater treatment. *Nat. Commun.* **14**, 3538, (2023).
43. Zhao, D.-X.; Lu, G.-P.; Cai, C. Efficient visible-light-driven Suzuki coupling reaction over Co-doped BiOCl/Ce-doped Bi₂O₂CO₃ composites. *Green Chem.* **23**, 1823-1833, (2021).
44. Huang, Y. *et al.* Visible light Bi₂S₃/Bi₂O₃/Bi₂O₂CO₃ photocatalyst for effective degradation of organic pollutions. *Appl Catal. B: Environ.* **185**, 68-76, (2016).
45. Liu, X. *et al.* In Situ Modulation of A-Site Vacancies in LaMnO_{3.15} Perovskite for Surface Lattice Oxygen Activation and Boosted Redox Reactions. *Angew. Chem. Int. Ed.* **60**, 26747-26754, (2021).
46. Chen, H. *et al.* Understanding oxygen-deficient La₂CuO_{4.8} perovskite activated peroxymonosulfate for bisphenol A degradation: The role of localized electron within oxygen vacancy. *Appl Catal. B: Environ.* **284**, 119732, (2021).
47. Chen, Z. *et al.* Single-atom Mo-Co catalyst with low biotoxicity for sustainable degradation of high-ionization-potential organic pollutants. *Proc. Natl. Acad. Sci.* **120**, e2305933120, (2023).
48. Tang, Y. *et al.* Engineering magnetic N-doped porous carbon with super- high ciprofloxacin adsorption capacity and wide pH adaptability. *J. Hazard. Mater.* **388**, 122059, (2020).

We hope these explanations can help you understand the relevant questions, and thank you again for your professional comments.

Table R1 corresponding to Supplementary Table 11.

Comment 13. How are the values extracted from graph 4b compared to those from graph S18?

Response: Thank you for the professional suggestions. The band gap energy of a semiconductor describes the energy needed to excite an electron from the valence band to the conduction band. An accurate determination of the band gap energy is crucial in predicting photophysical and photochemical properties of semiconductors[1]. In particular, this parameter is referred to when photocatalytic properties of semiconductors are discussed, and they are often tested by UV-vis spectroscopy, which is a relatively simple and fast method to enable the absorption behavior of the material to be examined in a short time and thus promises fast access to the bandgap of solid materials compared to other analysis techniques. In 1966 Tauc proposed a method of estimating the band gap energy of amorphous semiconductors using optical absorption spectra[2]. The Tauc method is based on the assumption that the energy-dependent absorption coefficient α can be expressed by the following equation:

$$(\alpha \times hv)^\gamma = B(hv - E_g) \quad (1)$$

where h is the Planck constant, ν is the photon's frequency, E_g is the band gap energy, and B is a constant. The γ factor depends on the nature of the electron transition and is equal to 1/2 or 2 for the indirect and direct transition band gaps, respectively.

According to the theory of P. Kubelka and F. Munk presented in 1931, the measured reflectance spectra can be transformed to the corresponding absorption spectra by applying the Kubelka-Munk function ($F(R_\infty)$, Eq. 2):

$$F(R_\infty) = \frac{K}{S} = \frac{(1 - R_\infty)^2}{2R_\infty} \quad (2)$$

Where $R_\infty = \frac{R_{\text{sample}}}{R_{\text{stand}}}$ is the reflectance of an infinitely thick specimen, while K and S are the absorption and scattering coefficients, respectively. Putting $F(R_\infty)$ instead of α into Eq. (1) yields the form Eq. (3):

$$(F(R_\infty) \times hv)^\gamma = B(hv - E_g) \quad (3)$$

Figure R9b shows the Tauc plots of LFO and LFCO (direct band gap semiconductor) transformed according to eq 1 plotted against the photon energy. The region showing a steep, linear increase of light absorption with increasing energy is characteristic of semiconductor materials. The x-axis

intersection point of the linear fit of the Tauc plot gives an estimate of the band gap energy. However, defective, doped, bulk, or surface-modified materials, may introduce intraband gap states. It reflects in the absorption spectrum as an Urbach tail, i.e., an additional absorption band. Also, Its presence influences the Tauc plot and therefore must be taken into account to determine the band gap energy. Therefore, a new bandgap of 1.68 eV we identified is rational, and it corresponds to the new charge-transfer transitions opened up by introducing Cu atoms.

Fig. R9 a UV-Vis diffuse reflectance spectra and **b** corresponding Tauc plots of LFO and LFCO.

Also, many other literature have offered similar explanations. A recent work proved the clear sub-states in PCN and APCN by K-M plot[3]. Also, the presence of humps in the absorption spectra in NPCN, in contrast to the smooth absorption spectra of PCN indicates induced mid-gap states. The transition energy (E_t) from the valence band to the mid-gap states of the NPCN catalyst is calculated to be 2.47 eV from the Kubelka-Munk plot (Fig. R10a). The experimental and theoretical simulations suggest that mid-gap states in NPCN catalysts are the main platform for charge-carrier separation that enhances the overall photocatalytic performance. Utilizing NPCN with a mid-gap state as a photocatalyst presented NADH photo-oxidation efficiency of over 98% and a high hydrogen production rate of $11.18 \text{ mmol g}^{-1} \text{ h}^{-1}$ with an apparent quantum yield of 9.16% (420 nm), outperforming other state-of-art metal-free photocatalysts. For another example, Shaohua Shen believed that a step-like absorption tail extending to 900 nm appears in the UV-visible DRS of the BDCNN, indicating that dopant and defect-related midgap states (MS) are generated in the forbidden band of the BDCNN, which together contribute to their redshift in the optical absorption edge[4]. As

determined from the transformed Kubelka–Munk function in Fig. R10b, the intrinsic bandgaps of the CNN and BDCNN were calculated to be 2.72 and 2.37 eV, respectively. Moreover, a bandgap of 1.80 eV was identified for the BDCNN, corresponding to the electronic transition between the VB and MS, which effectively facilitates charge separation and transfer. The obtained BDCNN achieves stoichiometric H₂ and O₂ evolution in the presence of Pt and Co(OH)₂ co-catalysts, and the solar-to-hydrogen efficiency reaches 1.16% under one-sun illumination.

[REDACTED]

Fig. R10 a Kubelka-Munk (K-M) plot of as-prepared catalysts[3]. **b** Tauc plots for CNN and BDCNN using $(F(R)hv)^{1/2}$ (K-M parameter) as a function versus the photon energy[4].

According to the reviewer's suggestion, we have added the method in the captions of Fig. 4 in the revised manuscript:

Tauc plots for LFO and LFCO using $(F(R)hv)^2$ (Kubelka-Munk parameter) as a function versus the photon energy (The x-axis intersection point of the linear fit of the Tauc plot gives an estimate of the band gap energy).

We hope these explanations can help you understand the relevant questions, and thank you again for your professional comments.

Fig. R9a corresponds to Supplementary Figure 19; Fig. R9b corresponds to Figure 4b.

References:

1. Makuła, P. *et al.* How To Correctly Determine the Band Gap Energy of Modified Semiconductor Photocatalysts Based on UV–Vis Spectra. *J. Phys. Chem. Lett.* **9**, 6814–6817, (2018).
2. Tauc, J. *et al.* Optical properties and electronic structure of amorphous germanium. *Phys. Stat. Sol.* **15**, 627–637, (1966).

3. Bhojar, T. *et al.* Accelerating NADH oxidation and hydrogen production with mid-gap states of nitrogen-rich carbon nitride photocatalyst. *iScience* **25**, 105567, (2022).
4. Zhao, D. *et al.* Boron-doped nitrogen-deficient carbon nitride-based Z-scheme heterostructures for photocatalytic overall water splitting. *Nat. Energy* **6**, 388-397, (2021).

Comment 14. This reviewer expected an overall mechanism of the reaction at the end of the discussion. The scheme shown in Fig 4h starts with the expected reduction of ONP, which needs two H⁺. What about the pH of the solution is 7.1 always? what about the evolution of the pH with time? We can assume that the reduction starts in the LFCO, and then the ROS from PMS play the oxidation role. However, is it reasonable to eliminate an Ar-NH₂ to give an Ar-H by the action of ¹O₂? Technically speaking, this process is a reduction.

Response: Thank you for the professional suggestions.

1. We have specified the initial pH of the solution as 7.1 in the revised version. Also, the evolution of the pH with time was supplemented. Of note is the slight decrease occurred at solution pH during the reaction as shown in Fig.R11. The pH of the reaction solution changes from neutral to weakly acidic during the reduction stage (0 ~ 10 min), which is conducive to providing H⁺. It returns to neutral as the reaction proceeds.

Fig. R11 The changes of pH during reaction.

One of the reasons for the phenomenon may be that the PMS oxidization process could contribute to the concentration of H^+ and 1O_2 according to Eq. (3), while PMS reduction at the Fe site can generate SO_4^{2-} and $\bullet OH$ [1]. We further proposed the microscopic mechanism by which the LFCO catalyst showed excellent reduction-oxidation coupling degradation performance for ONP: First, Cu doping is conducive to the efficient generation and migration of photogenic electrons and holes. Then, the activation of PMS adsorbed at the Cu/Fe site generated ROS and H^+ at the same time, thus forming a local acid-like reaction environment in a neutral solution. This local acidic environment provides a good reaction condition for the reduction of nitro, to achieve the ordered reduction-oxidation coupling process. This is a question that we have neglected before. In the following work, according to the inspiration of reviewers, we will further explore whether the catalyst surface forms a “local acidic environment”, to accelerate the coupled reduction-oxidation degradation process.

2. eliminate an Ar-NH₂ to give an Ar-H

According to the reviewer's question, we found that the $\bullet OH$ of the PMS reduction product was incorrectly written as $SO_4^{\bullet -}$, which was corrected in the revised version. The reviewer may wonder how P4 becomes P5. Therefore, we described the degradation path more specifically as follows:

It is worth noting that the N12 atom ($f^+ = 0.1121$) with a high Fukui index is the most reactive site and easy to be attacked by e^- . The rapid reduction of the $-NO_2$ to $-NH_2$ groups to form OAP (P3) requires electron transfer ($6 e^-$) coupled with hydrogenation (proton from PMS oxidization, Eq. (3)) (Fig. 4h), which is consistent with previous studies and our experiments³⁷⁻³⁹. In the subsequent oxidization process, hydroxylation, deamination, and dehydrogenation reactions on the benzene ring were induced by $\bullet OH$ due to its high reactivity with strong electron-donating substituents (e.g., $-OH$ and $-NH_2$) to generate 2-aminobenzene-1,4-diol (P4), hydroquinone (P5), and benzoquinone (P6), while 1O_2 tends to attack the phenolic group with rich electrons (P6) and break up P6 into small molecular acids (P7-P8)⁴⁰. Finally, the generated byproducts are completely mineralized into CO_2 and

H₂O, which can be confirmed by TOC results.

Fig. R12 The degradation pathway of ONP in the LFCO/PMS/Vis system system. The green arrow represents the reduction stage induced by e⁻, The green arrow represents the oxidization stage induced by •OH and ¹O₂.

We hope these explanations can help you understand the relevant questions, and thank you again for your professional comments.

Fig. R11 corresponds to Supplementary Figure 38c;

Eqs. (1) ~ (4) corresponds to Eqs. (1) ~ (4) in the revised manuscript;

Fig. R12 corresponds to Figure 4h.

Reference:

1. Wu, X. *et al.* Outlook on Single Atom Catalysts for Persulfate-Based Advanced Oxidation. *ACS ES&T Eng.* **2**, 1776-1796, (2022).

Comment 15. In the experimental section, important details are missing: 1) source of CFC; 2) in the synthesis of LFCO@CFC, the ratio LFCO vs CFC is crucial for readers to reproduce the synthesis of the catalysts; 3) in the section degradation experiments the authors describe a system in which the amount of ONP solution is 50 mL; however, they do not mention any detail on the degradation in the flow system. Please add these details.

Response: Thank you for the professional question of the reviewer. Our system goes through three stages: pre-experiment stage, pilot-scale stage, and scale-up application, which are three parts that are very closely related to each other. The reaction of all three is the same reaction, which means that their reaction principle is the same. But in the subtle operation, the three are always more or less different.

① Pre-experiment in the beaker.

In this stage, LFCO powder was used to develop and optimize methods, determine the best process

route, have simple equipment and technical conditions, process flow, abundant and inexpensive chemical material sources, and the best degradation efficiency.

② Pilot-scale stage.

Subsequently, we conducted a pilot scale testing using LFCO@CFC as an FBR reactor, which is a necessary transitional step from small-scale experiments to industrial applications. Based on small-scale experimental research on feasible industrialization plans, we further investigate whether the process is mature and reasonable in a certain scale of scale-up devices, and solve problems that cannot be solved or discovered in the laboratory, providing a design basis for industrial applications. Although the essence of chemical reactions does not change with different experimental production, the optimal parameters for reactions may vary with external conditions such as experimental scale and equipment. For example, when the reaction is amplified, the issue of mass transfer is prominently exposed because of the increase in reaction space. Therefore, pilot scale-up is an important transitional stage from rapid and high-level to industrial application, and its level represents the level of industrialization.

③ Large-scale application.

Finally, 1:50 equal-scale amplification was used to achieve the real industrial significance of the economic scale of large-scale applications.

According to the reviewer's suggestion, we add the following information:

1. The commercial CFC was purchased from CETECH CO., LTD.;

2. in the synthesis of LFCO@CFC:

Preparation of different weight ratios of LFCO vs CFC. By changing the total amount of chemicals (including citric acid and metal nitrate) in the mixed solution, CFC with different loading amounts of LFCO were obtained. The weight of LFCO is determined by the difference of CFC and LFCO@LFC. Unless indicated, LFCO@LFC in the manuscript refers to the sample with 1.5 g LFCO.

3. Detail on the degradation experiments:

FBR Degradation experiments.

Experiments were performed in flow-through (dead-end) mode. Under light irradiation at room temperature (298 K), the thoroughly mixed ONP and PMS solution (500 mL) flowed through LFCO@CFC FRB by a peristaltic pump (30W) with no feed recirculation. The key system elements are as described below: The as-prepared LFCO@CFC (9 cm²) was inserted in a quartz glass tube to allow the transmission of light. A 350W Xenon lamp (CEL-HX F300, Beijing, China) with a 420 nm

and 780 nm cutoff filter was used as the light source and the light intensity is 560 mW cm^{-2} . The system was operated at different water fluxes and hydraulic residence times by controlling the pump flow rate. Unless indicated, 7500 mL h^{-1} flux, HRT 60 s, 298 K, initial pH 7.1, 30 ppm ONP, and 3 mM PMS (introduced with the ONP stock solution) were chosen as optimal conditions for all experiments. All samples were collected and analyzed by HPLC to calculate the ONP removal efficiency according to Eq. (1):

$$\text{Removal \%} = \frac{C_t}{C_0} \quad (1)$$

where C_t is the concentration of ONP at a certain reaction time (t) and C_0 refers to the initial concentration after adsorption equilibrium.

Ideally, the relationship between $\ln(C/C_0)$ and reaction time was matched well with the integral rate equation of the first-order reaction. Therefore, K_{obs} can be expressed by the pseudo-first-order kinetic model as described below:

$$\ln\left(\frac{C_t}{C_0}\right) = -K_{\text{obs}}t \quad (2)$$

The actual degradation rate ($\text{m}^{-3} \text{ s}^{-1}$) in a heterogeneous reaction can be calculated based on Eq. (3):

$$-r_{\text{ONP}}'' = -\frac{1}{A \times V} \times \frac{dN_{\text{ONP}}}{dt} = \frac{\text{ONP reacted}}{A \times V \times t} = \frac{1}{A \times V} \times Q_f \times c_f \times r_f \quad (3)$$

where A is the catalyst weight (g), V is the volume of carbon fiber felt, t is the reaction time (s), Q_f (L s^{-1}) is the feed flow rate, c_f (g L^{-1}) is the feed concentration of ONP, and r_f is the removal efficiency of ONP.

Beaker experiments

To facilitate the *in-situ* detection of reaction processes and intermediate products, we also used beaker experiments for ONP degradation in LFCO/PMS/Vis system. Unless indicated, 50 mL 30 ppm ONP solution including 0.3 mM PMS and 0.3 g L^{-1} powder catalyst under 560 mW cm^{-2} light irradiation at 298 K initial pH 7.1 were chosen as standard conditions for all beaker experiments. Before the light irradiation and adding PMS, the solution was stirred in the dark for 30 minutes to reach adsorption-desorption equilibrium. During the reaction, 4 mL solution was extracted at intervals and immediately filtered through an organic membrane filter ($0.22 \mu\text{m}$) for UV-Vis, TOC, HPLC, and UPLC-MS analysis. At the same time, the obtained solid catalysts after filtering were also further characterized.

We hope these explanations can help you understand the relevant questions, and thank you again for

your professional comments.

Minor points:

-Line 68: ONP definition should appear here instead of in line 71

Response: Thanks for the Reviewer's professional advice. As suggested by reviewers, we have modified the errors.

-Line 347: is rhodamine used along the MS?

Response: Thanks for your careful reading. As suggested by reviewers, we have deleted rhodamine in the revised version.

-Figure S1: typo: "Illustration"

Response: Thanks for the Reviewer's professional advice. As suggested by reviewers, we have modified the errors:

Supplementary Figure 1. The illustration of the fabrication of LFCO catalyst.

-The temperature of calcination should be coincident with the one described in the experimental section (line 357: 800 °C).

Response: Thanks for the Reviewer's professional advice. As suggested by reviewers, we have corrected the errors in the revised version:

Fig R. The illustration of the fabrication of LFCO catalyst.

-Figures S3 and S4: details on the ONP concentration are missing.

Response: Thanks for the Reviewer's professional advice. As suggested by reviewers, we have modified the errors:

- Supplementary Figure 3.** Degradation of ONP in $\text{LaFe}_{1-x}\text{Cu}_x\text{O}_3$ ($x = 0, 0.01, 0.03, 0.05, 0.10$) in PMS system under visible light irradiation ($\lambda > 420 \text{ nm}$, 50 mL 30 ppm ONP solution including 0.3 mM PMS and 0.4 g L^{-1} catalyst, $\text{pH} = 7.1$).
- Supplementary Figure 4.** Degradation of ONP with $\text{LaFe}_{1-x}\text{Cu}_x\text{O}_{3-\delta}$ ($x = 0, 0.01, 0.03, 0.05, 0.10$)

in Visible light and **b** PMS system ($\lambda > 420$ nm, 50 mL 20 ppm ONP solution including 0.3 g L^{-1} PMS and 0.4 g L^{-1} catalyst, pH = 7.1).

-The legend of Figure S4 does not correspond to graphs.

Response: Thanks for the Reviewer's professional advice. As suggested by reviewers, we have modified the errors:

Supplementary Figure 4. Degradation of ONP with $\text{LaFe}_{1-x}\text{Cu}_x\text{O}_{3-\delta}$ ($x = 0, 0.01, 0.03, 0.05, 0.10$) in **a** Visible light and **b** PMS system ($\lambda > 420$ nm, 50 mL 30 ppm ONP solution including 0.3 mM PMS and 0.4 g L^{-1} catalyst, pH = 7.1).

-References S23 and S35 are the same

Response: Thanks for the Reviewer's suggestion. We have deleted References S35 in the revised version.

REVIEWER COMMENTS

Reviewer #1 (Remarks to the Author):

The authors have revised the ms. intensively and I think it can be accepted at this point.

Reviewer #2 (Remarks to the Author):

The revision was improved according to the comments which have been addressed carefully, I recommend its acceptance for publication in Nature Communications due to the novelty and high quality.

Reviewer #3 (Remarks to the Author):

The authors tried to provide answers and justifications to the flaws of the systematic and routinary work conducted in this manuscript. However, these issues are still reflected in the manuscript and showcase disagreement with well-known and understood concepts in literature. Acceptance cannot be encouraged.

Reviewer #4 (Remarks to the Author):

The authors have address all the comments made. Upon reading how they have also addressed the responses to other reviewers my opinion is that the revised version of the MS has better quality. In the opinion of this reviewer the MS can now be accepted.

Response to the comments of the reviewers

Response to Reviewer 1 1
Response to Reviewer 2 2
Response to Reviewer 3 and additional comments of Reviewer 1..... 3
Response to Reviewer 4 26

Comments from Reviewer 1:

The authors have revised the ms. intensively and I think it can be accepted at this point.

Response: Thanks very much for your consideration on acceptance of our work.

Comments from Reviewer 2:

The revision was improved according to the comments which have been addressed carefully, I recommend its acceptance for publication in Nature Communications due to the novelty and high quality.

Response: Thanks very much for your consideration on acceptance of our work.

Comments from Reviewer 3:

The authors tried to provide answers and justifications to the flaws of the systematic and routinary work conducted in this manuscript. However, these issues are still reflected in the manuscript and showcase disagreement with well-known and understood concepts in literature. Acceptance cannot be encouraged.

Additional comments from Reviewer 1:

This work is thorough and commendable. However, I concur with Reviewer 3's critique regarding the authors' lack of clarity regarding the novelty of their approach. In the previous literatures, there were extensively explored methods for degrading nitroaromatic compounds, thus the proposed method appears to lack significant novelty and importance, based on the authors' response to Reviewer 3. The significance of research in this area lies in the potential for cost-effective large-scale application. Regrettably, this research falls short in this regard, as only indicated by the theoretical and computational analysis presented.

Response: Thank you sincerely for your valuable suggestions regarding the modifications made to this work. We analyzed and revised it in detail in the following responses according to your comments.

1. The novelty of the degradation mechanism proposed in this article

Nitroaromatic compounds (NACs) are aromatic compounds with one or more nitro groups ($-\text{NO}_2$). The commonly used NACs are depicted in Fig. R1¹. $-\text{NO}_2$ has certain unique properties that make NACs extensively used in different industries and are synthesized in large quantities, such as used as the raw materials in chemical synthesis of a variety of compounds like drugs, dyes, cosmetics, herbicides, pesticides, fungicides, explosives, paints, preservatives, antioxidants, gasoline additives, corrosion inhibitors, and other industrial chemicals. As the world's largest industrial country, China's industrial wastewater discharge accounts for about the country's total sewage discharge A quarter. This leads to the large discharge of NACs with industrial wastewater into the environment, thus contaminating water and soil as well as air, which present hazards to humans and other living organisms. Currently, approximately 65,000 distinct types of highly concentrated NACs have been identified in industrial wastewater. Hence,

NACs are listed as priority pollutants by the United States Environmental Protection Agency. Conventional processes such as physical and biological treatments have drawbacks such as low efficiency and secondary pollution, which make it difficult to meet the requirements of industrial wastewater discharge. **How to efficiently degrade NACs in industrial wastewater with complex water quality in practical application is a great challenge.**

[REDACTED]

Fig. R1 Structure of common nitroaromatic compounds¹

As viable environmental remediation technologies, advanced oxidation processes (AOPs) by activating peroxides to generate strong oxidizing free radicals (such as hydroxyl radical, ·OH) have been widely studied to oxidize the various toxic organic components in water bodies. **Notwithstanding AOPs may be able to oxidize various NACs, the degradation efficiency is still low due to two aspects:**

- (1) **AOPs cannot destroy the -NO₂ due to its passivation effect.** For example, a recent work published by *Proc. Natl. Acad. Sci. U.S.A.* has shown that the degradation dynamics of different pollutants are closely related to the structure of their parent pollutants (Fig. R2)². They classified the degradation rates (k_{obs} value) of 16 pollutants in different M-SACs (Fe, Co, Cu)/PMS systems and observed that there are three degradation kinetics trends for these pollutants in M-SACs/PMS systems. The pollutants with hydroxyl, amine, and other electron-donating groups (PCM, BPA, CP, NPX, SN, HBA, SMZ, SMT) are easily

oxidized in Fe-SAC/PMS system (k_{obs} value $> 0.4 \text{ min}^{-1}$). However, the oxidation efficiency of pollutants (NB, CPL, NBA) with electron-withdrawing groups (EWGs) such as $-\text{NO}_2$ is slower in Fe-SAC/PMS system (k_{obs} value $< 0.1 \text{ min}^{-1}$).

[REDACTED]

Fig. R2 (a) The k_{obs} values of different pollutants in M-SACs/PMS systems. (b) Comparison of other SACs/PMS systems for pollutants oxidation. (c) Linearity between the $\phi_{1/2}$ values of different pollutants and their $\ln k_{\text{obs}}$ values in M-SACs/PMS systems. (d) Linearity between the electrophilic indexes of different pollutants and their $\ln k_{\text{obs}}$ values in M-SACs/PMS systems. (e) Linearity between the nucleophilic indexes of different pollutants and their $\ln k_{\text{obs}}$ values in M-SACs/PMS systems².

Another work published by *Environ. Sci. Technol.* proposed that the presence of the EWGs (like X-F, Cl, Br, I; $-\text{NO}_2$; $-\text{COCH}_3$, $-\text{CN}$, etc.) induces an overall delocalization effect on the π electrons within the benzene ring structure, thereby augmenting the stability of the benzene ring and rendering the pollutants resistant to oxidation³. So they believed that EWGs are an immovable obstacle and practical challenges that need to be addressed. These findings are supported by many studies, like *Environ. Sci. Technol.*, 2023, 57, 13710; *Water Res.*, 2022, 218, 118453; *Environ. Sci. Technol.*, 2023, 4266; *ACS Catal.*, 2022,12, 58; *Water Res.*, 2022, 222 118959, and so on. Therefore, the

destruction of the $-\text{NO}_2$ is one of the most critical steps during NACs degradation, which is also consistent with our research background. Though these above studies analyzed the reasons for the low oxidative degradation efficiency of NACs, but did not provide reliable solutions.

(2) the effectiveness of AOPs is greatly affected by the water matrix. Despite the fundamentals of AOPs being well understood, industrial applications of AOPs are yet to adopt scalable approaches that allow more efficient, portable, cost-effective, and environment-friendly AOPs operations. In actual wastewater, co-existing water matrices including inorganic ions and natural organic matter (NOM) are commonly present and can greatly influence the efficiency, which can make AOPs unstable in practical engineering. Pedro and co-workers systematically compare radical and nonradical pathways in depth from seven perspectives, namely, degradation kinetics and mechanisms, water matrix interference, selectivity, temperature, pH, formation of halogenated byproducts, and electrical energy requirements, aiming to provide a strong empirical basis for choosing an appropriate approach to degrade a specific class of contaminants⁴. They found that radical-based oxidation with higher oxidation ability (i.e., $\text{SO}_4^{\cdot-}$ 2.5 ~ 3.1 V vs. NHE, $\cdot\text{OH}$ 1.8 ~ 2.7 V vs. NHE, respectively), but indiscriminateness, they are easily quenched by water matrix in industrial wastewater, resulting in ineffective NACs removal rate and limiting the large-scale application of AOPs. The measured redox potential of $^1\text{O}_2$ is 0.81 V vs. NH, which is significantly lower than that of. however, they have excellent independence of solution pH, strong resistance to versatile inorganic ions and background organic matters, which universally exist in domestic sewage and industrial wastewaters, and can significantly reduce toxic by-products. Also, nonradical oxidation processes can avoid the inefficient consumption of PMS, thus improving its stoichiometric efficiency. These characteristics fully indicate that nonradical-based catalytic oxidation reaction has a broad research and application prospect in the field of environmental pollution treatment⁵. At the same time, the reaction rate of organic pollutants with $\text{SO}_4^{\cdot-}$ and $\cdot\text{OH}$ is generally high ($10^7 \sim 10^9 \text{ M}^{-1} \text{ s}^{-1}$), and the reaction rate of organic pollutants with $^1\text{O}_2$ is low ($10^4 \sim 10^7 \text{ M}^{-1} \text{ s}^{-1}$), indicating that $^1\text{O}_2$ is difficult to act as the dominant ROS

in the degradation process of pollutants. However, in the degradation of certain phenols, although $\text{SO}_4^{\bullet-}$ and $\bullet\text{OH}$ are present in higher concentrations, the ROS that plays a dominant role in the system is $^1\text{O}_2$ rather than $\text{SO}_4^{\bullet-}$ and $\bullet\text{OH}$. Studies have shown that some ROS are more likely to attack certain special structures or groups of pollutants. For example, electron-rich phenol groups tend to react with $^1\text{O}_2$, while $\text{SO}_4^{\bullet-}$ tends to directly attack specific functional groups through electron transfer reactions, and tends to react with electron-donating groups such as amino ($-\text{NH}_2$), hydroxyl ($-\text{OH}$), and unsaturated bonds. **Therefore, how to mobilize and give full play to the reactivity and characteristics of different ROS is also our research focus.**

Solving these two problems at the same time is extremely challenging, and most of the researches on the degradation of NACs focus on a single oxidation system. **For the first time, we innovatively designed multiple catalysts with spatial separation ability to simultaneously activate reducing species (such as electrons) and oxidizing species (such as reactive radicals and non-radicals) at catalytic sites in different spatial locations. By regulating the catalyst structure and reaction conditions in the LFCO/PMS/Vis system, a ration-matched reduction/oxidation half-reaction rate can be achieved, which can inhibit the quenching of reactive species. Additionally, we make use of the complementary advantages of radicals and non-radicals to improve the degradation and mineralization efficiency of NACs under the environmental background.** These advantages result in an efficient degradation rate of 0.079 min^{-1} , which was 7.7 times higher than that of LFO (Fig. R3). We further elucidated the degradation pathway and determined the formation of key reduction products, providing direct evidence for a new mechanism (Fig. R4). Also, we believe that our groundbreaking work can provide new schemes for the effective degradation of other refractory organic pollutants with EWGs, such as brominated flame retardants (BFRs), organophosphorus flame retardants (PFRs), organochlorine pesticides (OCPs) and perfluoroalkylated substances (PFASs).

Fig. R3 The degradation efficiency of ONP in different systems. FBR conditions (unless indicated otherwise): $\lambda > 420$ nm radiation, Water flux 7500 mL h^{-1} , HRT 60 s, 298 K, initial pH 7.1, 30 ppm ONP, and 3 mM PMS (introduced with the ONP stock solution).

Fig. R4 The degradation pathway of ONP in the LFCO/PMS/Vis system. Conditions: $\lambda > 420$ nm, 50 mL 30 ppm ONP solution, 0.3 mM PMS, $0.3 \text{ g} \cdot \text{L}^{-1}$ powder catalyst, 298 K, initial pH 7.1.

2. The constructed reactor in this article has a higher practical value.

For nitroaromatic compounds (NACs), we face not only a lack of rapid and efficient degradation mechanisms but also challenges in achieving high removal efficiency in actual industrial applications, making its complete removal difficult. For the first time, we constructed a reduction-oxidation coupled degradation process and utilized the complementary benefits of free radicals and non-free radicals, which is an effective strategy for addressing this challenge. However, there is still a long way from practical application.

At present, most technologies concentrate on exploring catalytic performance and reaction mechanisms at the laboratory stage, leaving the understanding of relevant industrial application problems still incomplete. Maintaining the effective

concentration of radicals until reaching the target is a major obstacle in the development of highly efficient AOPs. From the beaker experiments under laboratory conditions, we find that LFCO achieves the highest PMS activation rate for generating high-concentration ROS, resulting nearly 100% ONP removal rate. This shows that our work has large-scale application potential. As lab-scale experimental conditions are significantly different from actual water/wastewater matrix and operational conditions, a successful laboratory treatment process does not guarantee its success on a larger scale. Pilot-scale studies are, thus, required to simulate real treatment processes before full-scale execution. A proper pilot-scale plant with a whole set of “real” operational parameters can provide an accurate portrait of a full-scale plant, albeit some cautions of the scaling must be taken into account.

Firstly, choosing a suitable reaction system is difficult. Flow chemistry has been proposed in modern organic chemistry as a means for process intensification, to improve the control over reaction performance, and to achieve higher yield. However, many open issues can be evidenced regarding the true possibility of scale-up, as well as the currently lacking information for process design and economic evaluation. Batch processes are commonly used in fine, specialty, and pharmaceutical chemistry due to their versatility, flexible production planning and scheduling. However, they are often difficult to scale up because of heat and mass transfer problems. In addition, they require significant intermediate storage capacity between process stages, resulting in large inventories of feedstock organic chemicals and sensitive intermediates. Continuous systems typically require smaller equipment volumes than batch ones and have a lower need for human intervention. Therefore, the possibility of continuous operations offers many advantages: lower costs, reduced waste, and decreased time-to-market for new drugs. Continuous flow reactors can deliver significantly higher yields, while solvent and energy waste can be decreased up to 90%⁶. Notable continuous-flow technologies employed for AOPs for the treatment of wastewater are fixed-bed reactors, which are energy-efficient, low-cost, easy to construct and set up, flexible to adapt to the fluctuation in influent water characteristics and flow rate, and simplicity of operation. This is why we chose CFC as a fixed bed. Also, a continuous-flow reactor

can in situ produce and dose of ROS, and provide flexible control over permeation channels, particularly interlayer channels, to balance selectivity and permeability requirements. Regarding practical wastewater treatment, continuous flow reactors can maximize reaction throughput by enhancing mass diffusion dynamics at the multiphase interface and minimizing waste generated during catalyst separation. **Therefore, we chose the continuous flow reactors to realize large-scale applications.**

Secondly, the scale-up strategy is one of the major hurdles from laboratory to application. There has been an extraordinary burst of recent research in photochemistry and photocatalysis driven in part by the environmentally benign appeal of light as a source of reactivity. However, many of the studies showcase small-scale reactions, and scale-up relies on a patchwork of different technologies that can require substantial trial and error to optimize. Such as mixing efficiency, heat, and mass transport phenomena in large batch reactors can hardly maintain the same values as those encountered in small batch reactors. And, more importantly, the distribution of photons becomes more problematic with increasing reactor dimensions due to the attenuation effect of photon transport (Bouguer-Lambert-Beer law). A work published by *Science* reported a combined software and hardware platform that iteratively determines optimal, substrate-specific conditions for photochemical processes in a scalable, flow-based architecture. The closed-loop Bayesian optimization approach enhances overall and space-time yields of a variety of distinct reactions⁷. Also, practical applications often require specialized reactors to provide the necessary reaction conditions for the catalyst. The core function of an AOP is its ability to produce ROS. The capability of transporting radicals from the point of production to the point of delivery as well as the capability of their efficient dosing will have an impact on the scalable application of different radicals for industrial purposes. While short half-lives are not problematic in AOPs (sometimes can be beneficial), this fact needs to be taken into consideration during the engineering design of continuous-flow reactors for scalable production of radical species. For example, radicals with short half-lives such as $\bullet\text{OH}$ (10 ~ 4 μsec) and $\text{SO}_4^{\bullet-}$ (30 ~ 40 μsec) must be produced in situ at the location of their application rather than transported after production. Only a few demonstrations of continuous flow

operations have been reported in the implementation of AOPs. For example, a work published in *Nature Nanotech.* performed the Pd₁-catalysed Suzuki coupling reaction in a packed-bed flow reactor, however, it exhibited a low productivity of ~0.3 g h⁻¹ due to the slow flow rates utilized⁸. This necessitates the synthesis of leaching-resistant single-atom catalysts (SACs) and the customization of flow reactors for SAC catalytic reactions, demanding high flow rates. **Consequently, we studied the degradation efficiency under different operating conditions in detail to ensure optimal conditions.**

Thirdly, cost-effectiveness is often overlooked in basic research. While the cost (i.e., energy consumption, chemical input, recovery of catalysts) were secondary concerns in demonstrating a new synthesis method, they became primary considerations during the transition from small-scale to large-scale application. Therefore, among the main obstacles hindering its adoption, the primary concerns revolve around scaling up photocatalytic reactions, recycling the catalyst, and purifying the product. To address this issue, Bolton and coworkers developed figures of merit for the comparison of advanced oxidation processes⁹. These are based on electrical energy consumption which often represents a major fraction of the AOP operating costs. For low contaminant concentrations (typically < 100 mg/L), the kinetics of destruction of organic contaminants by AOPs can often be described phenomenologically by simple pseudo-first-order rate expressions. Thus, the oxidant or energy dosage scales with the volume and treatment goals (i.e. orders of magnitude of reduction per unit volume). Consequently, the figure of merit for electrical-driven AOPs is defined as E_{EO} (electrical energy per order). This figure of merit was accepted by the International Union of Pure and Applied Chemistry (IUPAC) in 2001 and numerous E_{EO} values have been reported since then in literature for various oxidation processes and applications⁹. Giving a direct link to the electrical efficiency of the AOP, this approach allows not only for a simple comparison of different AOP technologies but also provides the requisite data for scale-up and economic as well as sustainability analyses for comparison with conventional treatment technologies (e.g., activated carbon adsorption, air stripping)¹⁰. **Thus, we used E_{EO} values in our work to evaluate the feasibility of the LFCO/PMS/Vis**

system.

In summary, it is imperative and equally important to systematically design and optimize various catalytic reaction devices compared with proposing a new mechanism. Designing poses challenges. In this work, we devote ourselves to the translation of new mechanisms into practical applications, endeavoring to design a reactor with high efficiency, low cost, and large throughput, and promote the further development of AOPs. During the design process of our reactor, its specific parameters and dimensions are determined based on industrial wastewater treatment standards as well as practical application considerations:

Initially, we refer to the concentration of nitrobenzene wastewater in actual water treatment plants to set the pollutant concentration treated by the reactor. Currently, the concentration range of nitrobenzene wastewater discharged by industries varies widely, typically ranging from 10 ppm to 50 ppm. The concentration standard set by the experiment aligns with actual water treatment standards, and the treated water quality meets the national discharge standard for first-class sewage. Secondly, based on parameter design references from several relevant photocatalytic pilot conversion devices, we establish three sizes and scales for gradual enlargement to assess degradation effects at various scales. A recent work published in *Nat. Commun.* showed a photocatalytic reactor with a throughput of 1000 L/m²/h, while that of Kian Ping Loh operates at a flow rate of 7.5 mL/min, consistent with the water flow parameters of our reactor design (Fig. R5)¹¹. The actual design size of our device is informed by various literature sources, with the majority utilizing plate reactors as liquid-filled reactors, enhancing pollutant-catalyst contact, improving degradation mineralization time, and enabling more complete degradation and removal of pollutants. For example, an innovative work published in *Nature* achieved a solar-to-hydrogen (STH) efficiency of up to 9.2% in practical large-scale applications by designing Indium gallium nitride photocatalyst reactors (4×4cm and 80×80cm), utilizing pure water, concentrated sunlight, and indium gallium nitride photocatalyst¹².

Fig. R5 Photo of the flow cell setup with a peristaltic pump and a temperature controller for SAC-catalysed reactions¹¹.

Based on the findings of our studies, we designed a continuous flow reactor utilizing the REDOX coupling mechanism and coordinating both free radicals and non-free radicals (Fig. R6).

Fig. R6 Practical applications of Fixed reaction bed reactor (FRB). **A** Schematic diagram of the FRB. **B** Photograph of experiment device (ONP as indicator pollutant). **C**, **D** Photograph (**C**) and SEM image (**D**) of the LFCO@CFC. **E** XRD of CFC and LFCO@CFC. **F** XPS spectra of the overall survey of LFCO@CFC. **G** The simplified schematic showing a cross-section view of ONP, $\bullet\text{OH}$, and $^1\text{O}_2$ in the photocatalytic membrane.

Then we optimized the operational parameters including water flux, hydraulic residence time, the dosage of catalyst and PMS, and light intensity (Fig. R7). Ultimately, reduction-oxidation coupling (ROC) degradation of ONP was achieved using the LFCO/PMS/Vis system, resulting in a degradation rate of 0.079 min^{-1} , which was 7.7

times higher than that of LFO.

Fig. R7 Removal of ONP under different operational parameters. **A, B** The removal efficiency and K_{obs} of ONP were investigated by changing water flux (**A**) and hydraulic retention time (**B**). **C** The removal efficiency and r'' of ONP were investigated for varying the weight ratio of LFCO vs CFC, the dosage of PMS, and visible light intensity in the FRB system. FBR conditions (unless indicated otherwise): $\lambda > 420$ nm radiation, Water flux 7500 mL h⁻¹, HRT 60 s, 298 K, initial pH 7.1, 30 ppm ONP, and 3 mM PMS (introduced with the ONP stock solution).

Inspired by the above encouraging removal performance of the FBR system, environmentally feasible and cost-effective for large-scale applications were then investigated. Predicting the residual structure and toxicity of degradation products is an important process to comparably evaluate their ecosystem risks. As shown in Fig. R8-9, the toxicity estimation software tool (T.E.S.T.) was used to calculate the toxicity indicators of all products, which includes the 50% lethal concentration of the product to organisms (Oral rat LD50), bioaccumulation factor (BAF), development toxicity and mutagenicity. It clearly shows that OAP (P1) has relatively higher ecosystem risks than ONP, but all the oxidized products (P6-P10) have low toxicity levels. This proves the importance of the sequential reduction-oxidization pathway of ONP degradation. At the same time, LFCO@CFC prevents catalyst leaching and ensures long-term material

stability, thereby demonstrating practical advantages for large-scale degradation of NACs.

Fig. R8 Thermal map of toxicity analysis.

Fig. R9 a Oral rat LD50, b Developmental, c Toxicity mutagenicity and d Bioaccumulation factor of ONP and its transformation products.

Finally, we assess the catalytic activity of the device under various conditions to determine its optimal application parameters, ensuring its performance in complex practical applications. Then, we analyze the energy cost of ONP degradation in FBR systems via EEO methods (defined as the electrical energy required to achieve primary ONP removal). The results indicate that the total cost of LFCO@CFC (0.23 KWHL⁻¹) is significantly lower than that of LFO@CFC (1.43 KWHL⁻¹), further confirming its feasibility for large-scale application. The cost of treating industrial wastewater using

LFCO@CFC fast reactor is approximately 13.72 yuan/ton, significantly lower than the market price (30 ~ 60 yuan/ton). The results clearly demonstrate that the LFCO@CFC FBR exhibits superior contaminant removal performance and lower treatment cost compared to most reported catalysts in the same research fields., while also possessing strong environmental remediation capabilities (Table R1).

Table R1 Performance and cost comparison of recently reported photocatalysts

Catalyst	k (min ⁻¹)	Cost (CNY)	Ref.
B _{0.05} -C ₃ N ₄	0.0213	44.57	13
Fe ₃ O ₄ /TiO ₂ /CuO	0.0206	35.27	14
Ce-PDMS-PbO ₂ /SS	0.0270	30.89	15
PdNCs/CoAl(O)/rGO	0.0377	27.54	16
Co-SrTiO ₃	0.0487	19.67	17
BiC-0.05	0.0139	51.72	18
LaFe _{0.95} Cu _{0.05} O ₃	0.0790	13.72	This work
La-SrTiO ₃	0.0412	21.19	17
Co-Bi ₂ O ₂ CO ₃	0.0349	34.63	19
Bi ₂ O ₃ /Bi ₂ O ₂ CO ₃	0.0213	42.91	20
LaFeO ₃	0.0115	47.39	This work
Fe-CN	0.0642	24.66	21
La _{0.08} MnO _{3-σ}	0.0813	16.89	22
La ₂ CuO ₄	0.0198	54.81	23
Zn _x Co-ZIFs	0.0447	41.23	24

There are some similar works in different research fields. A work published by *Angew. Chem. Int. Ed.* designed a continuous flow membrane unit that enables catalyst separation and solvent exchange between reactions, but with a minimum hydraulic retention time of 10 hours at a conversion rate of about 95%. Reducing the hydraulic retention time to 5 hours results in a sharp drop in conversion. It is also possible to achieve a conversion rate of more than 95% at a lower chemical load but at the cost of 53 hours of long operation²⁵. Another work from *Angew. Chem. Int. Ed.* evaluated the performance of photocatalytic C-N cross-coupling reaction in a 150ml plug flow reactor, requiring a residence time of 40 minutes to achieve a conversion rate of more than 90%,

and a conversion rate of more than 80% can be achieved in about 20 minutes, which is also difficult to achieve rapid degradation and transformation of products²⁶. A work published by *Nat. Commun.* designed the Pt-MoS catalytic system that can achieve REDOX conversion of a variety of compounds, but the flow reactor is only operated at a low flow rate (1 mLmin⁻¹) to achieve a yield of 65~ 99%¹¹. **By comparison, we proved that our work has certain advantages in actual running time, processing efficiency, and cost.**

3. The whole process experiment: from small equipment, scaled-up setup to pilot application.

In our manuscript, the excellent performance of the LFCO/PMS/Vis system encouraged us to move from laboratory-scale towards device modules. **Notably, pilot-scale photocatalytic panel reactors have been reported for photocatalytic water treatment. However, there is a lack of research on the immobilized reactor system for photo- Fenton water treatment.** Therefore, it is urgent to develop scalable reactor to validate the performance of the lab developed catalyst. Preserving the catalytic activity and structure stability of perovskites is a key factor in achieving long-term operational stability.

(1) Small equipment

The stability and durability of LFCO@CFC equipment were investigated by continuous experiments with time extended to 7 days (Fig. R10). The degradation efficiency of ONP remained > 99 % in the real industrial wastewater from Shanxi Coking Coal Group Co., LTD.. Besides, ICP-MS analysis showed that the leaching concentration of Cu and Fe was negligible (Table R2).

Fig. R10 Long-term operation performance of the LFCO@CFC FRB system (Illustration is the photograph of the used LFCO@CFC). FBR conditions: $\lambda > 420$ nm radiation, Water flux 7500 mL h^{-1} , HRT 60 s, 298 K, initial pH 7.1, 30 ppm ONP, and 3 mM PMS (introduced with the ONP stock solution).

Table R2 Total metal ions detected by ICP-MS during reaction in 7 days.

Time (day)	Fe (mg/L)	Cu (mg/L)
0	0.011	0.005
1	0.012	0.004
2	0.012	0.006
3	0.012	0.006
4	0.012	0.006
5	0.014	0.006
6	0.014	0.006
7	0.014	0.006

Furthermore, a clear consensus has been reached that the fast activity decay is predominantly rooted in surface chemical reactions and particle cracking of catalysts. To illustrate the stability of catalysts, we characterized the catalysts after reaction by XRD, TEM and XPS. The intensity/half peak width of XRD peaks is not only dependent on the particle size but also the content of the crystal phase. After a 7-day catalytic reaction, we can see that the used LFCO catalyst shows the same peaks with fresh LFCO (Fig. R11a), indicating that no phase transition during the reaction. As

shown in SEM results (Fig. R11b), the catalyst is still tightly bound to CFC, indicating the active components were not lost with time. Besides, TEM results of used LFCO catalysts show that they maintained the particle structure and size with a clean surface and uniform distribution of elements, which is well agreed with XRD data (Fig. R11c-d). Results together proved the structure stability of LFCO catalysts.

Fig. R11 a XRD, b SEM, c TEM images and corresponding EDS spectra of used LFCO.

XPS analysis of the used and fresh LFCO catalysts offered valuable information about the valence state and the surface composition. As displayed in Fig. R12 and Table R3, all elements remain in clear outline and unchanged ratio, demonstrating the excellent stability of LFCO. As shown in Fig. R12b-c, although Fe^{3+} ions are mainly in the form of structure for both catalysts, the binding energy of Fe 2p_{1/2} level in used LFCO is lower than that of fresh LFCO, while Cu^{2+} has the opposite trend. Charge transfer from partially reduced neighboring Cu atoms would increase the electron density on Fe causing the shift of binding energy. By following the redox cycle between

Fe and Cu, the superior catalytic activity and outstanding stability of the LFCO catalyst were perpetuated even after 7 days.

Fig. R12 a The La 3d, b Fe 2p, c Cu 2p, and d O 2p XPS spectra of fresh and used LFCO.

Table R3 Elements ratio of catalyst detected by XPS.

Catalyst	La (atomic %)	Cu (atomic %)	Fe (atomic %)	O (atomic %)
LFO	13.30	0	22.91	63.79
LFCO	14.76	4.61	17.36	63.27
LFCO-used	14.47	4.59	17.41	63.53

(2) Scaled-up setup

According to reviewer's suggestion, we further validate the application potential of the LFCO@CFC FBR system and added them in the revised version (marked yellow). As depicted in Fig. R13a and b, a larger LFCO@CFC fixed bed reactor (diameter: 8 cm, high: 16 cm) was constructed for large-scale degradation of ONP. Compared to the small equipment, the scaled-up reactor volume is expanded to 16 times and the flow rate is amplified twice (adjusted to 15 L/h). The field setup system

can also realize high stability and durability in 7 days (Fig. R13c). The degradation efficiency of ONP remained $> 97\%$ in the real industrial wastewater from Shanxi Coking Coal Group Co., LTD.. Besides, ICP-MS analysis showed that the leaching concentration of Cu and Fe was negligible. With a larger reactor size, the ONP degradation rate is lower than that of the small equipment, possibly due to the uneven light intensity.

Fig. R13 (a) Scaled-up setup of the LFCO@CFC FBR systems with the fixed bed reactor (diameter: 8 cm, high: 16 cm), Xe lamp, and peristaltic pump. (b) Structure diagram of the fixed bed reactor. (c) Long-term operation performance. Conditions (unless indicated otherwise): $\lambda > 420\text{ nm}$ radiation, Water flux 1500 mL h^{-1} , HRT 60 s , 298 K , initial pH 7.1 , 30 ppm ONP, and 3 mM PMS (introduced with the ONP stock solution).

(3) Pilot application

Finally, we equally scale amplified the laboratory pilot equipment according to

the ratio of 1:50 and the current plant treatment scale of ONP (Fig. R14). To prevent unexpected situations in the actual treatment, we increased the catalyst load to twice the original content; At the same time, various measures such as light intensity and coverage area were increased to ensure the treatment effect of the pilot plant. In terms of the operation cycle of the device, we have compared the cyclic experiment results in the laboratory to show that the material has strong stability performance, only 2% loss in the seven-day reaction cycle, and the reaction rate is maintained at nearly 100%, indicating that the material has very superior stability and removal effect. Therefore, we have reason to believe that after strengthening the setting of various optimal parameters and improving the catalyst load rate, the operation cycle can be maintained for 30 days.

Fig. R14 Diagram of the large-scale application plant including 5 main parts: (1) wastewater collection (2) PMS dosing tank, (3) physical filtration, (4) FBR reactor, (5) effluent tank (data from 1:50 for equal scale amplification). The dimensions of core parts 2 and 4 are marked in the enlarged image on the right.

As the core of the plant, 6 cylindrical FRBs are evenly distributed around the central light source (1326 W) for maximizing the utilization rate of light energy and reducing energy and investment costs. According to the cost calculation, LFCO@CFC FBR costs about 13.72 CNY/ton for industrial wastewater treatment, which is much lower than the market price (30 ~ 60 CNY/ton). In addition, we prepared catalysts that have been widely studied in photo-assisted PMS activation systems and tested their activities for comparison. The results clearly show that the LFCO@CFC FBR outperformed most of

the recently reported catalysts' performance and lower processing costs, showing strong environmental remediation capabilities. **Overall, the above results successfully demonstrate the high performance and application potential of LFCO catalyst from small equipment, scaled-up setup to pilot application.**

We hope these explanations can help you understand the relevant questions, and thank you again for your professional comments.

Fig. R3 corresponds to Figure 3D; Fig. R4 corresponds to Figure 4H;

Fig. R6 corresponds to Figure 1; Fig. R7 corresponds to Figure 3A-C;

Fig. R8 corresponds to Figure 5A; Fig. R9 corresponds to Supplementary Figure 44;

Fig. R10 corresponds to Figure 3E; Fig. R11 corresponds to Supplementary Figure 17 ;

Fig. R12 corresponds to Supplementary Figure 18;

Fig. R13 corresponds to Supplementary Figure 45;

Fig. R14 corresponds to Figure 5E ;

Table R1 corresponds to Supplementary Table 11;

Table R2 corresponds to Supplementary Table 10

References:

- 1 Tiwari, J. *et al.* Environmental persistence, hazard, and mitigation challenges of nitroaromatic compounds. *Environ. Sci. Pollut. R.* **26**, 28650-28667, (2019).
- 2 Guo, J. *et al.* Fenton-like activity and pathway modulation via single-atom sites and pollutants comediate the electron transfer process. *Proc. Natl. Acad. Sci. U.S.A.* **121**, e2313387121, (2024).
- 3 Xie, Z.-H. *et al.* Effects of Molecular Structure on Organic Contaminants' Degradation Efficiency and Dominant ROS in the Advanced Oxidation Process with Multiple ROS. *Environ. Sci. Technol.* **56**, 8784-8795, (2022).
- 4 Yan, Y. *et al.* Merits and Limitations of Radical vs. Nonradical Pathways in Persulfate-Based Advanced Oxidation Processes. *Environ. Sci. Technol.* **57**,

- 12153-12179, (2023).
- 5 Zhou, X. *et al.* Nonradical oxidation processes in PMS-based heterogeneous catalytic system: Generation, identification, oxidation characteristics, challenges response and application prospects. *Chem. Eng. J.* **410**, 128312, (2021).
- 6 Rossetti, I. *et al.* Chemical reaction engineering, process design and scale-up issues at the frontier of synthesis: Flow chemistry. *Chem. Eng. J.* **296**, 56-70, (2016).
- 7 Slattery, A. *et al.* Automated self-optimization, intensification, and scale-up of photocatalysis in flow. *Science* **383**, 6681, (2024).
- 8 Chen, Z. *et al.* A heterogeneous single-atom palladium catalyst surpassing homogeneous systems for Suzuki coupling. *Nat. Nanotechnol.* **13**, 702-707, (2018).
- 9 Bolton, J. R. *et al.* Figures-of-merit for the technical development and application of advanced oxidation technologies for both electric-and solar-driven systems (IUPAC Technical Report). *Pure Appl. Chem.* **73**, 627-637, (2001).
- 10 Miklos, D. B. *et al.* Evaluation of advanced oxidation processes for water and wastewater treatment—A critical review. *Water Res.* **139**, 118-131, (2018).
- 11 Chen, Z. *et al.* Addressing the quantitative conversion bottleneck in single-atom catalysis. *Nat. Commun.* **13**, 2807, (2022).
- 12 Zhou, P. *et al.* Solar-to-hydrogen efficiency of more than 9% in photocatalytic water splitting. *Nature* **613**, 66-70, (2023).
- 13 Zhan, H. *et al.* Photocatalytic O₂ activation and reactive oxygen species evolution by surface B-N bond for organic pollutants degradation. *Appl. Catal. B: Environ.* **310**, 121329, (2022).
- 14 Kianfar, A. H. *et al.* Synthesis, characterization and investigation of photocatalytic and catalytic applications of Fe₃O₄/TiO₂/CuO nanoparticles for degradation of MB and reduction of nitrophenols. *J. Environ. Chem. Eng.* **8**, (2020).

- 15 Li, H. *et al.* Ultrasonic-electrodeposition construction of high hydrophobic Ce-PDMS-PbO₂/SS electrode for p-nitrophenol degradation: Catalytic, kinetics and mechanism. *Appl. Catal. B: Environ.* **335**, 122884, (2023).
- 16 Wang, Q. *et al.* Hierarchical-Structured Pd Nanoclusters Catalysts x-PdNCs/CoAlO/rGO-T by the Captopril-Capped Pd Cluster Precursor Method for the Highly Efficient 4-Nitrophenol Reduction. *ACS Appl. Mater. Interfaces*, **14**, 27775-27790, (2022).
- 17 Zhang, D. *et al.* Dynamic active-site induced by host-guest interactions boost the Fenton-like reaction for organic wastewater treatment. *Nat. Commun.* **14**, 3538, (2023).
- 18 Zhou, Q. *et al.* Novel hierarchical carbon quantum dots-decorated BiOCl nanosheet/carbonized eggshell membrane composites for improved removal of organic contaminants from water via synergistic adsorption and photocatalysis. *Chem. Eng. J.* **420**, 129582, (2021).
- 19 Zhao, D. X. *et al.* Efficient visible-light-driven Suzuki coupling reaction over Co-doped BiOCl/Ce-doped Bi₂O₂CO₃ composites. *Green Chem.* **23**, 1823-1833.
- 20 Huang, Y. *et al.* Visible light Bi₂S₃/Bi₂O₃/Bi₂O₂CO₃ photocatalyst for effective degradation of organic pollutions. *Appl. Catal. B: Environ.* **185**, 68-76, (2016).
- 21 Li, H. *et al.* Fe(III)-Doped g-C(3)N(4) Mediated Peroxymonosulfate Activation for Selective Degradation of Phenolic Compounds via High-Valent Iron-Oxo Species. *Environ. Sci. Technol.* **52**, 2197-2205, (2018).
- 22 Liu, X. *et al.* In Situ Modulation of A-Site Vacancies in LaMnO_{3.15} Perovskite for Surface Lattice Oxygen Activation and Boosted Redox Reactions. *Angew. Chem. Int. Ed.* **60**, 26747-26754, (2021).
- 23 Chen, H. *et al.* Understanding oxygen-deficient La₂CuO_{4-δ}perovskite activated peroxy-monosulfate for bisphenol A degradation: The role of localized electron within oxygen vacancy. *Appl. Catal. B: Environ.* **284**, 119732, (2021).
- 24 Chen, Z. *et al.* Single-atom Mo-Co catalyst with low biotoxicity for sustainable degradation of high-ionization-potential organic pollutants. *Proc. Natl. Acad. Sci. U.S.A.* **120**, e2305933120, (2023).

- 25 Corcoran, E. B. *et al.* Photon Equivalents as a Parameter for Scaling Photoredox Reactions in Flow: Translation of Photocatalytic C-N Cross-Coupling from Lab Scale to Multikilogram Scale. *Angew. Chem. Int. Ed.* **59**, 11964-11968, (2020).
- 26 Peeva, L. *et al.* Continuous Consecutive Reactions with Inter-Reaction Solvent Exchange by Membrane Separation. *Angew. Chem. Int. Ed.* **55**, 13576-13579, (2016).

Comments from Reviewer 4:

The authors have address all the comments made. Upon reading how they have also addressed the responses to other reviewers my opinion is that the revised version of the MS has better quality. In the opinion of this reviewer the MS can now be accepted.

Response: Thanks very much for your consideration on acceptance of our work.

REVIEWER COMMENTS

Reviewer #1 (Remarks to the Author):

The authors highly addressed the raised questions and novelty, and I think it can be accepted at this point.